# 2′−5′ oligoadenylate synthetase-like 1 (OASL1) protects against atherosclerosis by maintaining endothelial nitric oxide synthase mRNA stability

Tae Kyeong Kim[1,10], Sejin Jeon[1,8,10], Seonjun Park [2], Seong-Keun Sonn[1], Seungwoon Seo[1], Joowon Suh[1], Jing Jin[1], Hyae Yon Kweon[1], Sinai Kim[1], Shin Hye Moon[1], Okhee Kweon[1], Bon-Hyeock Koo[3,9], Nayoung Kim [4,5], Hae-Ock Lee [4,5], Young-Myeong Kim [6], Young-Joon Kim[7], Sung Ho Park[2] & Goo Taeg Oh [1] ✉

Endothelial nitric oxide synthase (eNOS) decreases following inflammatory stimulation. As a master regulator of endothelial homeostasis, maintaining optimal eNOS levels is important during cardiovascular events. However, little is known regarding the mechanism of eNOS protection. In this study, we demonstrate a regulatory role for endothelial expression of 2′−5′ oligoadenylate synthetase-like 1 (OASL1) in maintaining eNOS mRNA stability during athero-prone conditions and consider its clinical implications. A lack of endothelial *Oasl1* accelerated plaque progression, which was preceded by endothelial dysfunction, elevated vascular inflammation, and decreased NO bioavailability following impaired eNOS expression. Mechanistically, knockdown of PI3K/Akt signaling-dependent *OASL* expression increased Erk1/2 and NF-κB activation and decreased *NOS3* (gene name for eNOS) mRNA expression through upregulation of the negative regulatory, miR-584, whereas a miR-584 inhibitor rescued the effects of *OASL* knockdown. These results suggest that OASL1/OASL regulates endothelial biology by protecting *NOS3* mRNA and targeting miR-584 represents a rational therapeutic strategy for eNOS maintenance in vascular disease.

The endothelium performs a gate-keeper role in the blood vessel wall by restricting the passage of immune cells and bio-reactive molecules, thereby protecting vessels from inflammation[1,2]. Atherosclerosis develops through inflammatory cross-talk between aortic endothelial cells (ECs) and immune cells, which is followed by EC activation and leukocyte recruitment[3]. A number of phenomena from endothelial dysfunction to atherosclerosis development occur preferentially at susceptible sites, such as arterial curvatures or branch points, where blood flow is slower or oscillatory, thus creating a favorable

environment for plaque formation[4]. The maintenance of appropriate endothelial nitric oxide synthase (eNOS, encoded by *NOS3*) and nitric oxide (NO) levels is a hallmark of the normal EC state and is essential for cardiovascular homeostasis[5,6]. Various pathological stimuli can downregulate eNOS expression through multiple regulatory mechanisms, which results in EC dysfunction[6,7]. In areas around locations occupied by dysfunctional ECs exhibiting decreased eNOS expression, vascular inflammation and leukocyte infiltration are increased and exacerbate lesion development[4,8]. A major contributing factor to

reduced eNOS levels is the loss of *NOS3* mRNA stability at the post-transcriptional stage[9–11]. The recent use of mRNA-based vaccines for Covid-19 prevention has attracted attention toward RNA-based therapies. This has also renewed interest in targeting microRNAs (miRNAs), which are involved in the regulation of mRNA stability[12–14]. In fact, it has been demonstrated that the 3′-UTR region of *NOS3* mRNA is targeted and down-regulated by several miRNAs[15–17]. However, putative protective factors that regulate *NOS3* mRNA to maintain consistent levels under athero-prone conditions remain unknown.

Mouse 2′−5′ oligoadenylate synthetase-like 1 (OASL1), an interferon (IFN)-stimulated gene, participates in the regulation of inflammation-associated gene expression as a post-transcriptional and translational modulator[18,19]. We previously demonstrated that OASL1 inhibits translation of IFN-regulatory factor 7 (*Irf7*) mRNA, creating a negative feedback loop that prevents excessive type I IFN signaling[18]. OASL1 acts through its RNA-binding activity to control viral RNA following stress granule formation[19]. OASL1 expression is induced by immunological stimuli and it plays a role in inflammatory diseases, such as cancer and multiple sclerosis[20,21]. Recently, single-cell RNA sequencing (scRNA-seq) analyses revealed that *Oasl1* transcripts are expressed by aortic ECs rather than other known vascular cell types[22,23]; however, little is known regarding the role of OASL1 in inflammatory vascular diseases such as atherosclerosis.

In this study, we examined the role of mouse OASL1 and its human ortholog, OASL, in atherosclerosis. We demonstrated that OASL and OASL1 are expressed in ECs in athero-prone regions. We also found that the absence of *Oasl1* in *Apoe*[−/−] mice (*Oasl1*[−/−]*Apoe*[−/−] mice) enhances the leukocyte infiltration and development of plaques, which are preceded by formation of dysfunctional ECs. These effects reflect the regulatory role of OASL1 in NO synthesis and the inflammatory response. Similar to OASL1 in aortic ECs, human *NOS3* levels were dependent upon endothelial OASL expression, which in turn, was mediated by phosphoinositide 3-kinase (PI3K)/Akt signaling. Mechanistically, *OASL* silencing decreased *NOS3* mRNA stability through dysregulation of the post-transcriptional repressor, miR-584. Our results suggest that OASL functions as a homeostatic mediator in ECs through regulation of the miR-584-eNOS axis and may lead to new treatment strategies for atherosclerosis and other cardiovascular disorders mediated through endothelial dysfunction.

## Results

### Human OASL and murine OASL1 are expressed in aortic ECs within athero-prone regions

To determine whether OASL expression is associated with atherosclerosis, we measured OASL expression levels in atheroma tissue with or without plaques. Human *OASL* mRNA levels were higher in atherosclerotic aorta tissues with plaques compared with those without plaques (Fig. 1a). Immunofluorescence (IF) and immunoblot analyses confirmed that OASL was primarily expressed in the ECs of plaque-containing atheroma tissues (Fig. 1b, c, Supplementary Fig. 1b). Given the potential role of OASL in human vascular disease, we examined OASL expression in the GTEx (Genotype-Tissue Expression) database and found that OASL was clearly expressed in EC-enriched normal human tissues, including the aorta and coronary artery (Supplementary Fig. 1a). Similar to the results for human OASL, the expression of murine *Oasl1* transcripts in the aorta of *Apoe*[−/−] mice with advanced atherosclerosis was altered in plaque-containing atherosclerotic aortas compared with those without plaques (Fig. 1d). An *en face* IF analysis of atherosclerotic aortas from *Apoe*[−/−] mice confirmed that OASL1 was primarily expressed in activated ECs, but not in other vascular cell types associated with plaques (Fig. 1e, Supplementary Fig. 1c). Collectively, these findings indicate that both human OASL and murine OASL1 exhibit inducible expression in aortic ECs under pathogenic conditions.

Because the role of ECs is important from the early stages of atherosclerosis and their characteristics are known to vary by

location, we measured the changes in Oasl1 expression in both athero-prone and athero-resistant aortic sites. *Oasl1* mRNA levels were significantly higher in aortas from *Apoe*[−/−] mice compared with those from *Apoe*[+/+] mice (Fig. 1f). Next, we examined the co-localization of OASL1 expression with CD31, an EC marker. High OASL1 levels were observed in CD31[+] ECs in athero-prone areas compared with atheroprotective areas in both *Apoe*[−/−] and *Apoe*[+/+] mice (Fig. 1g). These athero-prone regions included the lesser curvature of the aortic arch and the branch point of arteries. The mean fluorescence intensity of (MFI) of OASL1 in the CD31[+] ECs of *Apoe*[−/−] mice was approximately 50% higher and nearly twice as high, respectively, compared with that in *Apoe*[+/+] mice (Fig. 1g), suggesting athero-prone, site-specific endothelial expression of OASL1.

Moreover, activation of human umbilical vein ECs (HUVECs) and human arterial ECs (HUAECs) by stimulation with atheroprone shear stress (Supplementary Fig. 1d), or tumor necrosis factor alpha (TNFα) and interferon gamma (IFNγ) markedly increase *OASL* mRNA (Fig. 1h) and protein levels (Fig. 1i). These findings were replicated in mouse aortic EC (MAECs), which exhibited increased *Oasl1* mRNA (Fig. 1j) and protein (Fig. 1k) following treatment with TNFα and IFNγ. The observed human OASL and murine OASL1 expression in athero-prone aortic ECs suggests their participation in the development of atherosclerotic lesions, particularly in susceptible areas of the aorta.

### *Oasl1* deficiency exacerbates endothelial dysfunction and atherosclerosis in athero-prone regions of the aorta

To determine whether OASL1 contributes to the progression of atherosclerosis, we assessed atherosclerotic plaque formation using a region-specific analysis of *en face* preparations of whole aortas from *Oasl1*[−/−]*Apoe*[−/−] and *Apoe*[−/−] mice. *Oasl1*[−/−]*Apoe*[−/−] mice exhibited increased atherosclerotic plaque development in the whole aorta (Fig. 2a) without a change in the plasma profile of lipoproteins associated with atherogenesis (Supplementary Table 1). Atherosclerotic plaques were larger in athero-prone regions including the aortic arch and abdominal aorta (Fig. 2a) in *Oasl1*[−/−]*Apoe*[−/−] mice compared with *Apoe*[−/−] mice. To identify the mechanism by which *Oasl1* deficiency contributes to atherogenesis, we conducted an scRNA-seq analyses of athero-prone aortic arches that exhibited the greatest plaque development in response to *Oasl1* deficiency (Fig. 2b, Supplementary Fig. 2). These analyses revealed that *Oasl1* deficiency increased the number of aortic leukocytes, including macrophages and T cells while reduced smooth muscle cell contents, which was considered as secondary phenotype of plaque progression (Fig. 2c). A Gene Set Enrichment Analysis (GSEA) of the total aorta revealed that terms enriched by *Oasl1* deficiency were associated with the augmentation of the immune or inflammatory response to stress, cell adhesion, and migration, whereas those that were decreased included the regulation of blood circulation-mediated processes (Fig. 2d). Similarly, *Oasl1* deletion markedly increased the mRNA levels of the EC-expressed adhesion molecules, *Icam1*, *Selplg*, and *Sele* in the aortas of *Apoe*[−/−] mice compared with control mice (Fig. 2e, Supplementary Fig. 3). A flow cytometry analysis of atherosclerotic aortic single cells (Supplementary Fig. 4) supported these results, indicating that the number of CD45[+] leukocytes was higher in *Oasl1*[−/−]*Apoe*[−/−] aortas compared with those in littermate controls (Fig. 2f). Of note, there was a significant increase in the number of macrophages and neutrophils in arteries from *Oasl1*[−/−]*Apoe*[−/−] mice compared with those from *Apoe*[−/−] mice (Fig. 2g). Taken together, these data suggest increased adhesion and infiltration of leukocytes into atherosclerotic aortas of *Oasl1*[−/−]*Apoe*[−/−] mice, which suggests that *Oasl1* deficiency triggers excessive endothelial activation and vascular inflammation during atherogenesis.

To further explore whether endothelial activation during *Oasl1* deficiency reflects EC dysfunction before the onset of plaque formation, we assessed blood flow dynamics as a function of each aortic site in *Oasl1*[−/−]*Apoe*[−/−] and *Apoe*[−/−] mice by serial echocardiography

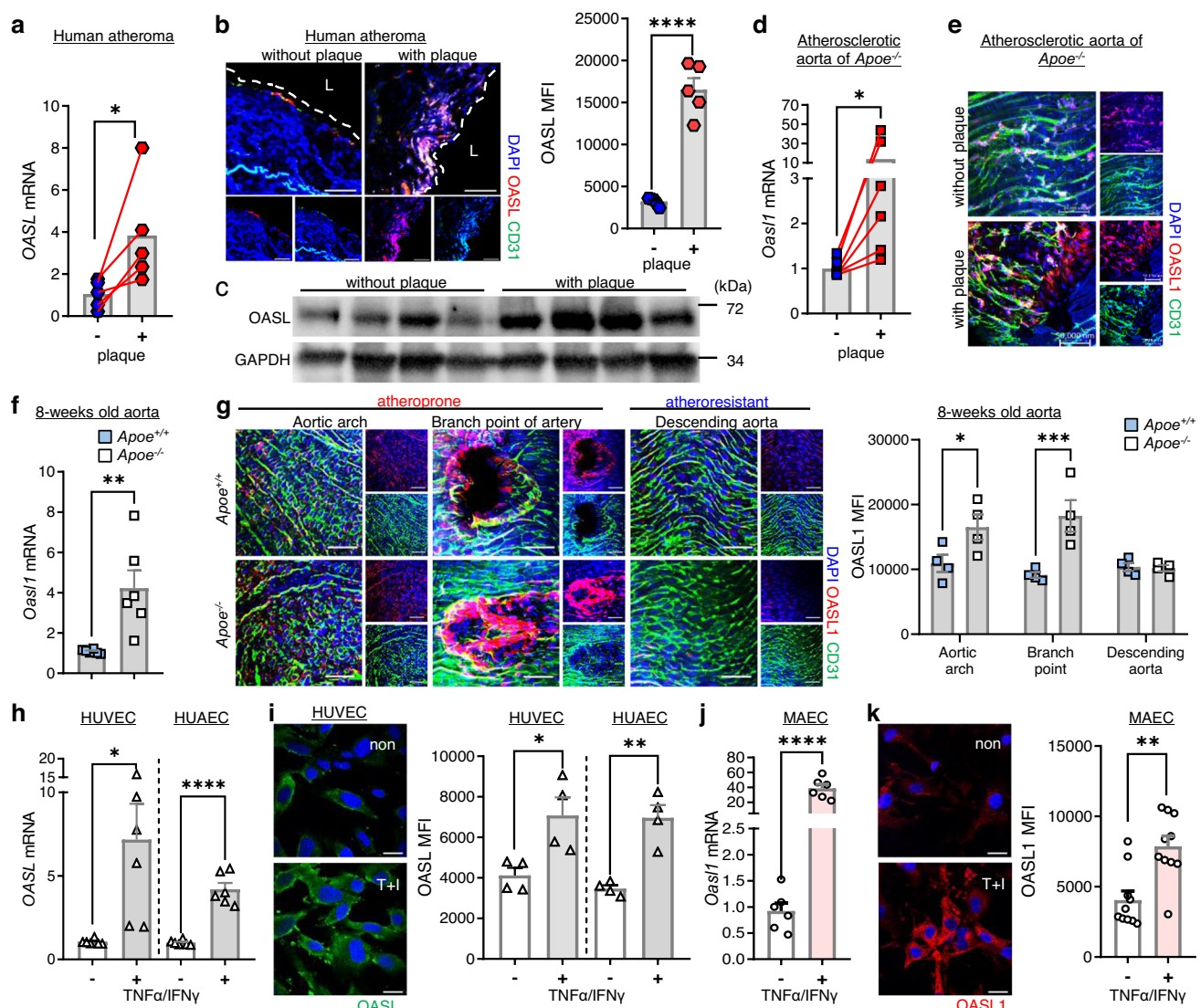

**Fig. 1 | Expression of human OASL and murine OASL1 are increased in athero-prone aortic ECs. a–c** Human atheroma tissues with or without plaques were used to measure OASL expression. **a** *OASL* mRNA levels ($p = 0.0463$, $n = 5$) were measured by quantitative PCR. **b** Left: Immunofluorescence (IF) staining of OASL protein and CD31+ ECs ($p < 0.0001$, $n = 5$ per group). Right: Quantitation of mean fluorescence intensity (MFI). Scale bar, 50 μm. L, lumen. (**c**) Western blot analysis of OASL expression ($n = 4$ per group). **d, e** Atherosclerotic aortas were isolated from *Apoe*−/− mice fed a normal chow diet (NCD) for 28 weeks. *Oasl1* mRNA levels ($p = 0.0313$, $n = 6$) (**d**) and *en face* IF staining for OASL1 and CD31+ aortic ECs ($n = 5$ per group) (**e**) in matched aortic specimens with and without plaques are shown. Scale bar, 50 μm. **f, g** Aorta tissues were isolated from *Apoe*+/+ and *Apoe*−/− mice fed an NCD for 8 weeks. **f** mRNA levels of *Oasl1* in aortas of each group ($p = 0.0055$, $n = 6$ per group) as measured by quantitative PCR. **g** Left: *En face* IF staining for OASL1 and CD31+ ECs in athero-prone or athero-resistant regions. Right:

Quantitation of MFI (arch: $p = 0.036$, branch: $p = 0.0008$, descending: $p = 0.9994$, $n = 4$ per group). Scale bar, 50 μm. **h, i** *OASL* mRNA level (HUVEC: $p = 0.017$, HUAEC: $p < 0.0001$, $n = 6$ per group) (**h**) and immunocytochemical detection of OASL protein (HUVEC: $p = 0.0222$, HUAEC: $p = 0.0017$, $n = 4$ per group) (**i**) in HUVECs and HUAECs stimulated with TNFα and IFNγ. Scale bar, 20 μm. **j, k** *Oasl1* mRNA level ($p < 0.0001$, $n = 6$ per group) (**j**) and immunocytochemical detection of OASL1 expression ($p = 0.001$, $n = 10$ per group) (**k**) in MAECs stimulated with TNFα and IFNγ. Scale bar, 20 μm. Data in **b, e, g, i** and **k** are representative of at least 3 independent experiments. Data are presented as the means ± SEMs (*$p \le 0.05$, **$p \le 0.01$, ***$p \le 0.001$, ****$p \le 0.0001$; **a**, two-sided paired *t*-test; **d**, two-sided Wilcoxon matched-pairs signed rank test; **b, f, h–k**, two-sided unpaired Student's *t*-test; **g**, two-way ANOVA with Sidak's test for multiple comparisons). Source data are provided as a Source Data file.

(Supplementary Fig. 5a)[24]. *Oasl1* deficiency resulted in decreased flow velocity and wall shear stress (WSS) in athero-prone regions, such as the lesser curvature of the aortic arch and abdominal branch point in *Apoe*−/− mice compared with littermate *Apoe*−/− controls (Supplementary Fig. 5b, c). These phenomena were observed from 8 to 20 weeks of normal chow diet (NCD) and disappeared at 28 weeks when the plaques had developed and the aortic diameter was reduced (Fig. 2a, Supplementary Fig. 5). Collectively, these findings demonstrate that the loss of *Oasl1* promotes endothelial dysfunction−dependent atherosclerotic plaque formation, particularly in athero-susceptible regions of the aorta.

To confirm the endothelial specificity of atherogenesis, we generated chimeric *Oasl1*−/−*Apoe*−/− mice with a vascular EC-specific deficiency of *Oasl1* by transplanting *Apoe*−/− bone marrow (BM) into *Oasl1*−/−*Apoe*−/− recipient mice (Vc*Oasl1*−/−-*Apoe*−/− mice) (Fig. 3a) and subsequently crossing *Oasl1*-floxed (*Oasl1*fl/fl) *Apoe*−/− mice with *Tie2*-Cre transgenic mice to generate *Oasl1*fl/fl *Tie2*-cre+ *Apoe*−/− mice (Fig. 3c). An *en face* analysis revealed that plaque formation was increased in whole aortas of Vc*Oasl1*−/−-*Apoe*−/− mice compared with those of *Apoe*−/− chimeric mice (Fig. 3b) whereas BM*Oasl1*−/−-*Apoe*−/− chimeric mice exhibited no differences. In addition, plaques were significantly larger in the *Oasl1*fl/fl *Tie2*-cre+ *Apoe*−/− mice compared

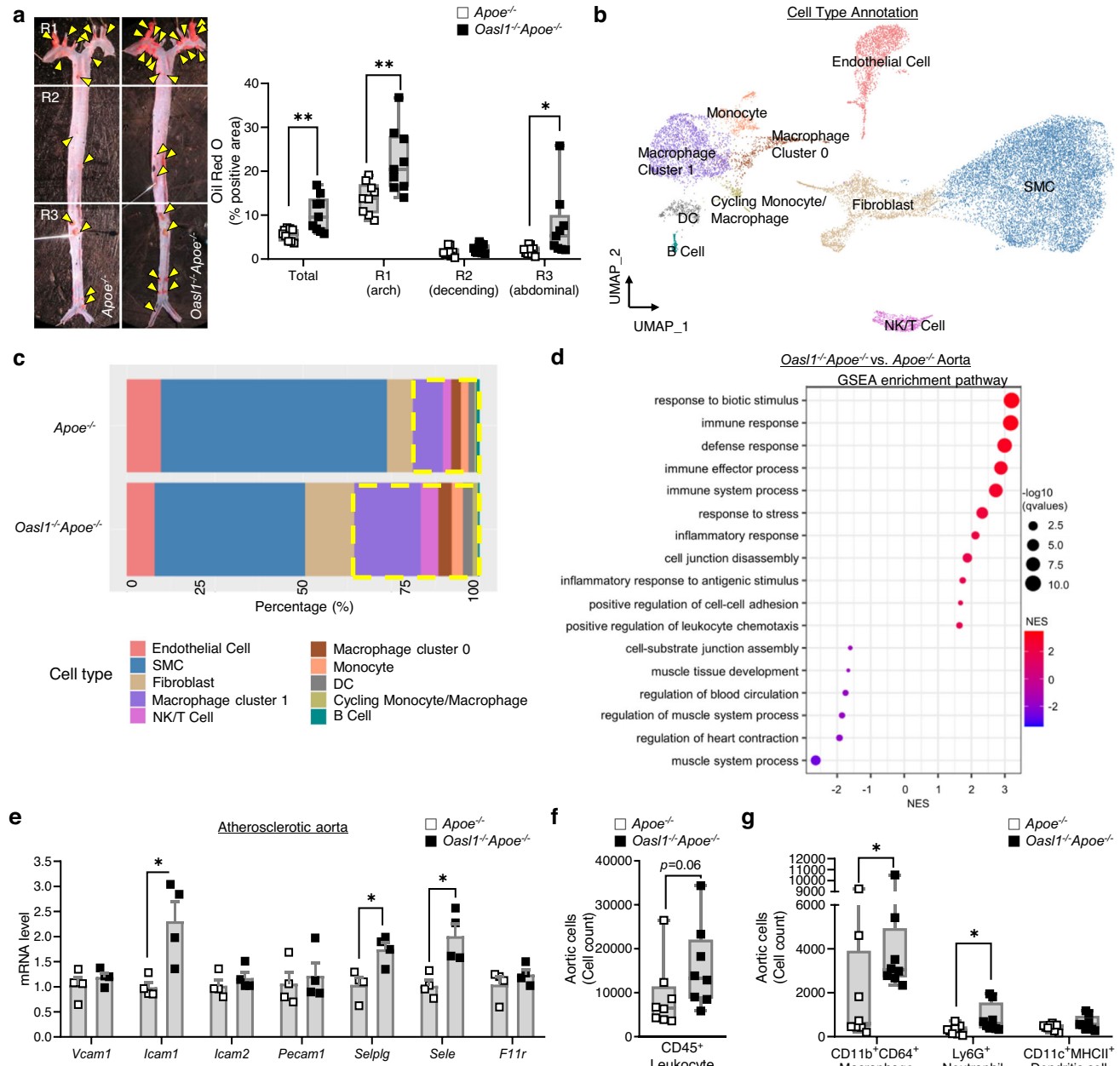

**Fig. 2 | *Oasl1* deficiency increases endothelial dysfunction, leading to vascular inflammation and leukocyte infiltration into the lesion.** *Apoe⁻/⁻* and *Oasl1⁻/⁻* *Apoe⁻/⁻* mice were fed a normal chow diet (NCD) for 28 weeks to allow athero-sclerotic conditions to develop. **a** Left: Oil red O-stained lesions in whole aortas. Right: Quantitation of stained areas in *en face* preparations (Total: *p* = 0.0040, R1: *p* = 0.0071, R2: *p* = 0.1175, R3: *p* = 0.0484, *n* = 9 per group). Yellow arrow-head: plaque in aorta. **b**–**d** Analysis of the transcriptomes of single cells from atherosclerotic aortas using the 10x Genomics platform. **b** Uniform manifold approximation and projection (UMAP) plot of 20,876 total aortic cells, colored by clusters. **c** Bar graphs showing the relative proportion of each cell type per group. **d** Dot plot showing Gene Set Enrichment Analysis (GSEA) according to the normalized enrichment score (NES) for Gene Ontology (GO) terms of aortas from *Oasl1⁻/⁻Apoe⁻/⁻* versus

the littermate control *Oasl1^fl/fl^Apoe⁻/⁻* mice. **e** Quantitative PCR analysis of the adhesion-related molecules in atherosclerotic aortas (*Vcam1*, *Pecam1*, *F11r*: *p* = 0.4857, *Icam1*, *Selplg*, *Sele*: *p* = 0.0286, *Icam2*: *p* = 0.2000, *n* = 4 per group). **f**, **g** Flow cytometry of single cells obtained from atherosclerotic aortas (*n* = 8 per group), including total leukocytes (**f**) and macrophages (*p* = 0.0499), neutrophils (*p* = 0.0433), and dendritic cells (DC; *p* = 0.1755) (**g**). Data in **a** are representative of each group. **a**, **f** and **g**, Box plots are shown as median of each value and the interquartile range (IQR, the range between the 25th and 75th percentiles); whiskers indicate 1.5 times the IQR. Data are pre-sented as the means ± SEMs (**p* ≤ 0.05, ***p* ≤ 0.01; **a**, **f** and **g**, two-sided unpaired Student's *t*-test; **e**, two-sided Mann-Whitney *U* test). Source data are provided as a Source Data file.

with the littermate control *Oasl1^fl/fl^Apoe⁻/⁻* mice (Fig. 3d), whereas they were no different in *Oasl1^fl/fl^ Lyz2*-cre⁺ *Apoe⁻/⁻* mice. We examined the role of OASL1 in ECs using a *Cdh5*-cre/ERT2 system (Fig. 3e, Supplementary Fig. 6) and *Oasl1^fl/fl^ Cdh5*-cre/ERT2⁺ *Apoe⁻/⁻* mice showed increased lesion formation compared with *Oasl1^+/+^ Cdh5*-cre/ERT2⁺ *Apoe⁻/⁻* mice (Fig. 3f). This indicates that the EC-specific defi-ciency of *Oasl1* promotes atherosclerosis in *Apoe⁻/⁻* mice, further demonstrating a potential endothelial effect.

## Endothelial deletion of *Oasl1* reduces the bioavailability of NO in athero-prone regions of the aorta

To confirm the contribution of endothelial *Oasl1* deletion-mediated endothelial dysfunction to atherogenesis, we analyzed the effects of *Oasl1* deficiency on the total EC cluster (cluster 4). A GSEA revealed that endothelial *Oasl1* deletion enriched the immune response, cell adhe-sion, and chemotaxis pathways in the EC cluster, which are findings consistent with scRNA-seq analyses of whole aortas (Figs. 4a, 2d).

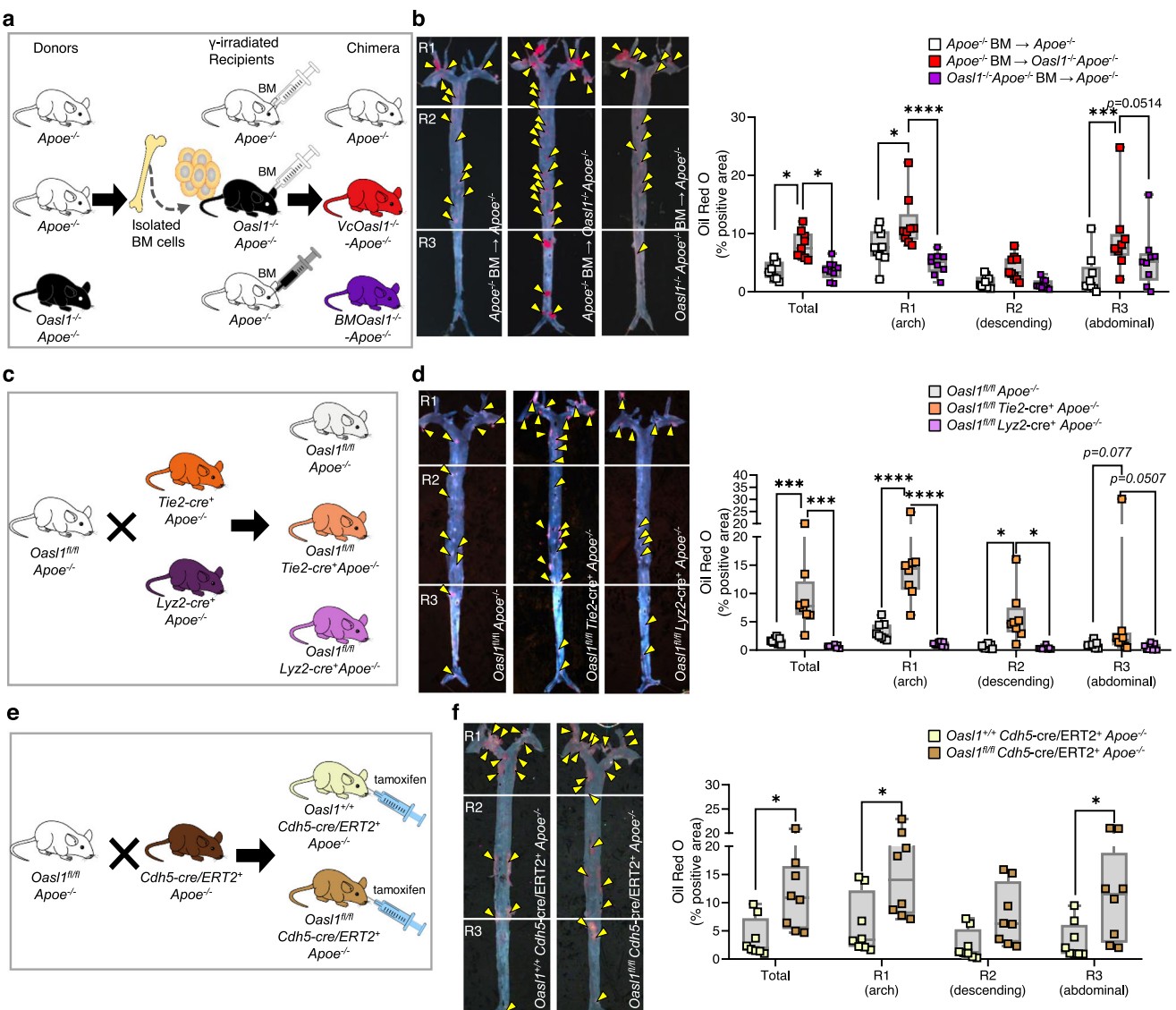

**Fig. 3 | Vascular EC-specific *Oasl1* deficiency exacerbates atherosclerotic plaque formation. a, b** *Apoe*[−/−] and *Oasl1*[−/−]*Apoe*[−/−] recipient mice, reconstituted with bone marrow (BM) from *Apoe*[−/−] mice or *Oasl1*[−/−]*Apoe*[−/−] mice, after γ-irradiation were fed a normal chow diet (NCD) for 28 weeks. **a** Experimental scheme. **b** Left: Oil red O-stained lesions in *en face* preparations of whole aortas ($p = 0.0145$; $p = 0.0148$, $n = 9$ per group). Yellow arrowhead: plaque in aorta. Right: Quantitation of oil red O-stained areas in the whole aorta and separate regions, including the aortic arch (R1; $p = 0.0212$; $p < 0.0001$), descending (R2; $p = 0.2079$; $p = 0.1467$) and abdominal (R3; $p = 0.0002$) regions. **c, d** *Oasl1*[fl/fl] *Apoe*[−/−], *Oasl1*[fl/fl] *Tie2*-cre[+] *Apoe*[−/−] and *Oasl1*[fl/fl] *Lyz2*-cre[+] *Apoe*[−/−] mice were fed a NCD for 28 weeks. (**c**) Experimental scheme. **d** Left: Oil red O-stained lesions in *en face* preparations of whole aortas ($p = 0.0009$; $p = 0.0002$, n = 8 per group). Yellow arrowhead: plaque in aorta. Right: Quantitation

of Oil red O-stained areas in the whole aorta and separate regions (R1: $p < 0.0001$; $p < 0.0001$, R2: $p = 0.0257$; $p = 0.0175$). **e, f** *Oasl1*[+/+] *Cdh5*-cre/ERT2[+] *Apoe*[−/−] and *Oasl1*[fl/fl] *Cdh5*-cre/ERT2[+] *Apoe*[−/−] mice were fed a western diet (WD) for 13 weeks. **e** Experimental scheme. **f** Left: Oil red O-stained lesions in *en face* preparations of whole aortas ($p = 0.0220$, $n = 8$ per group). Yellow arrowhead: plaque in aorta. Right: Quantitation of Oil red O-stained areas in the whole aorta and separate regions (R1: $p = 0.0106$, R2: $p = 0.1934$, R3: $p = 0.0231$). Data in left panels in **b**, **d**, and **f** are representative of each group. **b**, **d**, and **f**, Box plots are shown as median of each value and the IQR; whiskers indicate 1.5 times the IQR. Data are presented as means ± SEMs (*$p \leq 0.05$, **$p \leq 0.01$, ***$p \leq 0.001$, ****$p \leq 0.0001$; **b** and **d**, two-way ANOVA with Tukey's test; **f**, two-way ANOVA with Sidak's test for multiple comparisons). Source data are provided as a Source Data file.

Furthermore, endothelial *Oasl1* deletion profoundly elevated the adhesion and proinflammatory molecules, *Icam1*, *Icam2*, *Selplg*, *Il1b*, *Il6*, and *Tnfa* (Fig. 4b, Supplementary Fig. 7a). This was confirmed by measurements of *Icam1*, *Icam2*, *Selplg*, *Il1b*, *Il6*, *Ifng*, and *Tnfa* mRNA in activated MAECs isolated from the aortas of *Oasl1*[−/−] mice compared with *Oasl1*[+/+] mice (Fig. 4c, d). Moreover, monocyte adhesion (Fig. 4e) and transmigration (Fig. 4f) were enhanced in *Oasl1*[−/−] MAECs compared with controls, indicating that the loss of endothelial *Oasl1* promotes endothelial dysfunction, with the concomitant occurrence of a positive feedback loop of increased inflammation, leukocyte adhesion, and infiltration.

To characterize *Oasl1*-deficient ECs with respect to endothelial dysfunction, we defined subpopulations of ECs underlying

atherosclerosis using a clustering analysis. We identified eight distinct EC sub-clusters (Fig. 4g). The proportions of clusters 1 and 2, expressing *Cytl1*, *Klk10*, *Sfrp1*, and *Adh7*, which are canonical markers of prototypical ECs (Supplementary Fig. 7b), were reduced in the *Oasl1*[−/−]*Apoe*[−/−] group (Fig. 4h). The main cluster populated by the *Oasl1*[−/−]*Apoe*[−/−] mouse group (EC Sub-cluster 4) showed increased expression of genes associated with pulmonary hypertension and atherosclerosis, including *Selp*, *Edn1*, *Ctla2a*, *Ccl2*, *Mmrn1*, *Plvap*, and *Lgmn* (Fig. 4h, Supplementary Fig. 7b). This cluster also exhibited decreased NO biosynthetic activity (Fig. 4i). Collectively, these findings indicate that *Oasl1*-deficient ECs exhibit exaggerated endothelial dysfunction and inflammatory characteristics associated with a reduction in NO synthesis.

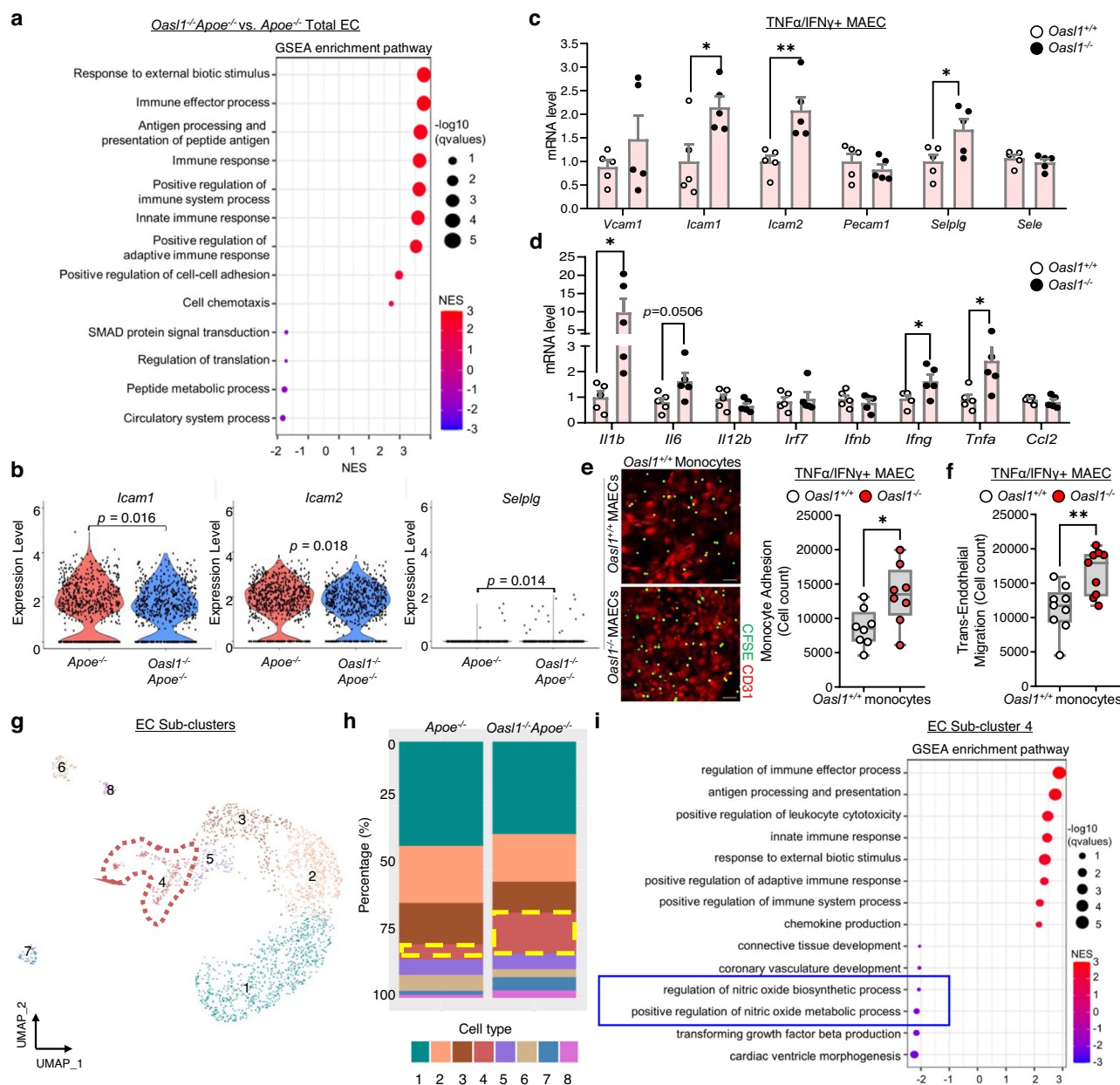

**Fig. 4 | *Oasl1* deficiency down-regulates NO synthesis and promotes endothelial inflammation. a** Dot plot showing Gene Set Enrichment Analysis (GSEA) according to normalized enrichment score (NES) for Gene Ontology (GO) terms of the total EC cluster from *Oasl1⁻/⁻Apoe⁻/⁻* mice versus that of *Apoe⁻/⁻* mice. **b** Violin plot showing a comparison of the adhesion molecules, *Icam1, Icam2,* and *Selplg,* in the aortic EC cluster of *Oasl1⁻/⁻Apoe⁻/⁻* versus *Apoe⁻/⁻* mice. **c, d** Quantitative PCR analysis of the adhesion molecules, *Vcam1* (*p* = 0.2965), *Icam1* (*p* = 0.0264), *Icam2* (*p* = 0.0080), *Pecam1* (*p* = 0.4082), *Selplg* (*p* = 0.0362), and *Sele* (*p* = 0.3979) (**c**), and the proinflammatory genes, *Il1b* (*p* = 0.0486), *Il6* (*p* = 0.0506), *Il12b* (*p* = 0.2400), *Irf7* (*p* = 0.7404), *Ifnb* (*p* = 0.4914), *Ifng* (*p* = 0.0477), *Tnfa* (*p* = 0.047) and *Ccl2* (*p* = 0.5093) (**d**) in MAECs isolated from *Oasl1⁺/⁺* and *Oasl1⁻/⁻* mice and stimulated with TNFα and IFNγ (*n* = 5 per group). **e** Assay of CFSE-labeled *Oasl1⁺/⁺* monocyte adhesion to TNFα- and IFNγ-stimulated MAECs isolated from *Oasl1⁺/⁺* and *Oasl1⁻/⁻*

mice (*p* = 0.0194, *n* = 8 per group). Scale bar, 50 μm. **f** Transendothelial migration assay of *Oasl1⁺/⁺* monocyte movement across activated *Oasl1⁺/⁺* and *Oasl1⁻/⁻* MAECs (*p* = 0.0044, *n* = 9 per group). **g–i** Among the total aortic cells, 2,121 aortic ECs were subclustered after performing a dimensionality reduction and were newly separated considering transcriptional similarity within the 2 experimental groups. **g** UMAP plot showing the distribution of eight defined endothelial sub-clusters. **h** Bar graph showing cell distribution according to genotype across the cell clusters. **i** Dot plot showing GSEA according to NES for GO terms of subclustered EC 4 derived from atherosclerotic aortas. Data in **e** are representative of each group. **e** and **f,** Box plots are shown as median of each value and the IQR; whiskers indicate 1.5 times the IQR. Data are presented as means ± SEMs (**p* ≤ 0.05, ***p* ≤ 0.01; **b–f,** two-sided unpaired Student's *t*-test). Source data are provided as a Source Data file.

As shown in the EC sub-clustering analysis of scRNA-seq data, one of the hallmarks of endothelial dysfunction is attenuation of eNOS/NO bioavailability accompanied by enhanced immune response[25]. Therefore, we determined whether decreased eNOS/NO bioavailability was reflected as a reduction in NO levels. NO levels in atherosclerotic plasma

(Fig. 5a) and aortic NO synthase enzymatic activity (Fig. 5b) in *Oasl1⁻/⁻Apoe⁻/⁻* mice were reduced compared with that of *Apoe⁻/⁻* controls, which supports our scRNA-seq results. A decrease in eNOS/NO bioavailability is known to contribute to vascular resistance and augmentation of BP[26,27]. *Oasl1* deficiency augmented mean blood pressure (BP) due to an increase in both

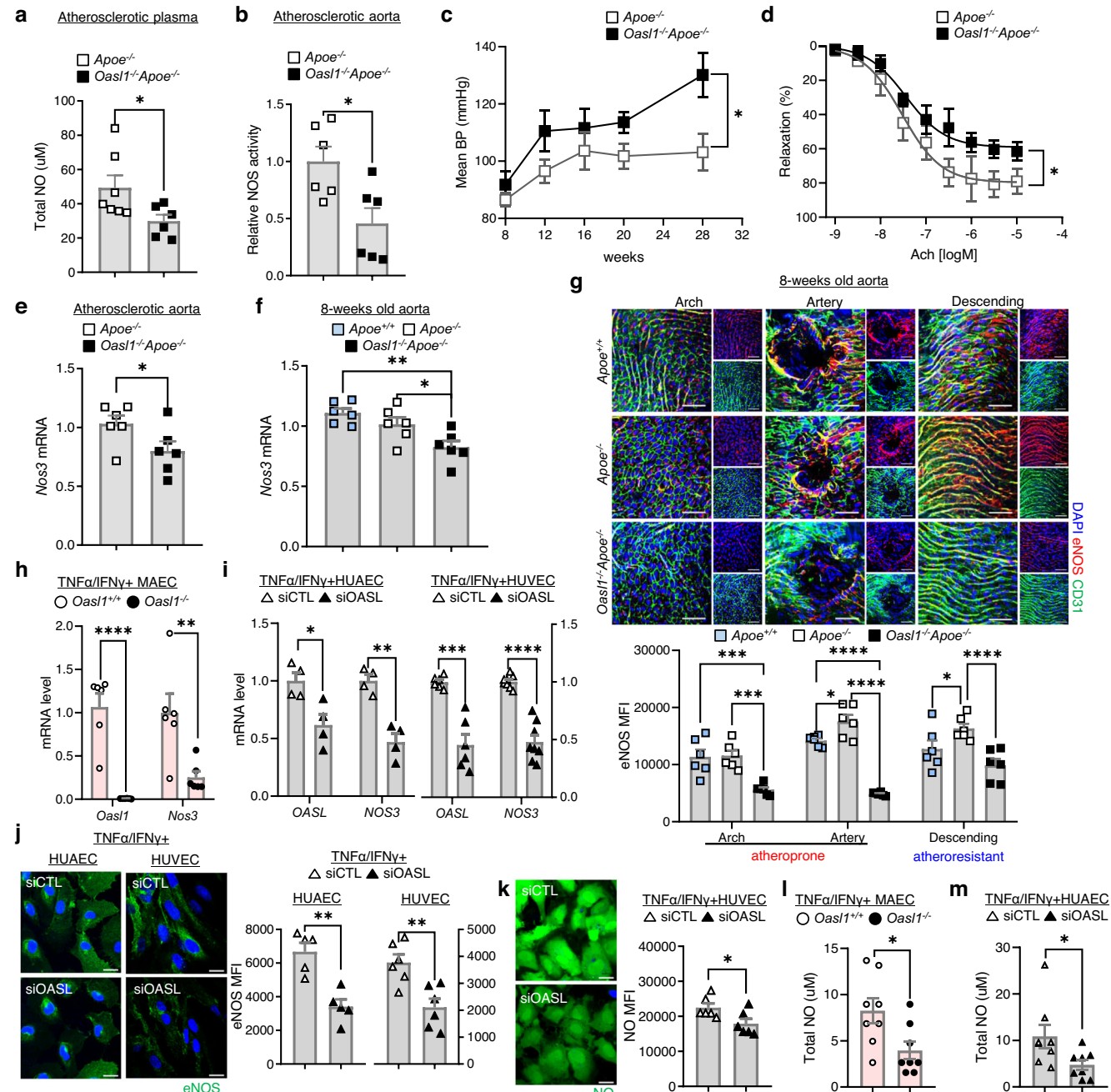

**Fig. 5 | *Oasl1* deficiency promotes endothelial dysfunction by reducing the levels of *Nos3* mRNA. a–e** *Apoe*⁻/⁻ and *Oasl1*⁻/⁻*Apoe*⁻/⁻ mice were fed a normal chow diet (NCD) for 28 weeks to allow atherosclerotic conditions to develop. Total nitric oxide (NO) levels in plasma (*p* = 0.0442, n = 7 for *Apoe*⁻/⁻, n = 6 for *Oasl1*⁻/⁻*Apoe*⁻/⁻) (**a**) and NOS enzymatic activity in atherosclerotic aortas (*p* = 0.0156, *n* = 6 per group) (**b**), as measured by the Griess method. **c** Mean blood pressure (BP) change measured using tail-cuff techniques (*p* = 0.0063, *n* = 7 per group). **d** The cumulative vascular relaxation response of the aortic rings to acetylcholine (Ach) as measured by myography (*p* = 0.0350, *n* = 4 per group). **e** Quantitative PCR analysis of *Nos3* mRNA in aorta tissues (*p* = 0.0318, *n* = 6 per group). **f–g** eNOS expression detected at multiple levels in the aortas of 8-week-old *Apoe*⁺/⁺, *Apoe*⁻/⁻ and *Oasl1*⁻/⁻*Apoe*⁻/⁻ mice. **f** *Nos3* mRNA as measured by quantitative PCR (*p* = 0.0029; *p* = 0.0413, n = 6 per group). **g** Top: *En face* IF staining of eNOS protein in CD31⁺ ECs (arch: *p* = 0.9877; *p* = 0.0004; *p* = 0.0002, branch: *p* = 0.0472; *p* < 0.0001; *p* < 0.0001, descending: *p* = 0.0306; *p* = 0.1017; *p* < 0.0001, *n* = 6 per group). Bottom: Quantitation of the corresponding MFI results. Scale bar, 50 μm. **h** Quantitative PCR analysis of *Oasl1* and *Nos3* mRNA levels in MAECs isolated from *Oasl1*⁺/⁺ and *Oasl1*⁻/⁻ mice and stimulated with TNFα and IFNγ (*Oasl1*: *p* < 0.0001, *Nos3*: *p* = 0.0087, *n* = 6 per group).

**i–k** HUAECs and HUVECs were transfected with non-targeting siRNA (siCTL) or OASL-targeting siRNA (siOASL) followed by stimulation with TNF-α and IFN-γ. **i** *OASL* and *NOS3* mRNA levels as measured by quantitative PCR (n = 4 for *OASL*: *p* = 0.0192, *NOS3*: *p* = 0.0011 in HUAEC; *n* = 6 for *OASL*: *p* = 0.0002, *n* = 8 for *NOS3*: *p* < 0.0001 in HUVEC). **j** Left: Immunocytochemical detection of eNOS. Scale bar, 20 μm. Right: Quantitation of corresponding MFI values (*p* = 0.0014, *n* = 4 for HUAEC; *p* = 0.0014, *n* = 6 for HUVEC). **k** Left: Immunocytochemical detection of NO. Scale bar, 20 μm. Right: Quantitation of corresponding MFI values (*p* = 0.0343, n = 6 per group). **l, m** Total NO was determined by ELISA using the Griess reaction in the supernatant of MAECs isolated from *Oasl1*⁺/⁺ and *Oasl1*⁻/⁻ mice (*p* = 0.0214, *n* = 8 per group) (**l**) and HUAECs transfected with siCTL or siOASL (*p* = 0.0382, *n* = 8 per group) (**m**). Data in **g**, **j**, and **k** are representative of at least 5 independent experiments. Data are presented as means ± SEMs (**p* ≤ 0.05, ***p* ≤ 0.01, ****p* ≤ 0.001, *****p* ≤ 0.0001; **a**, **b**, **e**, and **h–m**, two-sided unpaired Student's *t*-test; **c**, two-way ANOVA with Sidak's test; **f**, one-way ANOVA with Tukey's test; **g**, two-way ANOVA with Tukey's test for multiple comparisons; Nonlinear fit data in **d** was calculated using a Sigmoidal dose-response). Source data are provided as a Source Data file.

systolic and diastolic BP in *Apoe*[−/−] mice following atherosclerotic plaque formation compared with controls (Fig. 5c, Supplementary Fig. 8a), whereas NO supplementation significantly reduced BP in both groups (Supplementary Fig. 8b). There was no difference in heart rate or function, including ejection fraction, fractional shortening, or cardiac output between *Oasl1*[−/−]*Apoe*[−/−] and *Apoe*[−/−] mice (Supplementary Fig. 8c–f). This indicates that the augmentation of BP in *Oasl1* deficiency was dependent on vascular NO level regardless of heart function. The result of increased BP was repeated in *Oasl1*[fl/fl]Cdh5-cre/ERT2[+]*Apoe*[−/−] mice compared with the corresponding controls, suggesting an endothelial-specific *Oasl1* effect on hypertension following plaque formation (Supplementary Fig. 9). When the *Oasl1*-deletion effect was assessed without hyperlipidemic conditions (Supplementary Fig. 10a), blood velocity and WSS were significantly lower at the lesser curvature of the aortic arch in both *Oasl1*[−/−] and *Nos3*[−/−] mice compared with the *Oasl1*[+/+] controls (Supplementary Fig. 10b, c). Whereas there were no differences in BP between *Oasl1*[−/−] and *Oasl1*[+/+] mice, in which the value was smaller compared with that in *Nos3*[−/−] mice (Supplementary Fig. 10d). These results suggest that *Nos3*[−/−] mice phenocopied *Oasl1*[−/−] mice with respect to endothelial dysfunction from normal conditions and hyperlipidemia-activated endothelial dysfunction promoted an increase in BP that result from a reduction of eNOS/NO bioavailability in the vasculature of *Oasl1*[−/−]*Apoe*[−/−] mice. Moreover, aortic vascular relaxation was significantly reduced dependent on acetylcholine, whereas constriction was increased following a decrease in vascular fractional area change in *Oasl1*[−/−]*Apoe*[−/−] mice compared with controls (Fig. 5d, Supplementary Fig. 11). Treatment with the soluble guanylyl cyclase inhibitor, ODQ, reduced the percentage of relaxation in both groups by same amount (Supplementary Fig. 11c), indicating that *Oasl1* deficiency results in vascular functional impairment primarily derived from endothelial malfunction, rather than a defect in smooth muscle.

As a first step toward developing a mechanistic understanding of the preceding phenomena, we determined whether these changes were attributable to changes in *Nos3* expression. We found that *Nos3* mRNA levels in atherosclerotic aortic tissues of *Oasl1*[−/−]*Apoe*[−/−] mice were reduced compared with those of *Apoe*[−/−] controls (Fig. 5e), without affecting the expression of other *Nos* genes (Supplementary Fig. 12). This suggests that a reduction in eNOS-mediated NO synthesis caused by the absence of *Oasl1* contributes to an increase in susceptibility of arteries to atherosclerosis following endothelial dysfunction and systemic vascular resistance. Even at the initial stages following the induction of an atherosclerotic environment, *Nos3* mRNA levels were decreased in the aortic arches of *Oasl1*[−/−]*Apoe*[−/−] mice compared with the *Apoe*[−/−] and *Apoe*[+/+] controls (Fig. 5f). IF staining of *en face* aortic preparations revealed that eNOS expression was increased in the endothelium of the arteries or descending region of *Apoe*[−/−] mouse aortas compared with *Apoe*[+/+] controls, which showed little site- or geometry-dependent variation in eNOS expression (Fig. 5g). In contrast, vasoprotective eNOS expression was decreased in both athero-prone and athero-resistant regions of the aortas of *Oasl1*[−/−]*Apoe*[−/−] mice compared with *Apoe*[−/−] mice (Fig. 5g). NO and cyclic GMP (cGMP) production was also reduced in the aortas of *Oasl1*[−/−]*Apoe*[−/−] mice compared with *Apoe*[−/−] mice (Supplementary Fig. 13a, b), suggesting that endothelial *Oasl1* deficiency triggers athero-prone features in aortas through down-regulation of eNOS expression and NO production.

Examination of the cell-specific process of OASL1-mediated maintenance of *Nos3* revealed that *Oasl1* deficiency significantly reduced *Nos3* mRNA expression in MAECs compared with *Oasl1*[+/+] controls following stimulation (Fig. 5h). Similar to OASL1 in MAECs, *OASL* knockdown (KD) in both HUAECs and HUVECs, using small-interfering RNA targeting human OASL (siOASL), strongly reduced *NOS3* mRNA levels compared with that in siCTL-transfected controls under activated conditions (Fig. 5i). *OASL* knockdown in HUAECs and HUVECs also decreased immunostaining for eNOS (Fig. 5j), and endothelial NO (Fig. 5k) compared with controls. Moreover, total NO level was lessened in the supernatant of *Oasl1*-deficient MAECs (Fig. 5l) and *OASL*-KD HUAECs (Fig. 5m) compared to each of controls indicating that, as was the case for murine Oasl1, reduction of human *OASL* decreases *NOS3* mRNA levels in athero-prone ECs, lowering NO bioavailability. Concurrently, we validated that pretreatment of cells with L-NAME, an eNOS inhibitor, markedly attenuated NOS activity (Supplementary Fig. 13c, d) and reduced secreted NO (Supplementary Fig. 13e, f) and intracellular cGMP level (Supplementary Fig. 13g, h) as was in the absence of *Oasl1*, implying eNOS-dependent regulatory mechanism of endothelial OASL1.

## PI3K/Akt/OASL signaling maintains *NOS3* mRNA stability by down-regulating miRNAs

*NOS3*-mediated NO expression is regulated at multiple levels by a number of mechanical or biochemical stimuli to which the vasculature is exposed during disease conditions. To determine the upstream regulatory signaling associated with OASL-mediated *NOS3* expression, we examined representative inflammation-associated pathways that are known to be involved in *NOS3* expression or activation. Notably, *NOS3* mRNA levels in TNFα/IFNγ-activated HUVECs were markedly decreased by treatment with the PI3K/Akt inhibitor LY294002, but not by rapamycin, and were increased by the selective MEK inhibitor, U0126 (Fig. 6a). *OASL* mRNA levels in activated HUVECs were also decreased following treatment with Akt inhibitors but were unaffected by other signaling-pathway inhibitors (Supplementary Fig. 14a). This suggests that *NOS3 mRNA* maintenance under inflammatory conditions is dependent upon Akt activation and is controlled by the expression of *OASL*. *NOS3* mRNA levels were decreased in LY294002-treated HUVECs in a time-dependent manner (Fig. 6b), and siRNA-mediated *OASL* knockdown (Supplementary Fig. 14b) additionally inhibited *NOS3* mRNA expression (Fig. 6b). Consistent with these observations, NO production was significantly decreased in LY294002-treated HUVECs under stimulated conditions, and further down-regulated in *OASL*-KD HUVECs compared with the controls (Fig. 6c). Moreover, an assessment of the signaling pathways affected by *OASL* knockdown-dependent decreases in eNOS expression showed that Erk1/2 and NF-κB activation, known to reduce eNOS, were enhanced in *OASL*-KD HUVECs compared with the controls (Fig. 6d). Further inhibition of Akt phosphorylation increased the phosphorylation of Erk1/2 and p65 in activated HUVECs (Fig. 6d). Collectively, these data indicate that Akt activation mediates the preservation of *NOS3* mRNA and facilitates NO production via OASL expression, whereas decreased eNOS levels following a reduction in OASL contribute to augmentation of inflammatory activation.

Reduced eNOS in ECs exerts a variety of physiological responses including an increase in inflammatory signaling during pathological conditions[25]. Whether the reduced OASL-mediated decrease in *NOS3* affects inflammatory activation was further confirmed by *NOS3*-KD in HUAECs and MAECs. *OASL/NOS3*-KD HUAECs and *Nos3*-KD/*Oasl1*[−/−] ECs increased Erk1/2 and NF-κB activation with upregulation of adhesion molecules including *ICAM1* and *SELE* compared with *OASL*-KD or *Oasl1*[−/−] and the controls (Supplementary Fig. 15). Bulk RNA-seq of HUVECs transfected with siCTL or siOASL confirmed that *OASL* knockdown exacerbates various inflammation-associated processes following a reduction in eNOS levels. Examination of differentially expressed genes (DEGs) (Fig. 6e) coupled with Gene Ontology (GO) analyses (Fig. 6f) revealed that a deletion of endothelial *OASL* altered the regulation of biological processes, including NO synthesis, BP, adhesion, chemotaxis and activation of MAPK or NF-κB pathways. Further examination of the potential roles of the above activated

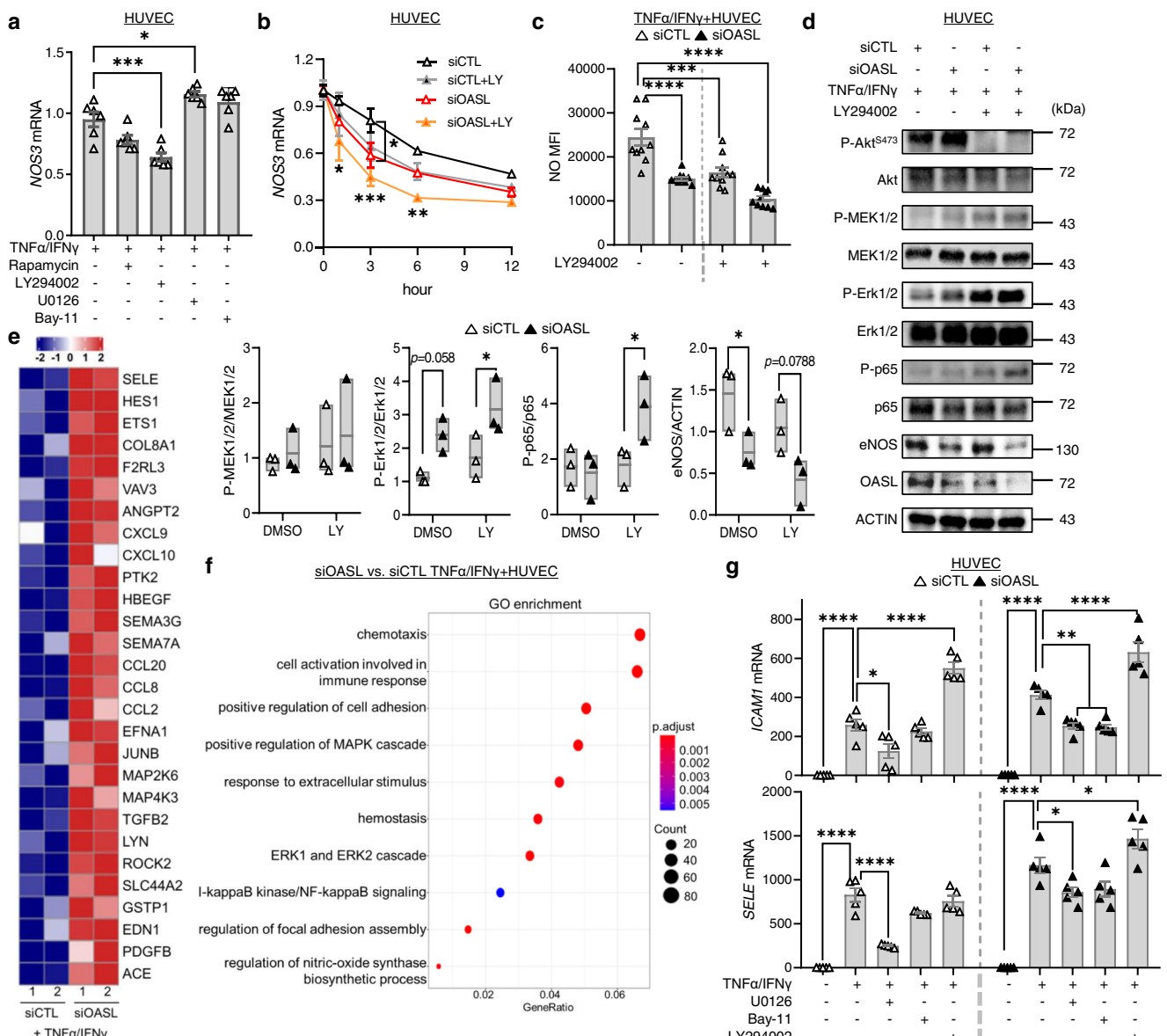

**Fig. 6 | Endothelial *OASL* knockdown reduces *NOS3* expression via the PI3K/Akt-dependent pathway. a, b** Quantitative RT-PCR analysis of *NOS3* mRNA in HUVECs pretreated with the mTOR inhibitor, rapamycin, PI3K inhibitor, LY294002, ERK 1/2 inhibitor, U0126, or the NF-κB inhibitor, Bay-11 (20 μmol/L for each) for 1 h before stimulation with TNF-α and IFN-γ for 3 h (LY: *p* = 0.0003, U: *p* = 0.0206, *n* = 6 per group) (**a**), or pretreated with the PI3K inhibitor, LY294002 (LY), transfected with siCTL or siOASL, and monitored over time (siCTL vs. siOASL 3 h: *p* = 0.0472; siCTL vs. siOASL+LY 1 h: *p* = 0.0146, 3 h: *p* = 0.0003, 6 h: *p* = 0.0036, *n* = 4 per group) (**b**). **c, d** siCTL- or siOASL-transfected HUVECs were pretreated with the PI3K inhibitor, LY294002 (20 μmol/L), and stimulated with TNF-α and IFN-γ. **c** Quantitation of MFI values for DAF-DM diacetate staining of nitric oxide (NO; *p* < 0.0001; *p* = 0.0004; *p* < 0.0001, n = 10 per group). **d** Right: Western blot analysis of Akt, Erk1/2, NF-κB (p65), and eNOS expression. Left: Quantitation of band density (P-ERK1/2: *p* = 0.0355, P-p65: *p* = 0.0407, eNOS: *p* = 0.0478). **e, f** Bulk RNA-seq analysis performed on siCTL- or siOASL-transfected HUVECs stimulated with TNFα and IFNγ

(n = 2 per group). **e** Heatmap showing different gene expression patterns between the two groups. **f** Gene ontology (GO) enrichment analysis of dysregulated biological processes in *OASL*-KD HUVECs. **g** Quantitative PCR analysis of the adhesion molecules, *ICAM1* (siCTL DMSO: *p* < 0.0001; U: *p* = 0.0330; LY: *p* < 0.0001, siOASL DMSO: *p* < 0.0001; U: *p* = 0.0055; Bay: *p* = 0.0034; LY: *p* < 0.0001) and *SELE* (siCTL DMSO: *p* < 0.0001; U: *p* < 0.0001, siOASL DMSO: *p* < 0.0001; U: *p* = 0.0399; LY: *p* = 0.0498), in siCTL- or siOASL-transfected HUVECs pretreated with the MEK1/2 inhibitor, U0126, the NF-κB inhibitor, Bay-11, or PI3K inhibitor, LY294002 (20 μmol/L each), for 1 h before stimulation with TNF-α and IFN-γ (*n* = 5 per group). Data in **d** are representative of 3 independent experiments. **d**, Box plots are shown as median of each value and the IQR. Data are presented as the means ± SEMs (**p* ≤ 0.05, ***p* ≤ 0.01, ****p* ≤ 0.001, *****p* ≤ 0.0001; **a**, one-way ANOVA with Bonferroni's test; **b**, **c** and **g**, two-way ANOVA with Tukey's test; **d** two-way ANOVA with Sidak's test for multiple comparisons; **f**, one-sided Fisher's exact test). Source data are provided as a Source Data file.

---

signaling pathways in upregulating the expression of EC-derived adhesive molecules in *OASL*-KD HUVECs verified that the MEK inhibitor, U0126, and NF-κB inhibitor, Bay-11, significantly decreased expression of the adhesion molecules, *ICAM1* and *SELE* (Fig. 6g). In contrast, activation of PI3K/Akt signaling was reported to restrain proinflammatory responses by inhibiting activation of the MAPK pathways[28–30]. Similarly, blocking Akt phosphorylation at Ser473 with

the PI3K inhibitor, LY294002, significantly increased the expression of *ICAM1* and *SELE* in activated HUVECs, similar to the effect observed following *OASL* knockdown (Fig. 6g). However, the type I IFN-related pathway, which is a proinflammatory mechanism affected by OASL and OASL1[18,20,31,32], did not significantly change with *Oasl1* deletion in HUAECs and MAECs (Supplementary Fig. 16, 17), demonstrating that proinflammatory and adhesion events are aggravated in *OASL*-KD

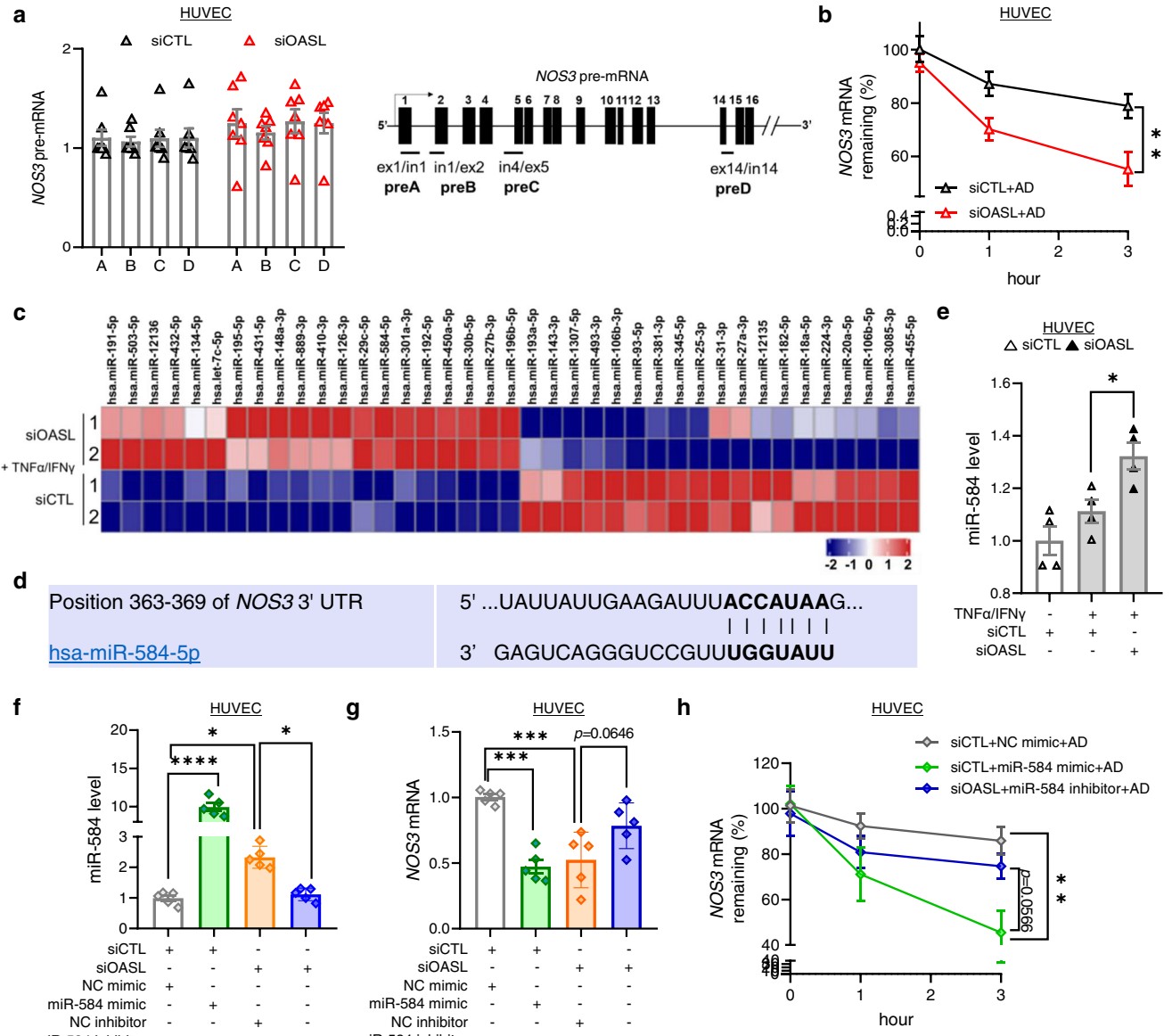

**Fig. 7 | Endothelial OASL contributes to *NOS3* mRNA stability by regulating *NOS3*-targeting miRNAs. a** *NOS3* pre-mRNA levels measured by detecting primary transcripts isolated from siCTL- or siOASL-transfected HUVECs (*n* = 7 per group). **b** qRT-PCR–based analysis of *NOS3* mRNA stability in siCTL- or siOASL-transfected HUVECs (3 h: *p* = 0.0035, *n* = 7 for siCTL+AD; *n* = 8 for siOASL+AD). *NOS3* mRNA was measured over time in the presence of actinomycin D (AD), which was added to block transcription. **c** microRNA (miRNA)-seq analysis performed on HUVECs transfected with siCTL or siOASL and stimulated with TNF-α and IFN-γ (*n* = 2 per group), depicted as a heatmap showing differences in miRNA expression patterns between the two groups. **d** The miR-584-5p binding site within the *NOS3* 3′-UTR region as predicted by TargetScan. **e** qRT-PCR analysis of miR-584 expression in

siCTL- or siOASL-transfected HUVECs stimulated with TNF-α and IFN-γ (*p* = 0.0399, *n* = 4 per group). **f**–**h** qRT-PCR analysis of miR-584 levels (*p* < 0.0001; *p* = 0.0185; *p* = 0.0348, *n* = 4 per group) (**f**), *NOS3* mRNA (*p* = 0.0002; *p* = 0.0007, *n* = 4 per group) (**g**) and *NOS3* mRNA stability in the presence of AD (3 h: *p* = 0.0057, *n* = 5 for NC mimic, miR-584 inhibitor; *n* = 7 for miR-584 mimic) (**h**) in siCTL- or siOASL-transfected HUVECs, post-transfected with 80 nM of negative control (NC) mimic, miR-584 mimic, NC inhibitor, or miR-584 inhibitor. Data are presented as means ± SEMs (*\**p* ≤ 0.05, \*\**p* ≤ 0.01, \*\*\**p* ≤ 0.001, \*\*\*\**p* ≤ 0.0001; **a**, **b** and **h**, two-way ANOVA with Tukey's test; **e**–**g**, one-way ANOVA with Tukey's test for multiple comparisons). Source data are provided as a Source Data file.

HUVECs primarily through attenuated eNOS expression and enhanced MAPK and NF-κB signaling.

Messenger RNA levels are regulated under various cellular signaling conditions through multiple processes that act at both the transcriptional and post-transcriptional levels[9,33,34]. *NOS3* expression was not increased in plaque-containing atheroma tissues or TNFα/INFγ-treated HUVECs, despite the induction of OASL expression, which suggested that augmented OASL does not regulate *NOS3* expression through transcriptional activation. We compared *NOS3* levels in unstimulated ECs to determine whether decreased OASL1/OASL had an effect on *NOS3* expression regardless of their inducible expression. *NOS3*

expression decreased by 50% in *Oasl1*[−/−] MAECs (Supplementary Fig. 18a) and this was repeated in *OASL*-KD HUVECs relative to controls without stimulation (*P = 0.0001*; Supplementary Fig. 18b). eNOS protein level was also decreased (Supplementary Fig. 18c) and this was followed by proinflammatory activation in *OASL*-KD HUVECs in a highly correlated manner under stimulated conditions (Supplementary Fig. 18d, e), This suggests that the presence of basal OASL determines *NOS3* mRNA expression and regulates endothelial homeostasis in athero-prone areas.

Consistent with these findings, the levels of *NOS3* precursor mRNA (pre-mRNA)[35] was not different between siCTL- and siOASL-transfected HUVECs (Fig. 7a), indicating that *OASL* knockdown does

not affect *NOS3* transcription. Therefore, using mRNA-decay assays containing actinomycin D (to inhibit *NOS3* transcription), we determined whether OASL regulates *NOS3* mRNA stability. The results indicated that *NOS3* mRNA stability was considerably decreased in siOASL-treated HUVECs compared with siCTL-treated controls (Fig. 7b, Supplementary Fig. 14c), indicating that *OASL* knockdown affects *NOS3* mRNA levels through a post-transcriptional mechanism. Because miRNAs are known to regulate their mRNA targets through degradation or translational repression[36], we identified endothelial miRNAs affected by OASL. A comparison of the sequences of miRNAs in HUVECs transfected with siCTL or siOASL revealed differentially expressed miRNAs (Fig. 7c, Supplementary Fig. 18f). We focused on *OASL*-KD–dependent up-regulated miRNAs that are capable of reducing *NOS3* mRNA. An in silico analysis using TargetScan Human v.7.2 (targetscan.org)[37] predicted that, among these miRNAs, miR-584-5p targets a 3′-UTR region of *NOS3* mRNA (Fig. 7d), suggesting that this may be a promising target for OASL. We confirmed that miR-584-5p was significantly increased in *OASL*-KD HUVECs compared with controls (Fig. 7e). A subsequent analysis revealed that miR-584 significantly reduced *NOS3* mRNA levels, mimicking the effect of *OASL* knockdown in HUVEC and HUAEC, whereas miR-584 inhibitors rescued *NOS3* expression (Fig. 7f, g, Supplementary Fig. 19a, b). Importantly, miR-584 treatment recapitulated the decrease in *NOS3* mRNA stability observed in the RNA decay assays, whereas treatment of *OASL*-KD HUVECs and HUAECs with a miR-584 inhibitor restored its stability (Fig. 7h, Supplementary Fig. 19c), demonstrating a possible role of miR-584 in regulating the stability of *NOS3* mRNA. Collectively, these findings suggest that the maintenance of *NOS3* mRNA is dependent on the continued presence of OASL in ECs, which acts through repression of the potential target miR-584 to prevent *NOS3* mRNA degradation.

### Endothelial *Oasl1* expression attenuates endothelial dysfunction and ameliorates atherogenesis in athero-prone aortic regions

Finally, to verify the positive in vivo effect of mouse endothelial OASL1 on vascular homeostasis and atherogenesis, we performed a BM transplantation (BMT), which mimics the effect of rescuing OASL1 only in vascular cells (especially ECs in our case). Because the presence of endothelial OASL1 by itself may affect *NOS3* levels, our strategy involved the transplantation of BM from *Oasl1*[−/−]*Apoe*[−/−] mice into *Oasl1*[−/−]*Apoe*[−/−] and *Apoe*[−/−] mice (Fig. 8a). The vascular cell-specific presence of *Oasl1* in *Apoe*[−/−] mice (Vc*Oasl1*[+/+]-*Apoe*[−/−] mice) ameliorated atherosclerotic plaque development in whole aortas compared with that in *Oasl1*[−/−]*Apoe*[−/−] mice (Fig. 8b). In particular, lesion formation in Vc*Oasl1*[+/+]-*Apoe*[−/−] mice was reduced in athero-prone regions including the aortic arch and abdominal aorta (Fig. 8b). Moreover, blood flow in Vc*Oasl1*[+/+]-*Apoe*[−/−] mice was dramatically higher in athero-prone regions, such as the lesser curvature of the aortic arch and abdominal branching point, compared with that in *Oasl1*[−/−]*Apoe*[−/−] mice (Fig. 8c). WSS in Vc*Oasl1*[+/+]-*Apoe*[−/−] mice was also significantly greater in athero-prone regions including the lesser curvature of aortic arch and abdominal branching point compared with that in *Oasl1*[−/−]*Apoe*[−/−] mice (Fig. 8c). We further discovered that the population of infiltrated CD45[+] leukocytes in aortas, including total macrophages and neutrophils, was decreased in Vc*Oasl1*[+/+]-*Apoe*[−/−] mice compared with whole-body *Oasl1*-deficient mice (Fig. 8d). In addition, the expression of *Oasl1* in ECs ameliorated adhesion (Fig. 8e) and transmigration (Fig. 8f) of *Oasl1*[−/−] monocytes. Taken together, these results demonstrate that the presence of OASL1, especially when restricted to aortic vascular cells, exerts a protective effect by alleviating plaque formation through reduced EC dysfunction, as evidenced by the rescue of WSS and blood velocity and decreased leukocyte infiltration.

## Discussion

eNOS and NO deficiencies in the endothelium result in chronic cardiovascular diseases and affect acute inflammatory vascular complications[8,38]. In this study, we demonstrated that athero-susceptible region-specific expression of endothelial OASL1 ameliorated atherosclerosis, likely by maintaining eNOS/NO bioavailability. Whole-body, as well as EC-specific Oasl1 deficiency, triggered excessive EC activation, leukocyte infiltration, and vascular inflammation originating from impaired eNOS-mediated cardiovascular homeostasis. Similar to murine *Oasl1*, knockdown of human *OASL* in human ECs reduced *NOS3* expression and increased inflammatory signaling via the PI3K/Akt-dependent pathway. Among candidate miRNA regulators, miR-584 was increased following *OASL* knockdown, and miR-584 inhibitors rescued *NOS3* mRNA stability. Finally, vascular-specific expression of OASL1 attenuated endothelial dysfunction and atherogenesis, highlighting the potential of OASL1 as a protector of eNOS.

Geometric features of the vasculature elicit distinct blood flow pattern-mediated endothelial WSS at regions of arterial bends and branch points known as athero-prone sites[39]. Although we need to further demonstrate the additive effect of hemodynamics and inflammatory stimuli, both athero-prone shear response and proinflammatory activators triggered the unique OASL1 expression pattern[4,40,41]. The reciprocal actions of dysfunctional EC formation and alterations in WSS reflecting low or oscillatory flow resulted in vascular remodeling and plaque formation[39,42]. Therefore, in human atherosclerosis, measurements of endothelial function are utilized as early disease risk indicators to provide prognostic information that is useful for primary prevention and treatment of coronary artery disease[43,44]. When we checked site-dependent alterations in vascular function, the absence of *Oasl1* reduced the percentage of the vascular fractional area change, followed by a decrease in blood velocity and endothelial WSS from the early phase of atherosclerosis. These phenomena mediated a more athero-prone environment that attracted leukocyte infiltration and promoted lesion formation and vasoconstriction with an increased BP[44]. This set of new kinetics data could unravelling a specific role of OASL1 in EC-dependent regulation of atherogenesis. Moreover, *Oasl1*-deficient ECs were characterized by terms including antigen presentation via MHC class I, cell-cell adhesion, and proinflammatory signaling, which are typical features of dysfunctional ECs exposed to athero-prone flow[40]. These observations suggest a possible role of OASL1 as a marker for EC dysfunction and highlight the importance of maintaining basal OASL1 expression, especially in athero-susceptible areas.

Although OASL1 was markedly increased in aortic plaques, which contain both activated ECs and infiltrating immune cells, the minimal atherosclerotic environment formed in young *Apoe*[−/−] mice sufficiently triggered the expression of OASL1 in the endothelium, suggesting that vascular ECs are the fundamental *Oasl1*-expressing aortic cell type. Our observations further revealed that the absence of whole-body *Oasl1* during chronic inflammatory conditions exacerbated plaque formation, unlike previously reported acute immune diseases[18,20,21], and that the effect in myeloid cell type was insignificant. Ultimately, both vascular cell- and EC-specific *Oasl1* deficiencies accelerated atherogenesis, consistent with the results obtained from whole-body *Oasl1* deficiency. Notably, the effects of OASL1 in ECs may surpass the impact of other cell types that participate in lesion development although OASL1 expression and function may be inducible in immune cells. The endothelial expression pattern of murine OASL1 in disease conditions was similar to that of human OASL; thus, the human ortholog, OASL, may be a valuable biomarker for assessing the prognosis of patients with athero-prone conditions. Moreover, a genome-wide association study validated the significant association of variants near OASL with multiple cardiovascular-related traits, such as low-density lipoprotein and C-reactive protein[45]. Even though further insight into the phenotypes affected by OASL variants is needed, identifying and modulating OASL expression may represent a therapeutic strategy for atherosclerosis treatment.

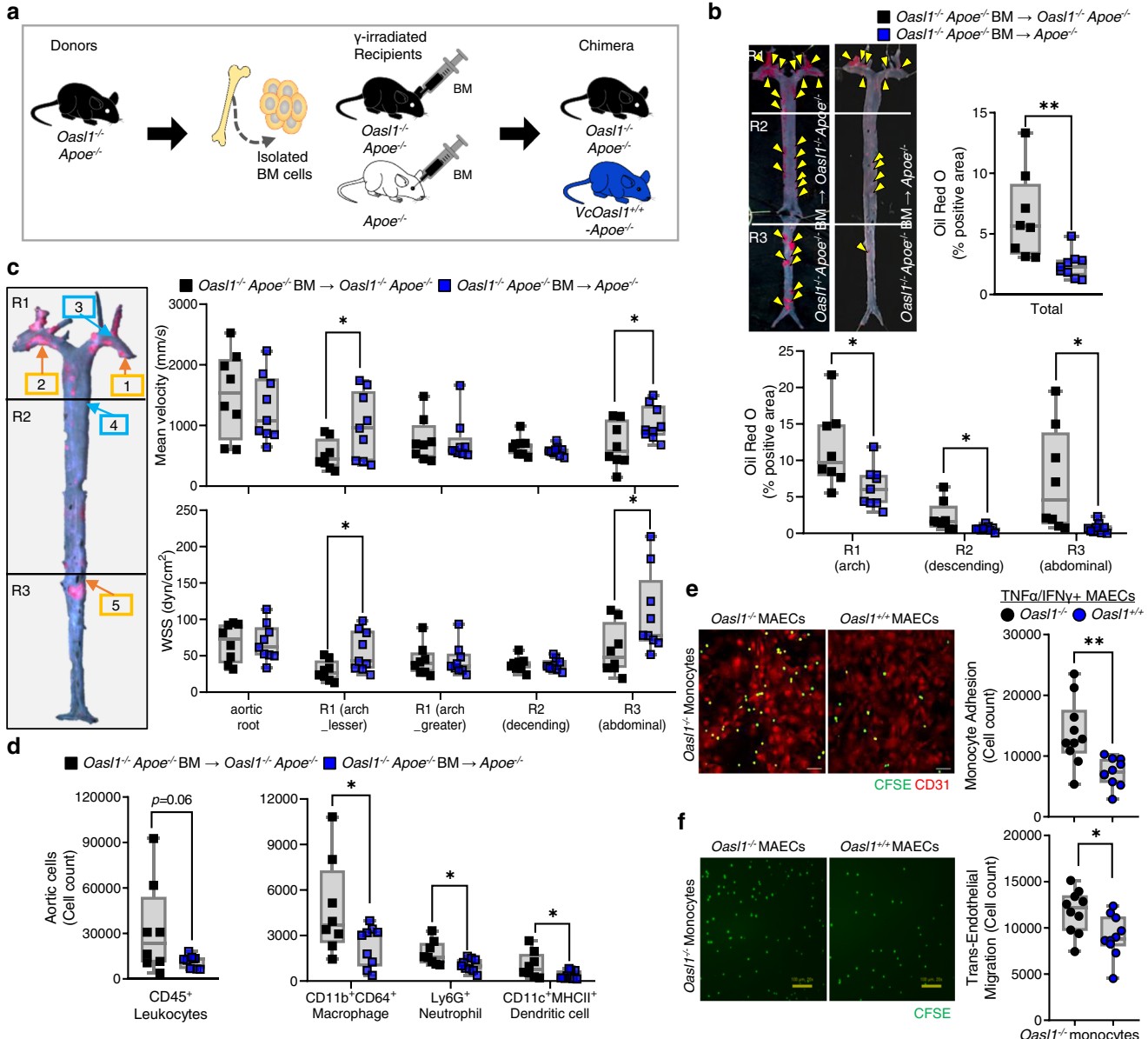

**Fig. 8 | Endothelial OASL1 attenuates leukocyte recruitment and lesion formation. a–d** Rescue effects of *Oasl1* in vascular cell types were assessed by reconstituting γ-irradiated *Oasl1⁻/⁻Apoe⁻/⁻* and *Apoe⁻/⁻* mice with bone marrow (BM) from *Oasl1⁻/⁻Apoe⁻/⁻* mice followed by a normal chow diet (NCD) for 28 weeks (*n* = 8 for *Oasl1⁻/⁻Apoe⁻/⁻*; n = 9 for Vc*Oasl1⁺/⁺-Apoe⁻/⁻*). **a** Experimental scheme. **b** Left panel: Oil red O-stained lesions in whole aortas. Yellow arrowhead: plaque in aorta. Right panel: Quantification of stained *en face* area. Bottom panel: Separate quantitation of Oil red O⁺ areas in aortic arch (R1), descending (R2) and abdominal (R3) regions (Total: *p* = 0.0058, R1: *p* = 0.0203, R2: *p* = 0.0310, R3: *p* = 0.0159). **c** Blood flow velocity (top) and endothelial wall shear stress (WSS; bottom) at the athero-resistant greater curvature, descending and athero-prone lesser curvature, or abdominal branching point of arteries were assessed in age-matched 20-week-old mice prior to identification of plaque formation by serial echocardiography

(velocity R1 (lesser): *p* = 0.0421, R3: *p* = 0.0383; WSS R1 (lesser): *p* = 0.0283, R3: *p* = 0.0478). **d** Flow cytometry analysis of single cells including total leukocytes, macrophages, neutrophils, and DCs, isolated from atherosclerotic aortas (Macrophage: *p* = 0.0499, Neutrophil: *p* = 0.0204, DC: *p* = 0.0427). **e** Assay of CFSE-labeled *Oasl1⁻/⁻* monocyte adhesion on TNFα- and IFNγ-stimulated MAECs isolated from *Oasl1⁻/⁻* and *Oasl1⁺/⁺* mice (*p* = 0.0028, *n* = 10 per group). Scale bar, 50 μm. **f** Transendothelial migration assay of *Oasl1⁻/⁻* monocyte movement across TNFα- and IFNγ-stimulated *Oasl1⁻/⁻* and *Oasl1⁺/⁺* MAECs (*p* = 0.0198, *n* = 10 per group). Scale bar, 100 μm. Data in **b**, **e**, and **f** are representative of each group. **b–f**, Box plots are shown as median of each value and the IQR; whiskers indicate 1.5 times the IQR. Data are presented as means ± SEMs (**p* ≤ 0.05, ***p* ≤ 0.01, ****p* ≤ 0.001; **b–f**, two-sided unpaired Student's *t*-test). Source data are provided as a Source Data file.

OASL1 and OASL appear to mediate atheroprotective responses through various mechanisms. Both are known to modulate type I IFN signaling[18,20,31,32], however, we could not see a difference in this pathway under chronic inflammatory conditions. Thus, we needed to identify another target mechanism in the ECs during atherosclerotic conditions. Our discovery of EC sub-clusters revealed a deeper mechanistic association between endothelial OASL1 and eNOS/NO bioavailability. We provided supporting evidence that *Nos3* deficiency

augmented vascular resistance and reduced WSS as observed with *Oasl1* deficiency. This resulted in leukocyte infiltration and the progression of an immune response, which are processes representative of endothelial proinflammatory activation following decreased eNOS/NO expression[4,25]. Similarly, *OASL*-KD HUVECs promoted immune cell adhesion and enhanced inflammatory signaling including the ERK1/ERK2 cascade and the NF-κB signaling pathway, both of which are known to decrease eNOS expression and promote atherosclerosis

progression[6]. Moreover, *NOS3*-knockdown in *Oasl1*-deficient and *OASL*-KD ECs exhibited an additive effect on inflammatory activation resulting from the anti-inflammatory and global protective effects of *NOS3*-eNOS[25,46]. Interestingly, both *OASL* and *NOS3* expression were dependent upon PI3K/Akt signaling which is known to inhibit the inflammatory response mediated by MAPKs and attenuate atherogenesis[30,47,48]. Inhibition of Akt activation by LY294002 treatment caused a reduction in *OASL* and *NOS3* expression together with a decrease in NO levels and an upregulation of adhesion molecules following the activation of Erk1/2 and NF-κB. Overall, these data indicate that *OASL* knockdown produces outcomes similar to that of suppressing Akt signaling, consistent with the concept that OASL-mediated maintenance of *NOS3* expression is regulated by upstream signaling through the PI3K/Akt pathway.

The effect of eNOS/NO bioavailability on endothelial homeostasis and the prevention of atherogenesis has been recognized for decades[5,6]. *Oasl1* deficiency reduced eNOS/NO bioavailability, which was revealed by a decrease in aortic NOS activity, cGMP, and plasma NO levels. This triggered the impairment of the relaxation response and resulted in an athero-susceptible and hypertensive environment in the vasculature, suggesting the idea that eNOS regulation concomitant to endothelial dysfunction is key to the phenotypic downstream of OASL1. Although the genetic absence of *Nos3* in *Apoe*[−/−] mice is known to exacerbate atherosclerosis, which reflects the anti-atherosclerotic role of eNOS-derived NO[49,50], *Nos3* overexpression is also unexpectedly found to promote atherogenesis as a result of eNOS uncoupling, which is caused by a lack of the eNOS cofactor, BH4[51,52]. Therefore, because targeting eNOS directly causes unpredictable outcomes, identifying the mechanism that maintains normal eNOS expression, rather than artificially increasing its level, may represent a viable therapeutic target for preserving vascular health. Both transcriptional and post-transcriptional processes were considered possible mechanisms to preserve the mRNA and protein levels of eNOS in ECs. Thus, we hypothesized that the OASL-mediated maintenance of *NOS3* levels depends on the ability of OASL to function as a post-transcriptional regulator and the availability of miR-NAs that target *NOS3* mRNA. Although additional studies are needed to fully elucidate the mechanisms by which OASL and miRNA cooperatively affect *NOS3* levels, our findings are consistent with previous reports that both OASL1 and OASL recognize and bind to dsRNA structures through the OAS domain[18,19,53].

The interplay between RNA-binding proteins and miRNAs is pivotal to maintaining the stability of target mRNAs[54,55]. Based on recent studies demonstrating the regulation of *NOS3* levels by several miRNAs[15–17,56], we hypothesized that OASL regulates miRNA candidates that reduce the persistence, rather than the production, of *NOS3* mRNA. Our miRNA sequencing analysis revealed that miR-584-5p expression, which was previously reported to inhibit *NOS3*[16], was augmented in *OASL*-KD HUVECs, whereas *NOS3* mRNA stability was reduced, as evidenced by its decreased half-life in the absence of continued transcription (i.e., presence of actinomycin D). Thus, our findings that treatment with an miR-584 mimic reproduced the effects of *OASL*-KD and that the administration of a miR-584 inhibitor restored *NOS3* expression and stability, established a function of miR-584, particularly in ECs. Because the 3′-UTR region of the *NOS3* transcript exhibits little conservation, whether OASL and OASL1 regulate the same miRNA targets in different species remains unclear. Nonetheless, the discovery of human OASL targets may, in turn, provide new therapeutic approaches. In addition, other miRNAs that were up-regulated in *OASL*-KD HUVECs may participate in cardiovascular disease, which suggests a potential use as markers of endothelial dysfunction[57,58]. Collectively, our study has established that OASL and OASL1 ameliorate EC dysfunction and atherogenesis, at least in part, by exploiting their effects in preventing plaque formation.

In conclusion, OASL1's functions go beyond mere inflammation, but it regulates also endothelial responses and biology with a protective net effect on the vasculature. We provided insight into the role of OASL1 as a modulator of *Nos3* expression that contributes to the maintenance of vascular homeostasis. Because murine endothelial OASL1-mediated atheroprotective effects are conserved in human OASL, controlling the OASL-miR-584-eNOS regulatory axis may represent an effective therapeutic strategy for treating human atherosclerosis and other vascular diseases mediated by endothelial dysfunction.

## Methods

### Human aortic tissue samples

Thoracic and abdominal aortic tissue samples, with or without atherosclerotic plaques, were harvested from patients during surgery ($n = 5$). Information including age over 50 years, diagnosis for atherosclerosis without other symptoms (hypertension, diabetes mellitus), and without statin treatment were concomitantly considered before being used for experiments. All human samples were collected at Severance Hospital, Yonsei University, Seoul, Korea. Written informed consent was obtained from all subjects. For the strict protection of privacy, all identifying information was removed from the samples before analysis. All procedures involving human subjects complied with all relevant ethical regulations and approved by the Institutional Review Board of Severance Hospital (4-2013-0688).

### Experimental animals

B6.129P2-*Apoe*[tm1Unc]/J (002052; *Apoe*[−/−]), B6.Cg-Tg(Tek-cre)12Fiv/J (004128; TIE2Cre), B6.129P2-*Lyz2*tm1[(cre)Ifo]/J (004781; LysMCre), and B6.129P2-*Nos3*[tm1Unc]/J (002684; *Nos3*[−/−]) strains were purchased from the Jackson Laboratory. The Tg(Cdh5-cre/ERT2)#Ykub strain was kindly donated by Yoshiaki Kubota (Keio University). B6.129P2-*Oasl1*[tm1Lms]/ (*Oasl1*[−/−]) mice were previously generated by deleting the promoter region and exon 2 of the *Oasl1* gene by homologous recombination[18]. The B6.*Oasl1*[tm1a(EUCOMM)Wtsi]/BcmMmucd (042237-UCD; *Oasl1*[tm1a]) strain, which was used to generate Oasl1-floxed (*Oasl1*[fl/fl]) mice, was purchased from the MMRRC. All transgenic mice were backcrossed onto a C57BL/6 background for more than 10 generations. Littermate control mice were used for all experiments and C57BL/6 J (000664; B6/J) strain was originally purchased from the Jackson Laboratory. The primer pairs used for the genotyping were listed in Supplementary Data 3. All mice were provided water and food *ad libitum*, were housed under a 12-hour light/dark cycle (light, 07:00 to 19:00 h) at 22 °C ± 2 °C with 55% ± 5% humidity and were euthanized with $CO_2$ before the experiments. All mice were bred in a specific pathogen-free facility and all experiments were approved by Institutional Animal Care and Use Committee (IACUC 19-004 and EWHA IACUC 21-065-1) of Ewha Womans University, Seoul, Korea.

### In vivo experiments and analysis of atherosclerotic plaques in mice

For the atherosclerosis studies, *Oasl1*[−/−] mice were crossed with *Apoe*[−/−] mice to generate *Oasl1*[−/−]*Apoe*[−/−] mice. The mice were maintained on a NCD for 28 weeks. For BMT experiments, *Oasl1*[−/−]*Apoe*[−/−] and *Apoe*[−/−] donor female mice were euthanized with $CO_2$ and the femurs with tibias were harvested. BM was flushed with sterile phosphate-buffered saline (PBS) and pooled. BM was depleted of mature T cells using CD5 antibody (Ab)-conjugated MACS beads (Miltenyi Biotec). 6-week-old *Oasl1*[−/−]*Apoe*[−/−] and *Apoe*[−/−] recipient male mice were lethally irradiated (10 Gy). After 24 h, BM cells from *Oasl1*[−/−]*Apoe*[−/−] or *Apoe*[−/−] controls ($1 \times 10^6$ cells) were resuspended in 100 µl of sterile PBS and intravenously injected into each *Oasl1*[−/−]*Apoe*[−/−] and *Apoe*[−/−] recipient mice. Vascular *Oasl1*[−/−]*Apoe*[−/−] and *Apoe*[−/−] chimeric mice were allowed to recover for 2 weeks and then fed a NCD for 28 weeks. To evaluate the specific effects of endothelial OASL1 on atherogenesis, we generated endothelial-specific *Oasl1*-deficient mice and created an atherogenic model by crossing Oasl1-floxed (*Oasl1*[fl/fl]) mice with *Cdh5*-cre/ERT2[+]

*Apoe*⁻/⁻ and injected tamoxifen (20 mg/mL dissolved in corn oil overnight) once every 24 h for 5 consecutive days before feeding with a western diet (WD; 150 g/kg body weight/day; Research Diets Western Diet D12079B, consisting of 0.15% cholesterol and 21% fat). After feeding with a WD for 13 weeks, *Oasl1*^fl/fl *Cdh5*-cre/ERT2⁺ *Apoe*⁻/⁻ and *Oasl1*^+/+ *Cdh5*-cre/ERT2⁺ *Apoe*⁻/⁻ mice were examined for plaque formation. *Oasl1*^fl/fl mice were crossed with *Tie2* or *Lyz2*-Cre-expressing *Apoe*⁻/⁻ mice were maintained on a NCD for 28 weeks and were examined for atherosclerotic plaque formation.

For in vivo studies, the mice were perfused with PBS via the left ventricle following euthanasia by $CO_2$ inhalation, and tissues from the heart and proximal ascending aorta to the bifurcation of the iliac artery were collected. For *en face* analyses, aortas were split longitudinally, pinned onto flat black silicone plates, and fixed overnight in 10% buffered formaldehyde in PBS. Fixed aortas were stained with Oil red O (Sigma Aldrich) overnight, washed briefly with PBS, and digitally imaged at a fixed magnification (Carl Zeiss, Axio Zoom.V16). Digitally processed images were subjected to morphometric measurements using AxioVision AC software.

## Immunofluorescence staining of aortic tissue
Following euthanasia, the mice were perfused with PBS followed by PBS containing 4% paraformaldehyde (4% PFA/PBS), after which aortic arches and descending thoracic aortas were isolated. *En face* preparations were blocked by incubation with normal goat serum and then incubated overnight at 4 °C with primary antibodies (see Supplementary Data 1), including rabbit polyclonal anti-mouse OASL1 (1:200), anti-α-SMA (1:200), Alexa Fluor 594 anti-mouse CD31 (1:300), Alexa Fluor 488 anti-mouse CD31 (1:300), and anti-eNOS (1:200) diluted in Antibody Diluent Reagent Solution (Invitrogen, 003118). After washing, the samples were incubated for 2 h at room temperature with the corresponding secondary antibodies (all 1:300, see Supplementary Data 1) including Alexa Fluor 488 Goat anti-Rabbit IgG (H + L), Alexa Fluor 594 Goat anti-Rabbit IgG (H + L) and Alexa Fluor 647 Goat anti-Rabbit IgG (H + L). For immunofluorescence staining of paraffin-sectioned human atheroma tissues, the samples were first incubated overnight at 4 °C with primary antibodies (see Supplementary Data 1) against OASL (1:200) or CD31 (1:200), and then incubated for 2 h at room temperature with secondary antibodies (all 1:300, see Supplementary Data 1) including Alexa Fluor 488 Goat anti-Rabbit IgG (H + L) and Alexa Fluor 594 Goat anti-Rabbit IgG (H + L). The aortic tissues or slides were then counterstained with 4′,6-diamidino-2-phenylindole (DAPI) and images were captured by confocal microscopy (Carl Zeiss, LSM 780).

## Plasma cholesterol measurements
Blood samples were collected in EDTA-coated tubes, incubated on ice for 30 min, and centrifuged at 1500 g for 15 min at 4 °C. The resulting plasma samples were subjected to biochemical analyses for the measurement of plasma glucose (GLU), triglycerides (TGs), total cholesterol (CHO), high-density lipoprotein-cholesterol (HDL-C), and low-density lipoprotein-cholesterol (LDL-C). All analyses were performed using a Hitachi blood autoanalyzer.

## Aortic single-cell preparation and flow cytometry analysis
For the preparation of aortic single cells, perivascular fat was carefully removed with microscissors under a dissecting microscope, and the isolated total mouse aortic segments were incubated with an enzyme mixture in Hank's balanced salt solution for 60 min at 37 °C with gentle shaking. The enzyme mixture contained 675 units/ml collagenase I (C0130), 187.5 units/ml collagenase XI (C7657), 90 units/ml hyaluronidase (H3884), and 90 units/ml DNase I (DN25) (all Sigma Aldrich). Aortic single cells were further processed by maceration through a 70-µm cell strainer (SPL). Fc receptors were blocked with a CD16/32 monoclonal antibody (eBioscience, 14-0161-82) and the cells were stained with the following fluorochrome-conjugated mouse antibodies: CD45:PerCP, CD11b:APC, CD64(FcYRI):BV421, Ly6G:BV650, CD11c:PE/Cy7, MHCII(I-A/I-E):APC/Cy7 (all 1:200, see Supplementary Data 1). An LSR Fortessa flow cytometer (BD Biosciences) was used to acquire the cytometric data, which was analyzed with FlowJo software (Three Star, Inc).

## Single-cell RNA sequencing (scRNA-seq) analysis
Aortic single cells isolated from the aortic root to the curvature as described above were stained with DAPI. BD FACSAria was used to sort live cells. Each cell suspension was subjected to 3′ single-cell RNA sequencing using the GemCode system (10x Genomics) according to the manufacturer's instructions. Libraries were sequenced on an Illumina HiSeq2500 platform and mapped to the mouse mm10 reference genome using the Cell Ranger toolkit (version 3.1.0). For each cell, we applied two quality measures to the filtered gene-cell-barcode matrix: mitochondrial genes (<10%) and gene count (>200) from the R package Seurat v4.0.3 (https://satijalab.org/seurat/). Doublets were detected and removed using DoubletFinder (version 2.0.3; https://github.com/chris-mcginnis-ucsf/DoubletFinder). The remaining data were log-normalized and then used in downstream analyses following z-transformation. The 2,000 top-ranked variably-expressed genes were selected using 'vst' algorithms in the Seurat FindVariableFeatures function and used to compute principal components (PCs). A subset of 12 significant PCs was selected using the Seurat ElbowPlot and JackStrawPlot functions. To correct for technical batch effects, we applied the Seurat integration method. Cell clustering and Uniform manifold approximation and projection (UMAP) visualization were performed using Seurat FindClusters and RunUMAP functions. To identify specific genes for each cell cluster, we applied the Seurat FindAllMarkers function using the default parameters. To characterize ECs, we extracted cell clusters based on the expression of canonical EC markers (*Cdh5*, *Pecam1*, *Fabp4*) and confidently defined 2,121 ECs. Total ECs were subclustered using the previously described procedure with 6 significant PCs and then used in the downstream analyses. To infer a biological function from the gene expression data, we performed a GSEA of the DEGs using clusterProfiler (version 4.4.2; https://github.com/YuLab-SMU/clusterProfiler). A list of DEGs was obtained using the MAST (version 1.20.0; https://github.com/RGLab/MAST) option of the Seurat FindMarkers function with default parameters. GO terms were obtained using the list of DEGs with a log2 fold-change as input to the clusterProfiler gseGO function. A P-value <0.05 was used as a cut-off.

## Ultrasound serial echocardiographic assessment
Serial echocardiography (echo) was performed using a VisualSonics Vevo 2100 machine, which is specifically designed for mice. Anesthesia was induced with 2% isoflurane and sustained with 1% isoflurane supplemented with oxygen during the echo procedure and mice with a heart rate of 360–460 bpm were used under these conditions. Hearts, aortic arches, and abdominal aortas were viewed and analyzed in M-mode, B-Mode, and PW-Mode. Age-matched *Oasl1*⁻/⁻*Apoe*⁻/⁻ mice and *Apoe*⁻/⁻ controls or *Oasl1*⁻/⁻, *NOS3*⁻/⁻ mice and *Oasl1*^+/+ controls, and vascular *Oasl1*⁻/⁻*Apoe*⁻/⁻ and *Apoe*⁻/⁻ chimeric mice were fed a NCD for 20 weeks. They were then used for assessments of blood flow dynamics and aortic diameters at athero-prone and athero-resistant aortic sites. Vevo strain analysis of the abdominal aorta was used for acquiring the value of the fractional area change (FAC). Parameters including ejection fraction, fractional shortening, and cardiac output in hearts were concurrently measured for assessments of cardiac function offline (Vevo software, version 5.5.1).

Endothelial WSS was calculated using a previously reported equation ($\tau_w = 4\mu Q / \pi d^3$)[24]. Blood viscosity and ultrasound data, including the internal diameter of arteries and blood flow velocity in each mouse aortic arch or abdominal aorta, were used to quantify in vivo endothelial WSS.

## BP measurement

Tail-cuff measurements of systolic, diastolic, and mean BP were obtained in age-matched $Oasl1^{-/-}Apoe^{-/-}$ mice and $Apoe^{-/-}$ controls or $Oasl1^{-/-}$, $NOS3^{-/-}$ mice and $Oasl1^{+/+}$ controls using the CODA-8 non-invasive BP System (Kent Scientific) according to the manufacturer's protocol. Briefly, mice were acclimated to the system over five consecutive days, followed immediately by BP measurements, which consisted of five acclimation readings and BP measurements 20 times at the same time using a multiple animal system. The tail temperature was monitored before and during the measurement and mice with a core temperature of 33 °C–35°C were used. The means of the accepted results from each mouse were used for data analysis.

## Western blot analysis

Whole-cell lysates of primary cultured HUVECs were obtained by lysing cells in RIPA buffer (50 mM Tris-HCl pH 7.4, 150 mM NaCl, 0.1% SDS, 1% NP-40, 0.25% sodium deoxycholate, 1 mM sodium fluoride, and 1 mM $Na_3VO_4$) containing a protease inhibitor cocktail (Roche Life Science, 11 697 498 001) and a phosphatase inhibitor cocktail (Roche Life Science, 04 906 837 001). For immunoblotting, the proteins were resolved by sodium dodecyl sulfate-polyacrylamide gel electrophoresis (SDS-PAGE) and transferred to PVDF (polyvinylidene difluoride) membranes. The membranes were blocked with 5% bovine serum albumin (BSA; Bovogen, BSAS0.1) in Tris-buffered saline (TBS; 50 mM Tris-Cl pH 7.5, 150 mM NaCl) containing 0.05% Tween-20 (0.05% TBST) and subsequently incubated with primary antibodies (all 1:1000, see Supplementary Data 1) in 1% BSA for overnight at 4 °C with gentle shaking. The membranes were then incubated with the corresponding secondary antibodies (all 1:1000, see Supplementary Data 1) including goat anti-rabbit IgG Antibody, (H + L) horseradish peroxidase (HRP) conjugate or rabbit anti-goat IgG Antibody, HRP conjugate. Immunoreactive bands were detected using ECL Western Blot reagents (GE Healthcare Life Sciences, RPN2106). ChemiDoc XRS + (BioRad), manual development, ImageLab Software (v6.1, BioRad), and ImageJ (v1.53c, NIH) were used to acquire and quantify western blot image data. Source data of the western blot are provided as a Source Data file.

## Enzyme-linked immunosorbent assay (ELISA)

Interferon beta (IFNB; R&D Systems, MIFNB0 [mouse] or DIFNB0 [human]) or tumor necrosis factor-alpha (TNFα; R&D Systems, MTA00B [mouse]) in the plasma of mice obtained from retro-orbital blood and in the culture supernatants of mouse or human aortic EC were measured according to the manufacturer's protocol using an ELISA.

## RNA isolation and real-time quantitative polymerase chain reaction (qPCR)

Total RNA was isolated using TRIzol reagent (Ambion, 15596018) and suspended in diethylpyrocarbonate (DEPC; Sigma Aldrich, D5758)-treated water and stored at −80 °C until use. Complementary DNA was synthesized using the RevertAid first-strand cDNA synthesis kit (Thermo Fisher Scientific, K1622). Real-time qPCR was performed using the KAPA SYBR FAST Master Mix (Kapa Biosystems, KK4602) with a StepOnePlus Real-Time PCR System (Applied Biosystems). Calculations were performed using the comparative method ($2^{-CT}$) with the housekeeping gene, $Gapdh$, or $GAPDH$ applied as an internal control. Some of the primers were obtained from https://www.origene.com/ and double checked by the NCBI Blast algorithm. The primer pairs used for the amplification of the mouse and human genes were listed in Supplementary Data 2. For pre-mRNA analysis of $NOS3$, the primer pairs were used according to a previous report[35].

For quantification of miRNA, the Mir-X miRNA First-Strand Synthesis Kit (TaKaRa, 638315) was used according to the manufacturer's protocol. Whole target miRNA nucleotide sequences were used as 5' primers and qPCR was performed using the KAPA SYBR FAST Master Mix. Calculations were done using the comparative method with U6 snRNA applied as an internal control.

## RNA-seq and miRNA-seq analysis

Total RNA from siCTL- or siOASL-transfected HUVECs treated with or without TNFα or IFNγ was extracted as described above and purified by ethanol precipitation. RNA-seq and miRNA-seq libraries were prepared using the TruSeq Stranded Total RNA LT Sample Prep Kit (Gold) and TruSeq Small RNA Library Prep Kit, respectively. The total RNA libraries were sequenced on the NovaSeq platform (Illumina) and the miRNA libraries were sequenced on the Hiseq2500 (Illumina). RNA sequence reads were counted using StringTie-2.1.3b or miRDeep2 (version 0.1.3; https://github.com/rajewsky-lab/mirdeep2). DESeq2 (version 1.34.0) was used to identify genes differentially expressed between siCTL- and siOASL-transfected HUVECs following treatment with TNFα and IFNγ. Genes showing <10 read counts in total were pre-filtered. $Q < 0.01$ was used as a cut-off for DEGs and a P-value ≤ 0.1 was established for differentially expressed miRNAs. Pathway enrichment ($p < 0.05$, $Q < 0.2$) of the DEGs was investigated using clusterProfiler. Significance values for the pathways were calculated using the enrichGO function in clusterProfiler.

## Primary cell isolation and culture

**Mouse aortic EC (MAEC).** MAECs were prepared from thoracic aortas of male and female $Oasl1^{+/+}$ and $Oasl1^{-/-}$ mice younger than 4 weeks of age. Briefly, each aorta was perfused with heparinized saline, opened, cut into several pieces, placed in Matrigel (Corning, 356234), and maintained in Endothelial Cell Growth Medium (EGM-2 BulletKit, Lonza, CC-3162) containing 10% fetal bovine serum (FBS), L-glutamine, and penicillin/streptomycin (P/S). After 3–4 days, the aortas were removed and the remaining MAECs were further cultured in Matrigel for up to 7 days. MAECs were isolated by treatment with dispase (Corning, 354235), maintained through passage 1, and used at passages 3–4 for experiments.

**Human umbilical vein EC (HUVEC).** HUVECs were obtained from the LONZA (C2519A). Before use, the HUVECs were tested for mycoplasma contamination and cultured in Endothelial Cell Growth Medium (EGM-2 BulletKit, Lonza, CC-3162) containing 10% FBS, L-glutamine, and P/S. HUVECs are guaranteed for 16 population doublings and to stain double-positive for CD31 and CD105 by manufacturer.

**Human aortic EC (HUAEC).** HUAECs were obtained from Promocell (c-12202). M199 media (Hyclone, SH30253.01) containing 20% FBS, heparin (5 units/ml), bFGF (3 ng/ml), and P/S was used for culturing HUAECs and the medium was changed to Endothelial Cell Growth Medium (EGM-2 BulletKit, Lonza, CC-3162) containing 10% FBS, L-glutamine, and P/S before the experiments. Cells at passages 5–7 were used for experiments to maintain their phenotype. HUAECs are guaranteed for 15 population doublings and to stain positive for CD31 and Dil-Ac-LDL uptake by manufacturer.

**Mouse vascular smooth muscle cell (VSMC).** Mouse VSMCs were derived from thoracic-abdominal aortas isolated from 3–4-week-old mice by removing the adventitia and endothelial layer using an enzyme mixture containing collagenase Type II (Worthington Biochemical, LS004174), soybean trypsin inhibitor (Worthington Biochemical, LS003570), and elastase (Worthington Biochemical, LS002279). They were maintained in DMEM/F12 media (Gibco, 11320) containing 20% FBS, L-glutamine, and P/S. Cells were used at passages under 10 for the experiments.

**Human aortic smooth muscle cell (HASMC).** HASMCs were obtained from the LONZA (CC-2571) and cultured in Smooth Muscle Cell Growth

Medium-2 (SmGM-2 SingleQuots Kit, Lonza, CC-3182) containing 5% FBS, L-glutamine, and P/S before the experiments. HASMCs are guaranteed for 15 population doublings and to stain positive for alpha smooth muscle actin and negative for von Willebrand Factor VIII after differentiation by manufacturer.

**Mouse bone marrow-derived macrophage (BMDM).** BM cells were isolated from the femurs and tibias of mice and cultured in DMEM containing 20% FBS and 20 μM β-mercaptoethanol (β-ME; Sigma-Aldrich, M6250) for 16 h. The suspended cells were plated and treated with 20 ng/ml recombinant human/mouse macrophage colony-stimulating factor (M-CSF; Peprotech, 300-25, 30 ng/ml) for 7 days to allow differentiation into macrophages.

### Reagents and inhibitors

MAECs, HUVECs and HUAECs were activated by treating with animal-free recombinant tumor necrosis factor alpha (TNFα; Peprotech, AF-315-01A [mouse] or AF-300-01A [human], 50 ng/ml) and animal-free recombinant murine interferon gamma (IFNγ; Peprotech, AF-315-05 [mouse] or AF-300-02 [human], 50 ng/ml) following serum starvation by incubating with serum-free EGM-2. The PI3K inhibitor, LY294002 (Cell Signaling Technology, 9901), MEK1/2 inhibitor, U0126 (Cell Signaling Technology, 9903), mTOR inhibitor, rapamycin (Cell Signaling Technology, 9904), NF-κB inhibitor, Bay-11 (Sigma, B5556), and the eNOS inhibitor, $N^G$-nitro-L-arginine methyl ester (L-NAME; Calbiochem, 15190-44-0, 1 mM), which are considered potent and selective inhibitors of their respective targets, were used to inhibit each signaling pathway. MAECs and HUVECs were treated with inhibitors for 1 h prior to stimulation at a concentration of 20 μmol/L. AccuTarget Negative Control siRNA (siCTL; SN-1003), siRNAs against human OASL (siOASL; SDO-1004: 8638) or human NOS3 (siNOS3; SDO-1001: 4846), and mouse Nos3 (siNos3; SDO-1001: 18127) were obtained from Bioneer (Korea). MAECs, HUVECs, and HUAECs were transfected with 80 nM siRNA using the Neon Transfection System (Invitrogen, MPK1096) and were used 48–72 h after transfection. AccuTarget miRNA Negative Control, mimic (SMC-2001) and inhibitor (SMC-2101), as well as human miRNA-584 mimic (SMM-003) and miRNA-584-5p inhibitor (SMI-002), were purchased from Bioneer (Korea) and used at a concentration of 80 nM.

### In vitro monocyte adhesion and transendothelial migration assay

CD11b$^+$ monocytes were isolated from the BM of $Oasl1^{+/+}$ and $Oasl1^{-/-}$ mice using CD11b Ab-conjugated MACS beads (Miltenyi Biotec, 130-049-601). Following RBC lysis, the BM cells were resuspended in 90 μl of MACS buffer (PBS pH 7.2, containing 0.5% BSA and 2 mM EDTA), then 10 μl of CD11b MicroBeads were added to each sample (per $1 \times 10^7$ total cells). The cells were incubated with beads for 20 min on ice, washed with MACS buffer, and magnetically sorted using MS Columns (Miltenyi Biotec, 130-042-201) according to the manufacturer's instructions. The CD11b$^+$ monocyte pellet was resuspended in RPMI-1640 medium (HyClone, SH30027.01) supplemented with 10% FBS, and the cells were counted for each experiment. The in vitro adhesion assays for CD11b$^+$ monocytes were performed in 24-well plates pre-coated with collagen. MAECs ($5 \times 10^4$) were plated in wells and stimulated with 10 ng/ml of TNFα or IFNγ for 6 h and then overlaid with $1 \times 10^5$ CFSE-labeled CD11b$^+$ monocytes. After 12 h, the wells were washed and the number of remaining cells that had adhered to the EC layer was estimated using a cell counter. The in vitro transendothelial migration assays for CD11b$^+$ monocytes were performed in 24-well plates fitted with inner wells containing polycarbonate filters with 5.0-μm pores (Corning, 3421) according to the manufacturer's instructions. MAECs ($5 \times 10^4$) were plated into the collagen-precoated inner wells, stimulated with 10 ng/ml of TNFα and IFNγ for 6 h, and then overlaid with $1 \times 10^5$ CFSE-labeled CD11b$^+$ monocytes. After 12 h, the

number of cells that had migrated into the lower chamber was estimated using a cell counter. Each experiment was performed using at least five donors per group.

### Immunocytochemistry analysis

For immunofluorescence staining of HUVECs, HUAECs, and MAECs, samples were fixed with 4% PFA/PBS, blocked with Ultra Vision Protein block (Thermo Fisher Scientific, TA-125-PBQ), and incubated for 2 h at room temperature with primary antibodies (see Supplementary Data 1) against human OASL (1:300) or mouse OASL1 (1:200) and eNOS (1:300), followed by 1 h at room temperature with secondary antibodies (all 1:300, see Supplementary Data 1) including Alexa Fluor 488 Goat anti-Rabbit IgG (H + L) and Alexa Fluor 594 Goat anti-Rabbit IgG (H + L). The cells were counterstained with DAPI, and the resulting images were obtained by confocal microscopy (Carl Zeiss, LSM 780).

### Nitric oxide synthase (NOS) activity measurement

NOS activity was measured using the NOS activity assay kit (abcam, ab211083) according to the manufacturer's protocol in whole aortas isolated from $Oasl1^{-/-}Apoe^{-/-}$ and $Apoe^{-/-}$ mice and in the cell lysates of MAECs and HUAECs incubated with or without L-NAME for 24 h. The samples were immediately assayed for NO synthesis activity by adding the substrate with cofactors and observing a series of reactions with Griess Reagent 1 and 2. The absorbance at 540 nm was used to calculate the relative enzymatic activity of the samples.

### Nitric oxide (NO) measurement

Total nitric oxide (NOx) levels were determined by measuring both nitrate and nitrite in the plasma of $Oasl1^{-/-}Apoe^{-/-}$ and $Apoe^{-/-}$ mice or in the supernatants of MAECs and HUAECs, respectively. Cells were incubated with or without TNFα and IFNγ or L-NAME for 24 h and the supernatants were collected. The plasma and supernatants were deproteinized using a 10 kDa column spin cut-off system (SARTORIUS, VS0101) and immediately assayed by the Griess method according using the NO assay kit (abcam, ab65328) according to the manufacturer's protocol. For the analysis of the supernatant, medium alone without cells was used as the negative control.

NO production in live cells was measured using the dye, 4-amino-5-methylamino-2′,7′-difluorofluorescein diacetate (DAF-FM DA; Invitrogen, D23844), according to the manufacturer's protocol. Briefly, for aortic tissue, mouse aortas were isolated, rapidly placed in media, split longitudinally, and pinned face up (en face preparation). Aortic tissues were pretreated with DAF-FM DA dye for 1 h, washed, and incubated in fresh media for 30 min. HUVECs were prepared and activated as indicated above. HUVECs were pretreated with DAF-FM DA for 20 min and washed for 15 min. Images were acquired by confocal microscopy (Carl Zeiss, LSM 780) with an FITC filter and the MFI was measured.

### Cyclic GMP measurement

For cGMP measurements, whole aortas isolated from $Oasl1^{-/-}Apoe^{-/-}$ and $Apoe^{-/-}$ mice were incubated with or without L-NAME for 24 h. HUAECs were co-cultured with HVSMCs at a ratio of 1 to 1 in the presence of L-NAME for 24 h. Intracellular cGMP levels were determined by ELISA (abcam, ab133052) and calculated relative to amount of total protein according to the manufacturer's protocol.

### NOS3 mRNA stability measurement

HUVECs and HUAECs transfected with siCTL or siOASL were allowed to stand in antibiotic-free EGM-2 media containing 1 μg/ml actinomycin D (Sigma-Aldrich, A9415), which was added to inhibit transcription. For the measurement of mRNA stability with miRNA mimic or inhibitor treatment, HUVECs and HUAECs were pre-transfected with siCTL or siOASL, and the miRNA mimics or inhibitors were post-transfected using the Neon Transfection System (Invitrogen, MPK1096), followed by incubation with actinomycin D for the indicated time periods. Total

RNA was harvested at 0, 1, and 3 h, reversed-transcribed into cDNA, and subject to qRT-PCR–based analyses to determine the decay rate of *NOS3* mRNA relative to the control.

## Vascular tension assay

Heparin was administered 1 h before the mice were sacrificed. The mice were anesthetized using isoflurane and the thoracic aorta from the aortic root to the bifurcation of the iliac arteries was rapidly isolated and cut into 1.5 mm rings. The aortic rings were placed in ice-cold oxygenated Krebs-Ringer bicarbonate buffer (118.3 mmol/L NaCl, 4.7 mmol/L KCl, 1.2 mmol/L $MgSO_4$, 1.6 mmol/L $CaCl_2$, 25 mmol/L $NaHCO_3$, and 11.1 mmol/L glucose, pH 7.4) and suspended between two wire stirrups (150 mm) in a myograph (Multi Myograph System; Hinnerup, DMT-620) containing 10 mL of Krebs-Ringer (95% $O_2$-5% $CO_2$, pH 7.4, 37 °C). One stirrup was connected to a three-dimensional micromanipulator and the other to a force transducer. The aortic rings were passively stretched at 10 min intervals in increments of 100 mg to reach the optimal tone of 600 mg. Dose responses to the vasoconstrictor phenylephrine (PE; 59-42-7, $10^{-9}$–$10^{-5}$ mol/L) were assessed, and responses to the vasodilators, acetylcholine (Ach; 60-31-1, $10^{-9}$–$10^{-5}$ mol/L) and sodium nitroprusside (SNP, 13755-38-9, $10^{-10}$–$10^{-6}$ mol/L), were measured after pre-constriction with PE ($10^{-5}$ mol/L). To further confirm the NO-dependent vasoconstriction activity, the aortic rings were treated with 1H-[1,2,4]oxadiazolo[4,3-a]quinoxalin-1-one (ODQ, 41443-28-1, $10^{-5}$ mol/L), a soluble guanylyl cyclase inhibitor (Sigma).

## Diethylenetriamine diazeniumdiolate (DETA/NO) treatment

DETA/NO (DETA NEONATE; Cayman, 60 mg/kg) was added intraperitoneally to $Oasl1^{-/-}Apoe^{-/-}$ and $Apoe^{-/-}$ mice at the age of 28-week-old for 7 days and BP was measured before and after treatment.

## Statistical analysis

Statistical analyses were performed using GraphPad Prism 8 software. An two-sided unpaired Student's *t*-test or nonparametric two-sided *Mann-Whitney* U-test was used for comparisons between two groups. A one-way or two-way analysis of variance (ANOVA) was used for comparisons among three or more groups with post-hoc test. All data are presented as means ± standard error of the mean (SEM) unless otherwise stated as means ± standard deviation (SD) and *P*-values ≤ 0.05 were considered statistically significant.

## Reporting summary

Further information on research design is available in the Nature Research Reporting Summary linked to this article.

## Data availability

All data generated during this study are available in the main text or the supplementary information files. Source data are provided with this paper. The single-cell RNA-seq data generated in this study have been deposited in the NCBI Gene Expression Omnibus (GEO) database under accession code GSE186355. The bulk RNA-seq data generated in this study have been deposited in the GEO database under accession code GSE186352. The miRNA-seq data generated in this study have been deposited in the GEO database under accession code GSE186354. GRCh38 [https://ftp.ebi.ac.uk/pub/databases/gencode/Gencode_human/release_38/GRCh38.primary_assembly.genome.fa.gz] and GRCm38 [https://ftp.ebi.ac.uk/pub/databases/gencode/Gencode_mouse/release_M25/GRCm38.primary_assembly.genome.fa.gz] were used for the reference genome. Source data are provided with this paper.

## Code availability

No unreported custom code was used in this manuscript and all code used is available from the authors upon request.

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

## Acknowledgements

The apparatus including LSM 780 NLO (Zeiss), LSRFortessa (Becton Dickinson), and FACSAria (Becton Dickinson) at Ewha Fluorescence Core Imaging Center were utilized for major experiments. We acknowledge Hyun Jung Park for designing the graphical illustration used in this study. The GTEx data were obtained from: the GTEx portal/dbGaP accession number phs000424.v8.p2 on 09/06/2022. This work was supported by a National Research Foundation of Korea (NRF) grant funded by the Korean government (NRF-2020R1A3B2079811; G.T.O. and NRF-2021M3E5E7023628; G.T.O.).

## Author contributions

T.K.K., S.J. and G.T.O. designed the experiments. T.K.K. performed most of the experiments with help from S.J. S.-K.S., S.S., J.S., J.J., H.Y.K., S.K., and S.H.M. also contributed to several experiments. O.K. performed the ultrasound measurement and N.K., H.-O.L., and Y.-J.K. provided essential reagents. B.-H.K. and Y.-M.K. contributed to the analysis of eNOS-related mechanisms and the relaxation response. S.P. and S.H.P. contributed to perform and analyze scRNA-seq, total RNA- and miRNA-seq. T.K.K. and G.T.O. wrote the manuscript. All authors discussed and revised the manuscript.

## Competing interests

The authors declare no competing interests.

## Additional information

[1]Heart-Immune-Brain Network Research Center, Department of Life Sciences, Ewha Womans University, Seoul, Republic of Korea. [2]Department of Biological Sciences, Ulsan National Institute of Science & Technology (UNIST), Ulsan, Republic of Korea. [3]Department of Biological Sciences, Kangwon National University, Kangwondae-gil 1, Chuncheon, Republic of Korea. [4]Department of Microbiology, College of Medicine, The Catholic University of Korea, Seoul, Republic of Korea. [5]Department of Biomedicine and Health Sciences, Graduate School, The Catholic University of Korea, Seoul, Republic of Korea. [6]Department of Molecular and Cellular Biochemistry, School of Medicine, Kangwon National University, Kangwondae-gil 1, Chuncheon 24341, Republic of Korea. [7]Department of Biochemistry, College of Life Science and Biotechnology, Yonsei University, Seoul, Republic of Korea. [8]Present address: Department of Biological Sciences and Biotechnology Major in Bio-Vaccine Engineering, Andong National University, Andong, Gyeongsangbuk-do, Republic of Korea. [9]Present address: McKay Orthopaedic Research Laboratory, Department of Orthopaedic Surgery, Perelman School of Medicine, University of Pennsylvania, Philadelphia, PA 19104-6081, USA. [10]These authors contributed equally: Tae Kyeong Kim, Sejin Jeon. ✉e-mail: gootaeg@ewha.ac.kr

