## [Peer Review File · Nature Communications]

OASL1 Protects Against Atherosclerosis by Maintaining Endothelial Nitric Oxide Synthase mRNA StabilityReviewers' Comments:

Reviewer #1:

Remarks to the Author:

The OASL family members are known as regulators of cytosolic nucleic acid sensors with a role in anti-viral immunity (Ghosh, Immunity 2019). The authors have previously shown that OASL1 is involved in an inhibitory feedback loop downstream of IFNs (Nat Imm 2017). In this manuscript they focus on the role of OASL1 in endothelium in atherosclerosis. The authors show that the expression of OASL1 is increased in endothelial cells in atheroprone regions of the aorta in a murine model of atherosclerosis and then in human plaques. In a global ko larger and more inflammatory plaques develop in the absence of changes in atherogenic lipoprotein circulating levels. The phenotype is of increased atherogenesis, pointing to an immunomodulatory role of OASL1 in the whole body. The manuscript demonstrates the anti-atherogenic effect of OASL1 in atheroprone regions by using whole body deficient mice and complex bone marrow and EC-conditional depletion via Tie2-cre. The loss of OASL1 in endothelial cells phenocopies the phenotype of the global ko, indicating that OASL1 function in endothelial cells is athero-protective. A single cell transcriptomic analysis of the aortas of these mice uncovers an enrichment of immune cell subsets, endothelial activation with enhanced ability to recruit inflammatory cells, and heterogeneity of endothelial cell subsets. One cluster of aortic EC was enriched with atherogenic genes and was characterised by a transcriptomic signature consistent with a decreased NO biosynthetic activity. Nos3 gene expression in the aorta is downregulated in Oasl1^{-/-} ApoE^{-/-} mice in early and late atherogenesis. In the single cell and bulk transcriptomic dataset an enrichment of Nos-related genes is observed. Knockdown of OASL1 in HUVECs in vitro downregulates Nos3 expression and phosphorylation of Akt in a Pi3K-dependent manner, possibly involving also post-transcriptional effect mediated by miR584.

The execution of the study relies on the use of a variety of complementary approaches spanning from complex chimeras and conditional ko, single cell RNASeq and signalling to demonstrate the anti-atherogenic role of Oasl1.

Interferon (IFN)-inducible genes such as OASL/OASL1 are abundantly expressed in the human and murine vasculature. Yet their function in the vasculature is largely unexplored. Additionally, the link between endothelial function and innate sensors is novel.

Key general comments

1. The phenotype of OASL1-deficient mice is quite complex and it involves changes in atherogenesis, blood pressure, endothelial cell heterogeneity and haemodynamic changes. This is in keeping with the key function of the endothelium across the vasculature. It would be important to discriminate a little more between cause and effect in this complex phenotype by examining the phenotype of Oasl1^{-/-} at steady state and by examining the temporal relationship between the emergence of atherogenesis and hemodynamics/pressure readouts. For instance, it would be important to clarify whether the increase in BP is present in Oasl1^{-/-} mice in absence of hyperlipidemia and whether in hyperlipidemic mice it precedes the initiation of atherogenic lesions in a kinetic study. Similarly, hemodynamic changes, such as wall shear stress, might relate to plaque growth and the subsequent lumen restriction, rather than to endothelial function: it is unclear if the changes are secondary to the development of lesions. Expanding the analysis to control mice or young ApoE^{-/-} mice in key readouts (including haemodynamic measurements) could be important to distinguish the primary and secondary features of the phenotype.

2. It is unclear whether in the in vivo study the immunomodulatory effects of OASL1 are completely mediated by regulation of Nos3 in the endothelium. The in vitro mechanism could be disjointed from the in vivo phenotype. The phenotype in vivo of the Oasl1 deficiency is highly linked to regulation of inflammation and the authors themselves and others have shown that Oasl1 can interfere with other inflammatory pathways. Could other functions of Oasl1 account for its effects? Additional mechanistic data and mining of the transcriptional data would be very useful to underscore the authors' conclusion. The study of a Nos3^{-/-} chimera could be useful to clarify the mechanism of protection.

3. An abundance of the mechanistic discovery contained in the manuscript (regulation of Nos3 by

OASL1) relies on the use of HUVECs, a set of endothelial cells that have per se a very immunity-friendly phenotype and not wholly accepted as a model of arterial endothelium. Albeit not insurmountable, this is a major flaw of the study. The key findings in HUVECs will need to be repeated with human primary aortic or coronary endothelial cells at an early passage in culture (to maintain phenotype).

4. The mechanism of increase in OASL1 expression in atheroprone region goes unexplained.

Major specific comments

5. Fig. 1a Regarding the expression of human plaques of OASL1: is it expressed on the luminal endothelium or on neo-vessels that are enriched in the human atheroma.

6. The mechanisms of upregulation of OASL1 at atheroprone regions is unclear. Is OASL1 expression dependent on haemodynamic or inflammatory changes in the atheroprone regions of the aorta?

7. Does the expression of OASL1 in atheroprone regions predates the development of atherosclerotic lesions?

8. Suppl. Figure 6. Do the haemodynamic changes appear before the inception of atherosclerotic lesions and lumen impingement? In other words, are those changes primary or secondary to the lesions formation?

9. Figure 2c: the authors should comment on the changes in specific populations and their identity in the single cell experiment. For instance: How does OASL1 regulate endothelial cell heterogeneity in single cell figure. What is the relative/absolute reduction in smooth muscle cells? What is the meaning and the increase in macrophage cluster 1? An increase in NKT cells and a slight decrease in ECs is also noted. What is their significance to the phenotype?

10. Fig. 4. The blood pressure is elevated in OASL1 deficient hyperlipidemic mice. Is blood pressure increased in OASL1-deficient mice without loss of ApoE? Is the vascular and haemodynamic phenotype dependent or independent of hyperlipidemia? Are the changes in blood pressure contributing to the increase in atherogenesis? Is Nos3 involved in the blood pressure changes?

11. Suppl. Figure 2. Is Tie2-cre the best tool to target endothelial cells?

12. Additional experimentation would be useful to show that the in vivo phenotype is mediated by Oasl1-dependent regulation of Nos3. What is the phenotype of Nos3 deficient mice? Would they phenocopy the OASL1 ko mice?

13. Fig 7a and Suppl. Fig 2 should be consolidated as Main Figure because the data they contain are highly complementary and key to the message (role of the endothelial expression of Oasl1). Fig 7a is missing Oasl1+/+ bone marrow controls, while Suppl. Figure 2 is missing the conditions of Oasl1-/- in both hematopoietic and non-hematopoietic cells. The effect of the global ko on lesions size (5%) seems inferior to the non-hematopoietic deficiency of Oasl1 (10%). I suggest also adding diagrams to help the readership navigate complex chimera experimental designs.

Minor comments

Line 134 (Suppl. Fig. 1a). The definition of what constitutes EC-rich human tissue is unclear. Pls rephrase indicating aorta and coronary artery as appropriate.

Line 181 why not all samples are selected? Its unclear why a response-dependent selection is used.

Line 402 Sentence is unclear.

Line 426 Sentence is unclear.

Line 440 "vitalising" should be replaced

Figure 1b The cellular localization of OASL1 should be better rendered. CD31 staining is also observed within the plaque, not only in the endothelial monolayer above the plaque. Are platelets also expressing it? Is it expressed in neovessels?

Figure 1f "origin of the artery" is unclear. Do the authors mean the "aortic root"?

Figure 1h: why is the stimulation with both IFN and TNF needed for upregulation? What is the effect of the single stimulation?

Figure 3g the relevance of the EC subclusters is unclear.

Figure 3i Was Nos3 differentially regulated by loss of OASL1 in vivo?

Figure 4c: was the blood pressure measured invasively or non-invasively?

Figure 5 pERK: large variability in the data set

Reviewer #2:

Remarks to the Author:

This is an interesting and detailed paper describing the effects of OASL1 in atherosclerosis. OASL1 has been previously shown, and is confirmed in further experiments done by the authors in the present study, to be a regulator of multiple gene pathways, particularly those involved in inflammation. The authors focus on a potential link between OASL1 and endothelial nitric oxide synthase [NOS3] function mediated by alteration of NOS3 mRNA stability.

Although potentially interesting, and including an impressive number of experiments, there are some fundamental limitations of the study that undermine the validity of some of the authors conclusions, and fall short of demonstrating a causal role for an OASL1 – eNOS - NO pathway in atherosclerosis. These include:

1. the choice of mouse lines to test global, bone marrow, endothelial cell and vascular deletion of OASL1 are problematic and are not applied in a consistent and valid manner throughout the experiments. For example, bone marrow chimeric mice, although widely used to test the effects of bone marrow versus non bone marrow deletion, have potential limitations in atherosclerosis because the repopulation of the irradiated recipient bone marrow with transferred cells will inevitably alter the immune repertoire. The presumption that transplantation of knockout bone marrow in to non knockout recipients will create a vascular cell specific knockout is completely invalid, even less for creation of an endothelial cell specific knockout. This approach, at best, will enable comparison of bone marrow derived versus all other cell type knockouts. The Tie2cre driver is also used as a model of EC knockout, but it is now widely known that Tie2cre is also highly effective at deleting floxed alleles in all bone marrow cells, indeed in the haematology literature it is used as a bone marrow specific Cre driver line. It would be preferable to use a more recently characterized and validated EC specific create driver such as VE Cadherin Cre. Apart from these technical limitations, the authors are too imprecise and vague in their use of terminology, implying presumptions which are invalid and greatly undermine their experiments. For example, describing bone marrow chimeric animals as "myeloid specific" KO is completely incorrect. They are bone marrow KOs, which will include all bone marrow cells and any other cells in the mouse that are derived from bone marrow, which may even include some vascular cells, given the clear evidence for vascular re-population by bone marrow derived stem cells particularly in atherosclerosis. Testing a myeloid specific knockout would require Cre drivers such as LysM Cre.

2. It is not clear why OASL1 deletion leads to changes in blood flow velocity, rather than the expected biological effect of disturbing endothelial cell inflammatory responses, ie altered inflammatory responses to the same level of blood flow or shear stress. Given that the authors state that heart rate and cardiac output we're not different in the OASL1 knockout, why would it be expected, and how is it explained, that blood flow velocity, or the hemodynamic effects of blood flow on the vascular wall, are altered? In the absence of any change in cardiac output, this would have to implicate a change in vascular dimensions, geometry or physico-elastic properties. This whole area of observations needs to be critically re-examined and supported by objective explanations.

3. The focus of the study is on NOS3 mRNA levels and stability, which are certainly tested very thoroughly, but the biological link with and eNOS protein, enzymatic activity and NO bioavailability is very weakly evidenced. In line 235, the authors jump straight from gene expression network analysis implicating endothelial nitric oxide synthase, (which is bound to be highly annotated and overrepresented in endothelial cell gene expression data sets) to a causal role for an NO bioactivity

which is at best speculative, and subsequently only weakly supported by later experiments. In particular, the hallmarks of and the NO effects and bioactivity have not been addressed. The use of DAF immunofluorescence is not adequate as a quantitative readout of eNOS activity or NO bioactivity. The critical importance of this conclusion requires support by complementary methods, given that this is known to be a technically challenging read out. This might include radio labelled arginine to citrulline conversion, measurement of cyclic GMP, EPR spin trapping techniques to detect authentic NO. Also functional studies should be done of endothelial function on isolated conduit and resistance vessels to quantify some of the classic read out of endothelial NOS activity such as relaxation responses to appropriate agonists or flow. Similar studies should be done in ECs.

4. The finding that the OASL1 knockout mouse has a high blood pressure is interesting, and consistent with abnormalities in endothelial cell function. However, this finding raises the potential of confounding in the observed effects on atherosclerosis. Specifically, high blood pressure would be expected to increase endothelial cell activation and increase atherosclerosis, irrespective of the underlying mechanism. Ideally, this should be addressed to see whether the high blood pressure phenotype is specifically mediated by endothelial OASL1, and how the effects on atherosclerosis could be controlled for, with appropriate blood pressure comparisons.

5. As the authors describe, multiple different pathways in endothelial and other cell types are impacted by OASL1. How do the authors have confidence in their conclusion that the effects of OASL1 are mediated specifically by endothelial nitric oxide synthase and NO, rather than the multiple other inflammatory pathways, all of which are credible and/or proven causal pathways in atherosclerosis? Is OASL1 implicated in human GWAS? Would a OASL1 knockout in ECs with eNOS knock down, or in a NOS3 knockout mouse no longer impact the inflammatory pathways or the development of high blood pressure or atherosclerosis?

6. Similarly, what are the other known or potential targets for Mir584? What are the effects of Mir584 agomirs or antagomirs in NOS3-deleted cells?

Reviewer #3:

Remarks to the Author:

In this study the authors uncover a regulatory role of endothelial expression of 2'-5' oligoadenylate synthetase-like 1 (OASL1) in maintaining eNOS mRNA stability under athero-prone conditions and consider its clinical implications.

Lacking endothelial Oas1 accelerated plaque progression, an effect preceded by endothelial dysfunction, following an elevated inflammatory response accompanied by decreased NO bioavailability, reflecting impaired eNOS expression.

Mechanistically, knockdown of PI3K/Akt signaling-dependent OASL expression induced an increase in Erk1/2 and NF- κ B activation and decreased the NOS3 mRNA levels by up-regulating the negative regulatory, miR-584, while miR-584 inhibitor rescued OASL knockdown effects. Their findings suggest that OASL1/OASL is a novel protector of NOS3 mRNA and that targeting miR-584 could be a viable therapeutic strategy for eNOS maintenance in vascular diseases.

Although the paper is well written and data support conclusion I have several issues with the manuscript:

Major:

Fig 1: The authors focus in human tissues on atherome plaques in the aorta, will it possible to show data from more distal arteries like coronary in which atherosclerosis will have a greater impact. N=3 should be increase to 5 and quantification of the IF should be provided. Finally, western blot should be performed.

Fig 2 do you see any apoptosis of EC ?

Fig 3 : is there any way to quantify directly NO production ?, what are the effects on other NOS expression.

Fig 4: what are the effects on cardiac output ? Can you reverse the increase in pressure by giving NO to the mice?

Fig 5: given the effect on AKT signalling, GSK3 a regulator of NFAT a critical TF regulating inflammatory is likely affected. Can you provide data on GSK3 and NFAT ?

Point-by-point response to the reviewers' concerns

Reviewer #1 (Remarks to the Author):

We appreciate your consideration and valuable comments regarding our study. Your comments have helped us to improve our manuscript. Below, we respond to your comments point by point. Original referee's comments are shown in black, whereas our answers are in blue and the appropriate changes in the revised manuscript are green. In particular, the square box indicates an overall summary of our conclusions for the corresponding question.

The OASL family members are known as regulators of cytosolic nucleic acid sensors with a role in anti-viral immunity (Ghosh, Immunity 2019). The authors have previously shown that OASL1 is involved in an inhibitory feedback loop downstream of IFNs (Nat Imm 2017). In this manuscript they focus on the role of OASL1 in endothelium in atherosclerosis. The authors show that the expression of OASL1 is increased in endothelial cells in athero-prone regions of the aorta in a murine model of atherosclerosis and then in human plaques. In a global ko larger and more inflammatory plaques develop in the absence of changes in atherogenic lipoprotein circulating levels. The phenotype is of increased atherogenesis, pointing to an immunomodulatory role of OASL1 in the whole body. The manuscript demonstrates the anti-atherogenic effect of OASL1 in athero-prone regions by using whole body deficient mice and complex bone marrow and EC-conditional depletion via Tie2-cre. The loss of OASL1 in endothelial cells phenocopies the phenotype of the global ko, indicating that OASL1 function in endothelial cells is athero-protective. A single cell transcriptomic analysis of the aortas of these mice uncovers an enrichment of immune cell subsets, endothelial activation with enhanced ability to recruit inflammatory cells, and heterogeneity of endothelial cell subsets. One cluster of aortic EC was enriched with atherogenic genes and was characterised by a transcriptomic signature consistent with a decreased NO biosynthetic activity. Nos3 gene expression in the aorta is downregulated in *Oasl1*^{-/-} *ApoE*^{-/-} mice in early and late atherogenesis. In the single cell and bulk transcriptomic dataset an enrichment of Nos-related genes is observed. Knockdown of OASL1 in HUVECs in vitro downregulates Nos3 expression and phosphorylation of Akt in a Pi3K-dependent manner, possibly involving also post-transcriptional effect mediated by miR584.

The execution of the study relies on the use of a variety of complementary approaches spanning from complex chimeras and conditional ko, single cell RNASeq and signalling to demonstrate the anti-atherogenic role of *Oasl1*.

Interferon (IFN)-inducible genes such as OASL/OASL1 are abundantly expressed in the human and murine vasculature. Yet their function in the vasculature is largely unexplored. Additionally, the link between endothelial function and innate sensors is novel.

Key general comments

1. The phenotype of OASL1-deficient mice is quite complex and it involves changes in atherogenesis, blood pressure, endothelial cell heterogeneity and hemodynamic changes. This is in keeping with the key function of the endothelium across the vasculature. It would be important to discriminate a little more between cause and effect in this complex phenotype by examining the phenotype of *Oasl1*^{-/-} at steady state and by examining the temporal relationship between the emergence of atherogenesis and hemodynamics/pressure readouts.

For instance, it would be important to clarify whether the increase in BP is present in Oasl1 -/- mice in absence of hyperlipidemia and whether in hyperlipidemic mice it precedes the initiation of atherogenic lesions in a kinetic study. Similarly, hemodynamic changes, such as wall shear stress, might relate to plaque growth and the subsequent lumen restriction, rather than to endothelial function: it is unclear if the changes are secondary to the development of lesions. Expanding the analysis to control mice **or** young ApoE-/- mice in key readouts (including hemodynamic measurements) could be important to distinguish the primary and secondary features of the phenotype.

*Oasl1* deficiency exhibited low wall shear stress (WSS) as a primary feature and hyperlipidemia-activated endothelial dysfunction promoted secondary phenotypes including plaque formation and an increase in blood pressure (BP).

- Based on the comments of the reviewer, we designed and performed specific experiments to distinguish the primary and secondary features of the phenotype. For this, we confirmed the sequential change of the hemodynamic velocity with WSS and BP from 8 to 28 weeks of a normal chow diet (NCD) under basal and hyperlipidemic conditions.

- Under steady state conditions, *Oasl1* deficiency exhibited lower mean blood velocity and WSS, particularly in the lesser curvature of the aortic arch (**Supplementary Fig. 10b**) without changes in BP compared with the controls (**Supplementary Fig. 10c, d**). This is indicative of endothelial dysfunction that can be the cause of vascular functional impairment. The phenotypic changes in the absence of hyperlipidemia are presented in Supplementary Fig.10 and described in the Results section (lines 255-263) of the revised manuscript.

**Supplementary Fig. 10**

[When the *Oas1*-deletion effect was assessed without hyperlipidemic conditions (Supplementary Fig. 10a), blood velocity and WSS were significantly lower at the lesser curvature of the aortic arch in both *Oas1*^{-/-} and *Nos3*^{-/-} mice compared with the *Oas1*^{+/+} controls (Supplementary Fig. 10b, c) and did not show a difference in BP between *Oas1*^{-/-} and *Oas1*^{+/+} mice, in which the value was smaller compared with that in *Nos3*^{-/-} mice (Supplementary Fig. 10d). These results suggest that *Nos3*^{-/-} mice phenocopied *Oas1*^{-/-} mice with respect to endothelial dysfunction from normal conditions and hyperlipidemia-activated endothelial dysfunction promoted an increase in BP that attributed to a reduction of eNOS/NO bioavailability in the vasculature of *Oas1*^{-/-}*Apoe*^{-/-} mice.]

→ Under hyperlipidemic conditions, *Oas1*^{-/-}*Apoe*^{-/-} mice similarly exhibited a lower velocity and WSS in the lesser curvature of the aortic arch and the abdominal region of the aortas, particularly from 16- to 20-week-old compared with *Oas1*^{+/+}*Apoe*^{-/-} mice (Supplementary Fig. 5), indicating a more athero-susceptible endothelial phenotype under hyperlipidemic conditions. These findings were added to Supplementary Fig.5 and described in the Results section (lines 194-200) of the revised manuscript.

Supplementary Fig. 5

[*Oas1* deficiency resulted in decreased flow velocity and wall shear stress (WSS) in athero-prone regions, such as the lesser curvature of the aortic arch and abdominal branch point in *Apoe*^{-/-} mice compared with littermate *Apoe*^{-/-} controls (Supplementary Fig. 5b, c). These phenomena were observed from 8 to 20 weeks of normal chow diet (NCD) and disappeared at 28 weeks when the plaques had developed and the aortic diameter was reduced (Fig. 2a, Supplementary Fig. 5). Collectively, these

findings demonstrate that the loss of *Oasl1* promotes endothelial dysfunction–dependent atherosclerotic plaque formation, particularly in athero-susceptible regions of the aorta.]

- Moreover, *Oasl1*^{-/-}*ApoE*^{-/-} exhibited a significant increase in BP and lesion formation at 28 weeks of NCD, although this was not significantly different from 8 to 20 weeks of measurements (Fig. 5c, Supplementary Fig. 8a). These new results regarding BP are shown in Fig. 5 and Supplementary Fig. 8, and are explained in the Results section (line 245-250) of the revised manuscript.

Fig. 5 and Supplementary Fig. 8

[A decrease in eNOS/NO bioavailability is known to contribute to vascular resistance and augmentation of BP.^{26, 27} *Oasl1* deficiency augmented mean blood pressure (BP; $P = 0.0465$ at 28 weeks) following an increase in both systolic ($P = 0.0425$) and diastolic ($P = 0.0085$) BP in *ApoE*^{-/-} mice during atherosclerotic conditions compared with controls (Fig. 5c, Supplementary Fig. 8a), whereas NO supplementation significantly reduced BP in both groups (Supplementary Fig. 8b).]

- Collectively, these data indicate that increased endothelial dysfunction, revealed by lowered WSS is the primary feature of *Oasl1*-deficient mice and hyperlipidemia-mediated endothelial activation and dysfunction accelerate both atherosclerotic plaque formation and the augmentation of BP in *Oasl1*^{-/-}*ApoE*^{-/-} mice as a secondary phenotype.

2. It is unclear whether in the in vivo study the immunomodulatory effects of OASL1 are completely mediated by regulation of Nos3 in the endothelium. The in vitro mechanism could be disjointed from the in vivo phenotype. The phenotype in vivo of the *Oasl1* deficiency is highly linked to regulation of inflammation and the authors themselves and others have shown that *Oasl1* can interfere with other inflammatory pathways. Could other functions of *Oasl1* account for its effects? Additional mechanistic data and mining of the transcriptional data would be very useful to underscore the authors' conclusion. The study of a *Nos3*^{-/-} chimera could be useful to clarify the mechanism of protection.

It is well known that endothelial cell (EC) dysfunction can initiate immune cell infiltration into the intima of the aorta resulting in progressive atherogenesis (Davignon et al, Circ. 2004, Weber et al, Nat Med. 2011). And further our EC-specific knockout studies (Cdh5-Cre/ERT2, and Tie2-Cre) clearly demonstrated that EC-specific *Oas1* deficiency accelerates atherogenesis.

We concur that we should consider other mechanisms, but we did not observe differences in type I interferon (IFN)-related pathways in *Oas1*-deficient ECs. *Nos3* deficiency initiated the inflammatory response by up-regulating adhesion molecules and activating ERK1/2 and NF- κ B in addition to *Oas1* deficiency, which were derived from their anti-inflammatory and protective function in the vasculature.

- First, we checked type I interferon (IFN)-related molecules, which are known to be associated with and regulated by OASL1. However, aortic type I IFN expression and plasma IFN α levels were not different between *Oas1*^{+/+}*ApoE*^{-/-} and *Oas1*^{-/-}*ApoE*^{-/-} mice under atherosclerotic conditions (**Supplementary Fig. 17a, b**). Therefore, we looked for another target of OASL1 in ECs. Moreover, we did not observe differences in type I IFN levels, including IFN α and IFN β (**Supplementary Fig. 17d-f**) or the activation of IFN-regulatory factor 7 (IRF7) and IRF3 in ECs following stimulation (**Supplementary Fig. 17g**).
- Furthermore, we examined human ECs by mining our transcriptional data of HUVECs and identified that human OASL-KD also did not affect type I IFN and IFN-stimulated gene expression (**Supplementary Fig. 16a, b**). Consistent with the results of MAECs, secreted IFN α levels (**Supplementary Fig. 16c**) and the activation of IFN-related signaling pathways (**Supplementary Fig. 16d**) were not different between OASL-KD and control HUAECs, suggesting the existence of other pathways that drive *Oas1* deficiency-mediated endothelial dysfunction.
- These data have been added to Supplementary Fig.16 and Supplementary Fig.17 and are described in the Results section (lines 342-346) of the revised manuscript.

Supplementary Fig. 16

Supplementary Fig. 17

[However, the type I IFN-related pathway, which is a pro-inflammatory mechanism affected by OASL and OASL1, did not significantly change with *Oasl1* deficiency in HUVECs and MAECs (Supplementary Fig. 16, 17), demonstrating that pro-

inflammatory and adhesion events are aggravated in OASL-KD HUVECs primarily through attenuated eNOS expression and enhanced MAPK and NF- κ B signaling.]

→ Following the reviewer's comment, we conducted additional mechanistic studies by reducing *NOS3* expression. *Oas1*^{-/-} MAECs were transfected with a *Nos3* siRNA to knockdown (KD) *Nos3* and HUAECs were concurrently transfected with siOASL and siNOS3. When we compared the inflammatory pathways affected by *Oas1* deficiency and *OASL*-KD, *NOS3*-KD had an additive effect on the reduction of *OASL* with respect to inflammatory activation because of its anti-inflammatory and protective effects of *NOS3* (**Supplementary Fig. 15**). In support of this study, there have already been reports that the reduction of *Nos3*-eNOS expression intensifies inflammatory activation and leukocyte adhesion during atherosclerosis (Lefer et al. Am J Physiol. 1999; Otsuka et al. Nat Rev Cardiol. 2007). These results are presented in the Results section (lines 325-330).

Supplementary Fig. 15

[Reduced eNOS in ECs exerts a variety of physiological responses including an increase in inflammatory signaling during pathological conditions.^{25, 28} Whether the reduced OASL-mediated decrease in *NOS3* affects inflammatory activation was further confirmed by *NOS3*-KD in HUAECs and MAECs. *OASL*'*NOS3*-KD HUAECs and *Nos3*-KD/*Oas1*^{-/-} ECs increased Erk1/2 and NF-κB activation with upregulation of adhesion molecules including *ICAM1* and *SELE* compared with *OASL*-KD or *Oas1*^{-/-} and the controls (Supplementary Fig. 15).]

- Our data suggest that reduced *NOS3* mRNA following a reduction of *OASL* (Fig. 7b) preceded the activation of inflammatory pathways, such as Erk1/2 or NF-κB (Fig. 6d, e) and an increase in adhesion molecule expression (Fig. 6g) that may be a possible cause of later inflammatory responses (Fig. 2d, 4a). Moreover, the expression of *Nos3* mRNA and eNOS protein was already reduced in *Oas1*-deficient endothelium from 8-week-old *Apoe*^{-/-} mice before disease progression (Fig. 5f, g). Thus, we can infer that inflammatory signaling was accelerated in *NOS3*-KD with *Oas1*^{-/-} or *OASL*-KD ECs compared with only *Oas1* deficiency. Collectively, an increase in leukocyte infiltration, inflammatory response, and plaque formation corresponding to the final in vivo phenotypes of *Oas1*-deficient mice may be caused by augmentation of endothelial dysfunction following a reduction of endothelial *Nos3* levels during early atherosclerosis.

3. An abundance of the mechanistic discovery contained in the manuscript (regulation of *Nos3* by *OASL1*) relies on the use of HUVECs, a set of endothelial cells that have per se a very immunity-friendly phenotype and not wholly accepted as a model of arterial endothelium. Albeit not insurmountable, this is a major flaw of the study. The key findings in HUVECs will need to be repeated with human primary aortic or coronary endothelial cells at an early passage in culture (to maintain phenotype).

- Based on the reviewer's comment, we added results using human arterial endothelial cells (HUAECs) at passages 5–7. We identified *OASL* and *NOS3* expression under activated conditions and *NOS3* mRNA stability following *OASL* reduction. In HUAECs, *OASL* expression was also increased with TNFα and IFNγ treatment and when transfected with si*OASL*, *NOS3* mRNA and protein levels were reduced as was the case for HUVECs. Moreover, we identified the effect of miR584 agonism and antagonism in HUAECs by transfecting miR-584 mimics or an inhibitor, which reduced and rescued *NOS3* mRNA stability, respectively. These findings are presented in Fig.1, Fig. 5, and Supplementary Fig. 19 and described in the Results section of the revised manuscript.

Fig. 1 and Fig. 5

Supplementary Fig. 19

- Other mechanistic studies done during the revision were performed using HUAECs and these results are presented in Supplementary Fig. 15 and 16 of the revised manuscript.
- The information utilizing HUAECs is in the Methods section of the revised manuscript.

4. The mechanism of increase in OASL1 expression in athero-prone region goes unexplained.

The expression of OASL1 is increased in the athero-prone region of aortas during early hyperlipidemic conditions, in which athero-prone shear initiates an inflammatory response. The combination of shear and inflammation induces OASL's region-specific expression.

- The mechanism that is responsible for the up-regulation of OASL1 in the athero-prone region is largely divided into two processes, which includes hemodynamics and inflammatory changes. In vivo OASL1 expression was up-regulated in aortic ECs, located where the athero-prone shear was

created, in the aortas of *Apoe*-deficient mice (*Apoe*^{-/-}; **Fig. 1f, g**). Under in vitro conditions, OASL expression increased following an oscillatory shear stress (OSS; athero-prone) condition compared with laminar shear stress (LSS; athero-resistant) (**Supplementary Fig. 1d**).

- We confirmed that inflammatory activation was also a trigger for OASL1 and conducted the experiments using TNF α and IFN γ as stimulators, which mimic the conditions of endothelial activation which underlies atherosclerosis and induces the expression of OASL1 (**Fig. 1h-k**). Although further research is needed regarding the mechanisms of expression in athero-prone regions, the combinatorial effect of various influencers may influence the expression pattern. This is discussed in the Discussion section (lines 416-418) and refer to comment 6 below for more details.

Major specific comments

5. Fig. 1a Regarding the expression of human plaques of OASL1: is it expressed on the luminal endothelium or on neo-vessels that are enriched in the human atheroma.

- Based on the reviewer's comment, we measured OASL expression in other human atheroma tissues and found that it was detectable in the luminal endothelium (**Fig. 1b**) and CD31⁺ ECs of neo-vessels (**Supplementary Fig. 1b**). We first focused on luminal endothelial cells (ECs), however, we identified that human atheroma tissues enriched with neo-vessels may be an additive source of OASL expression during atherogenesis, based on the reviewer's specific comment. These findings were added to Fig.1 and Supplementary Fig. 1 and described in the Results section (lines 134-135) of the revised manuscript.

Fig. 1

Supplementary Fig. 1

6. The mechanisms of upregulation of OASL1 at athero-prone regions is unclear. Is OASL1 expression dependent on hemodynamic or inflammatory changes in the athero-prone regions of the aorta?

- First, in vivo OASL1 expression was up-regulated in athero-prone regions of the aortas of *Apoe*^{-/-} mice (*Apoe*^{-/-}; Fig. 1f, g), demonstrating the contribution of the hemodynamic effect. It is well known that the athero-prone flow-mediated shear response also triggered pro-inflammatory activators (Cybulsky et al. *Arterioscler Thromb Vasc Biol.* 2014; Gimbrone Jr et al. *Circ Res.* 2016). Consistently, OASL1 expression was associated with inflammatory changes in the athero-prone regions, because endothelial OASL1 expression was significantly increased following stimulation with TNF α and IFN γ (Fig. 1h–k).

Fig. 1

- Based on the reviewer’s comment, we conducted an additional experiment to determine whether flow-mediated hemodynamics are attributable to the upregulation of OASL1 at athero-prone regions. To utilize the hemodynamic system, we had no choice but to proceed using HUVECs because of

the limitation to our experimental system. The results showed that OASL expression is upregulated by oscillatory shear stress (OSS; athero-prone) compared with the laminar shear stress (LSS; athero-resistant) condition (**Supplementary Fig. 1d**).

Supplementary Fig. 1d

- Although we need to further demonstrate the additive effects of hemodynamics on inflammatory stimuli, we partially identified that the combinatorial effect of athero-prone shear and inflammatory activation in the athero-prone regions trigger a distinct pattern of OASL1 expression. These are addressed in the Discussion section (lines 420-423) of the revised manuscript.

[Geometric features of the vasculature elicit distinct blood flow pattern-mediated endothelial WSS at regions of arterial bends and branch points known as athero-prone sites.³⁷ Although we need to further demonstrate the additive effect of hemodynamics and inflammatory stimuli, athero-prone, the flow-mediated shear response triggered pro-inflammatory activators which drove the unique OASL1 expression pattern.³⁸⁻⁴⁰]

7. Does the expression of OASL1 in athero-prone regions predates the development of atherosclerotic lesions?

- Yes. We checked OASL1 expression in endothelial cells (ECs) considering the regional specificity in the aortas of 8-weeks old *Apoe*^{+/+} and *Apoe*^{-/-} mice. We found that OASL1 expression was increased in the athero-prone, but not in the athero-resistant regions of aortas from *Apoe*^{-/-} mice compared with the *Apoe*^{+/+} controls (**Fig. 1f, g**), indicating that OASL1 is primarily expressed in athero-prone regions from the early stages before atherosclerotic plaque development.

Fig. 1

8. Suppl. Fig. 6. Do the haemodynamic changes appear before the inception of atherosclerotic lesions and lumen impingement? In other words, are those changes primary or secondary to the lesions formation?

Supplementary Fig. 5

– Yes, they appear before the inception of atherosclerotic lesions.

More specifically, the changes in blood velocity and WSS from 8- to 20-weeks-old were primary to lesion formation at 28 weeks of NCD,

particularly in the athero-prone region, including the aortic arch and abdominal region of aortas. *Oas1* deficiency in *ApoE*^{-/-} mice showed a lower velocity and WSS compared with controls, suggesting they are more prone to attract leukocytes and promote lesion formation. These findings are described in the Results section (lines 194-200) of the revised manuscript.

9. Fig. 2c: the authors should comment on the changes in specific populations and their identity in the single cell experiment. For instance: How does OASL1 regulates endothelial cell heterogeneity in single cell Fig.. What is the relative/absolute reduction in smooth muscle cells? What is the meaning and the increase in macrophage cluster 1? An increase in NKT cells and a slight decrease in ECs is also noted. What is their significance to the phenotype?

– We hypothesized that when *Oas1* is deficient, endothelial cell (EC) heterogeneity would be revealed because of changes in the downstream target of OASL1. We analyzed the endothelial sub-cluster to determine the characteristics of the cluster following *Oas1* deficiency and identified that the heterogeneity was partially caused by a decrease in the eNOS/NO bioavailability. Further analysis identified an increase in the expression of several apoptosis-related genes in *Oas1*-

deficient ECs following chronic inflammatory conditions, which was mediated by a decrease in *Nos3* levels.

Fig. 4

- Regarding a reduction in smooth muscle cells (SMCs), apoptosis of SMCs is associated with plaque rupture, arterial injury, and restenosis, which is considered a secondary phenotype of disease states (Bennett. *Cardiovasc Res.*, 1999). In addition, there was a report regarding enhanced susceptibility against apoptosis in SMCs following *Nos3* deficiency (Kim et al. *BMB Rep.*, 2019), which could partially explain our single cell RNA sequencing results. The cause of the increase in macrophage cluster 1 and NKT cells was considered to be mediated by an increase in the infiltration of immune cells into lesions because of dysfunction and inflammatory activation of ECs as one of the major phenotypes in atherosclerosis. These observations are described in the Results section (lines 175177) of the revised manuscript.

[These analyses revealed that *Oas1* deficiency increased the number of aortic leukocytes, including macrophages and T cells while reduced smooth muscle cell contents, which was considered as secondary phenotype of plaque progression (Fig. 2c).]

10. Fig. 4. The blood pressure is elevated in OASL1 deficient hyperlipidemic mice. Is blood pressure increased in OASL1-deficient mice without out loss of ApoE? Is the vascular and hemodynamic phenotype dependent or independent of hyperlipidemia? Are the changes in blood pressure contributing to the increase in atherogenesis? Is *Nos3* involved in the blood pressure changes?

- As mentioned in comment 1, only *Oas1* deficiency without loss of *Apoe* did not increase BP (Supplementary Fig. 10b); however, hemodynamic changes including lower blood velocity and WSS were already observed in *Oas1*-deficient mice independent of hyperlipidemia, which was the same as *Nos3*-deficient mice (Supplementary Fig. 10c, d). This result indicates that the hemodynamic phenotype of *Oas1*^{-/-} mice is associated with endothelial dysfunction derived from a

reduction in *Nos3* expression. These results are presented in Supplementary Fig.10 and described in the Results section (lines 255-263) of the revised manuscript.

Supplementary Fig. 10

Fig. 5 and Supplementary Fig. 8

– As mentioned above, elevated BP in *Oas1*^{-/-}*Apoe*^{-/-} mice is dependent on hyperlipidemia, which was a secondary or concurrent phenotype together with increased plaque formation (Fig. 5c, Supplementary Fig. 8a). Collectively, following *Oas1* deficiency, increased endothelial dysfunction reciprocally mediates the hemodynamic phenotype and vascular resistance, which promotes the athero-prone environment and further leads to plaque formation and increased BP.

11. Suppl. Fig. 2. Is Tie2-cre the best tool to target endothelial cells?

– Based on the comments of the reviewer, we selected *Cdh5*-cre/ERT2 (VE cadherin-Cre) as the best tool to target endothelial *Oas1* expression and conducted additional experiments to verify the

endothelial specific knockout effect of *Oas1* on atherogenesis. Using this model, we found that endothelial cell (EC)-specific *Oas1* deficiency exacerbated atherosclerosis after 13 weeks of a

western diet. These results are presented in Fig. 3 and Supplementary Fig.6 and described in the Results section (lines 209-213) of the revised manuscript.

Fig. 3

Supplementary Fig. 6

[We examined the role of OASL1 in ECs using a *Cdh5-cre/ERT2* system (Fig. 3e, Supplementary Fig. 6) and *Oas1^{fl/fl}* *Cdh5-cre/ERT2⁺* *Apoe^{-/-}* mice showed increased lesion formation compared with *Oas1^{+/+}* *Cdh5-cre/ERT2⁺* *Apoe^{-/-}* mice (Fig. 3f). This indicates that the EC-specific deficiency of *Oas1* promotes atherosclerosis in *Apoe^{-/-}* mice, further

demonstrating a potential endothelial effect.]

12. Additional experimentation would be useful to show that the in vivo phenotype is mediated by *Oas1*-dependent regulation of *Nos3*. What is the phenotype of *Nos3* deficient mice? Would they phenocopy the OASL1 ko mice?

→ As mentioned in comment 2, we conducted additional mechanistic studies by reducing *NOS3* expression. When comparing the inflammatory pathways affected by both *Oas1* deficiency and OASL-KD, *NOS3*-KD exerted an additive effect on inflammatory activation that contributes to the in vivo inflammatory phenotype in *Oas1* deficiency (Supplementary Fig. 15). Please refer to the answer in comment #2 for more details.

- Regarding phenotypes, hemodynamic changes including a reduction in blood velocity and WSS identified in *Oasl1*-deficient mice were equally observed in *Nos3*-deficient mice (**Supplementary Fig. 10c, d**). The percentage of fractional area change was also reduced in both the *Oasl1*- and *Nos3*-deficient mice, indicating decreased tension and a fundamental endothelial malfunction phenotype mediated by *Oasl1*-dependent regulation of *Nos3*. We observed that the hyperlipidemic condition promoted endothelial dysfunction in *Oasl1*-deficient mice, resulting in the elevation of BP, which was the case in *Nos3*-deficient mice (Knowels et al. J Clin Invest. 2000; Haperen et al. J Biol Chem. 2002), although this was not observed in the absence of hyperlipidemia (**Supplementary Fig. 8a, Supplementary Fig. 10d**). Moreover, it is known that *Nos3*-deficient *Apoe*^{-/-} mice exhibited increase in plaque formation and leukocyte adhesion (Chen et al. Circulation. 2001; Kuhlencordt et al. Circulation. 2001; Ponnuswamy et al. PLoS One. 2012) similar to that in *Oasl1*^{-/-}*Apoe*^{-/-} mice compared with *Apoe*^{-/-} controls.

13. Fig 7a and Suppl. Fig 2 should be consolidated as Main Fig. because the data they contain are highly complementary and key to the message (role of the endothelial expression of *Oasl1*). Fig 7a is missing *Oasl1*^{+/+} bone marrow controls, while Suppl. Fig. 2 is missing the conditions of *Oasl1*^{-/-} in both hematopoietic and non-haematopoietic cells. The effect of the global ko on lesions size (5%) seems inferior to the non-hematopoietic deficiency of *Oasl1* (10%). I suggest also adding diagrams to help the readership navigate complex chimera experimental designs.

- We concur with the reviewer's comment that bone marrow transplantation (BMT) and conditional knock out (cKO) data must be present in the main Figures. We added experimental diagrams together with the results; however, we designed previous supplemental Fig. 2 as a match to the corresponding models between BMT and the cKO system. Moreover, following the comments of the reviewer, we produced more concrete results regarding the endothelial-specific effect of *Oasl1* using *Cdh5*-cre/ERT2 mice. Therefore, we believe that there is no longer a need to include all of the BMT data together.
- We re-configured previous Fig. 7 to show the protective effect of vascular OASL1 in vivo during atherogenesis. Thus, we examined endothelial WSS, leukocyte population, adhesion, and infiltration together using this model and kept these figures together.

Minor comments

Line 134 (Suppl. Fig. 1a). The definition of what constitutes EC-rich human tissue is unclear. Pls rephrase indicating aorta and coronary artery as appropriate.

- We revised the manuscript based on the reviewer's comments. See lines 137-138 of the revised manuscript.

Line 181 why not all samples are selected? Its unclear why a response-dependent selection is used.

- Endothelial cells (ECs) vary in their characteristics according to their location, which contain a relatively small number of cells. Therefore, we first wanted to know the specific characteristics of athero-prone ECs affected by *Oas1* deficiency and selected samples for single cell RNA sequencing (scRNA-seq) from aortic arch and ascending aortas. If we used whole aortas, it would yield broad and variable transcriptomic data for ECs based on their variation in location. Based on the reviewer's comment, we plan to conduct an scRNA-seq of every EC population existing in the whole aorta and we look forward to identifying other targets or mechanisms of endothelial OASL1.

We apologize for the confusion and have revised the manuscript based on the comments. Line 402 Sentence is unclear.

- We revised the manuscript based on the reviewer's comments. See lines 436-439 of the revised manuscript.

[Although OASL1 was markedly increased in aortic plaques, which contain both activated ECs and infiltrating immune cells, the minimal atherosclerotic environment formed in young *Apoe*^{-/-} mice sufficiently triggered the expression of OASL1 in the endothelium, suggesting that vascular ECs are the fundamental *Oas1*-expressing aortic cell type.]

Line 426 Sentence is unclear.

- We revised the manuscript based on reviewer's comments. See lines 461-463 of the revised manuscript.

[Interestingly, both *OASL* and *NOS3* expression were dependent on PI3K/Akt signaling, and activation of this signaling inhibits the inflammatory response mediated by MAPKs and attenuates atherogenesis.]

Line 440 "vitalising" should be replaced

- See line 480 of the revised manuscript.

[identifying]

Fig. 1b The cellular localization of OASL1 should be better rendered. CD31 staining is also observed within the plaque, not only in the endothelial monolayer above the plaque. Are platelets also expressing it? Is it expressed in neovessels?

- Based on the comments of the reviewer, we conducted additional tests using immunofluorescence (IF) staining of human tissues with atheroma to support our previous results. Following these experiments, we were able to present more precise results regarding the relationship with human atherosclerosis. Human OASL was expressed higher in the arteries with plaques compared with tissues without plaques (Fig. 1b) and was stained in CD31⁺ neovessels contained in tissues with plaques (Supplementary Fig. 1b). These results are presented in Fig. 1 and Supplementary Fig.1.

- CD31 was observed within the plaque because the stain adhered well to the media or elastic layer to a degree, which may have created false positive results. Nonetheless, one can see the major CD31 staining in the EC layer of atheroma tissues. CD31 may also be stained in the plaque areas as it is a platelet and endothelial cell adhesion molecule 1 (PECAM1) and is found on the surface of platelets, although we have to further demonstrate their co-localization, specifically in platelets.

Fig. 1f “origin of the artery” is unclear. Do the authors mean the “aortic root”?

- The origin of the artery refers to the starting point of the arterial branch and the precise site is depicted below. We found and referred to a previous report (Yang et al. Nat Commun. 2018).

However, the manuscript was modified the words 'branch point of artery' to avoid confusion compared with the aortic root. See Fig. 1g of the revised manuscript.

(Yang et al. Nat Commun. 2018 Nov 7;9(1):4667.)

Fig. 1h: why is the stimulation with both IFN and TNF needed for upregulation? What is the effect of the single stimulation?

- We used both TNF α and IFN γ to mimic the endothelial activation environment under chronic inflammatory conditions during atherogenesis. Therefore, we adapted both stimulators in every experiment existed in our study. These were previously used in other studies of endothelial stimulation (Jaczevska et al. J Leukoc Biol. 2014). Based on the reviewer's comments, we performed additional experiments using a single stimulator and found that using TNF α and IFN γ showed a more powerful effect on the expression of OASL1 compared with TNF α or IFN γ alone.

(data not shown in manuscript)

Fig. 3g the relevance of the EC subclusters is unclear.

- We did not divide the endothelial cell (EC) subcluster with previously reported canonical marker genes, which were categorized by EC characteristics, but divided the clusters based on unbiased

differentially expressed genes (DEGs). Total ECs were sub-clustered using 6 significant principal components (PCs) and then used in all downstream analyses. The analysis method is described in the Methods section (lines 594-617) of the revised manuscript.

Fig. 3i Was *Nos3* differentially regulated by loss of *OASL1* in vivo?

- When we confirmed *Nos3* levels in our in vivo scRNA sequencing results, *Nos3* was clearly detected only in the EC cluster of atherosclerotic aortas. Although we obtained only tendency of reduction in the EC cluster, *Nos3* was significantly decreased in whole aortas by loss of *Oas11*.

Fig. 4c: was the blood pressure measured invasively or non-invasively?

- We measured blood pressure (BP) using the CODA noninvasive BP system (a tail-cuff Method, Kent Scientific Corporation) according to the manufacturer's instructions. This tail-cuff system utilizes a Volume Pressure Recording (VPR) to measure BP by determining the tail blood volume.

Fig. 5 pERK: large variability in the data set

- We checked our quantitation data for band intensity and found errors in the replicate values. The content was modified in Fig. 6d of the revised manuscript.

Reviewer #2 (Remarks to the Author):

We appreciate your valuable comments regarding our study. Your comments have helped to improve our manuscript. Below, we provide answers to your comments point by point. The original referee's comments are shown in black, whereas our answers are in blue and appropriate changes in the revised manuscript are green. In particular, the square box indicates the overall summary of our conclusion for the corresponding question.

This is an interesting and detailed paper describing the effects of OASL1 in atherosclerosis. OASL1 has been previously shown, and is confirmed in further experiments done by the authors in the present study, to be a regulator of multiple gene pathways, particularly those involved in inflammation. The authors focus on a potential link between OASL1 and endothelial nitric oxide synthase [NOS3] function mediated by alteration of NOS3 mRNA stability.

Although potentially interesting, and including an impressive number of experiments, there are some fundamental limitations of the study that undermine the validity of some of the authors conclusions, and fall short of demonstrating a causal role for an OASL1? eNOS - NO pathway in atherosclerosis. These include:

1. the choice of mouse lines to test global, bone marrow, endothelial cell and vascular deletion of OASL1 are problematic and are not applied in a consistent and valid manner throughout the experiments. For example, bone marrow chimeric mice, although widely used to test the effects of bone marrow versus non bone marrow deletion, have potential limitations in atherosclerosis because the repopulation of the irradiated recipient bone marrow with transferred cells will inevitably alter the immune repertoire. The presumption that transplantation of knockout bone marrow in to non knockout recipients will create a vascular cell specific knockout is completely invalid, even less for creation of an endothelial cell specific knockout. This approach, at best, will enable comparison of bone marrow derived versus all other cell type knockouts. The Tie2cre driver is also used as a model of EC knockout, but it is now widely known that Tie2cre is also highly effective at deleting floxed alleles in all bone marrow cells, indeed in the haematology literature it is used as a bone marrow specific Cre driver line. It would be preferable to use a more recently characterized and validated EC specific create driver such as VE Cadherin Cre. Apart from these technical limitations, the authors are too imprecise and vague in their use of terminology, implying presumptions which are invalid and greatly undermine their experiments. For example, describing bone marrow chimeric animals as "myeloid specific" KO is completely incorrect. They are bone marrow KOs, which will include all bone marrow cells and any other cells in the mouse that are derived from bone marrow, which may even include some vascular cells, given the clear evidence for vascular re-population by bone marrow derived stem cells particularly in atherosclerosis. Testing a myeloid specific knockout would require Cre drivers such as LysM Cre.

→ Based on the comments of the reviewer, we selected the Cdh5-cre/ERT2 (VE cadherin-Cre) as the best tool to target endothelial *Oas1* expression and conducted additional experiments to verify the endothelial specific knockout effect of *Oas1* on atherogenesis. Using this model, we demonstrated that endothelial cell (EC)-specific *Oas1* deficiency exacerbated atherosclerosis after 13 weeks of a western diet. These results are presented in Fig. 3 and Supplementary Fig.6 of the revised manuscript.

Fig. 3

Supplementary Fig. 6

→ We previously explained the bone-marrow (BM)-specific *Oas1* deficiency and myeloid-specific *Oas1* deficiency together in one sentence at the same time and did not clearly indicate the genotype of the mice. We apologize for the confusion regarding this point. We used both a BMT model (Fig. 3a, b) and the LysM Cre (Lyz2 cre) (Fig. 3c, d) system to identify the BM and myeloid-specific deletion effect of *Oas1* during atherogenesis. More detailed information regarding the mice is provided in the Methods section and these results are presented in Fig. 3 and described in the Results section (lines 201-213) of the revised manuscript.

Fig. 3

[To confirm the endothelial specificity of atherosclerosis, we generated chimeric *Oas1*^{-/-}*Apoe*^{-/-} mice with a vascular EC-specific deficiency of *Oas1* by transplanting *Apoe*^{-/-} bone marrow (BM) into *Oas1*^{-/-}*Apoe*^{-/-} recipient mice (*VcOas1*^{-/-}*Apoe*^{-/-} mice) (Fig. 3a) and subsequently crossing *Oas1*-floxed (*Oas1*^{fl/fl}) *Apoe*^{-/-} mice with *Tie2-Cre* transgenic mice to generate *Oas1*^{fl/fl}*Tie2-cre*⁺*Apoe*^{-/-} mice (Fig. 3c). An *en face* analysis revealed that plaque formation was increased in whole aortas of *VcOas1*^{-/-}*Apoe*^{-/-} mice compared with those of *Apoe*^{-/-} chimeric mice ($P = 0.0145$; Fig. 3b) whereas BM-*Oas1*^{-/-}*Apoe*^{-/-} chimeric mice exhibited no differences. In addition, plaques were significantly larger in the *Oas1*^{fl/fl}*Tie2-cre*⁺*Apoe*^{-/-} mice compared with the littermate control *Oas1*^{fl/fl}*Apoe*^{-/-} mice ($P = 0.0009$; Fig. 3d), whereas they were no different in *Oas1*^{fl/fl}*Lyz2-cre*⁺*Apoe*^{-/-} mice. We examined the role of OASL1 in ECs using a *Cdh5-cre*/*ERT2* system (Fig. 3e, Supplementary Fig. 6) and *Oas1*^{fl/fl}*Cdh5-cre*/*ERT2*⁺*Apoe*^{-/-} mice showed increased lesion formation compared with *Oas1*^{+/+}*Cdh5-cre*/*ERT2*⁺*Apoe*^{-/-} mice (Fig. 3f). This indicates that the EC-specific deficiency of *Oas1* promotes atherosclerosis in *Apoe*^{-/-} mice, further demonstrating a potential endothelial effect.]

2. It is not clear why OASL1 deletion leads to changes in blood flow velocity, rather than the expected biological effect of disturbing endothelial cell inflammatory responses, ie altered inflammatory responses to the same level of blood flow or shear stress. Given that the authors state that heart rate and cardiac output we're not different in the OASL1 knockout, why would it be expected, and how is it explained, that blood flow velocity, or the hemodynamic effects of blood flow on the vascular wall, are altered? In the absence of any change in cardiac output, this would have to implicate a change in vascular dimensions, geometry or physico-elastic properties. This whole area of observations needs to be critically re-examined and supported by objective explanations.

- Based on the other reviewer's comments, we found that *Oas1* deficiency with and without hyperlipidemia resulted in a lower mean blood velocity and WSS, particularly at the lesser curvature of the aortic arch (**Supplementary Fig. 5, Supplementary Fig. 10c**), indicating endothelial dysfunction that may be the cause of vascular functional impairment.
- To determine the rationality of decreased blood velocity and WSS following *Oas1* deficiency, we examined various vascular characteristics in the 16- to 20-week-old mice when velocity and WSS differed. First, there were no changes in the vascular maximum diameter (**Supplementary Fig. 10d, g**) or vascular geometry (**Supplementary Fig. 10i**) by the absence of *Oas1*. Therefore, we determined the degree of vascular tension by measuring vascular fractional area change using an ultrasound measurement, which may be another cause of blood flow change. *Oas1* deficiency decreased the degree of fractional area changes, which reflects the augmentation of vascular resistance and consequent decrease in flow velocity (**Supplementary Fig. 11f, h**). This was also thought to be reflected in the phenotypes, including a reduction of vascular relaxation and an increase in constriction (**Fig. 5d, Supplementary Fig. 11b**). In addition, *Nos3*-deficient mice phenocopied *Oas1*-deficient mice, suggesting that these phenomena were derived from attenuation of eNOS-NO bioavailability following the loss of *Oas1*. The collective implication of these new findings underlying *Oas1* deficiency are addressed in the Discussion section (lines 427-431) of the revised manuscript.

[When we checked site-dependent changes in vascular function and the absence of *Oas1* reduced the percentage of the vascular fractional area change, followed by a decrease in blood velocity and endothelial WSS from the early phase of atherosclerosis. These phenomena mediated a more athero-prone environment that attracted leukocyte infiltration, promoted lesion formation and vasoconstriction with an increased BP.⁴³]

3. The focus of the study is on NOS3 mRNA levels and stability, which are certainly tested very thoroughly, but the biological link with and eNOS protein, enzymatic activity and NO bioavailability is very weakly evidenced. In line 235, the authors jump straight from gene expression network analysis implicating endothelial nitric oxide synthase, (which is bound to be highly annotated and overrepresented in endothelial cell gene expression data sets) to a causal role for an NO bioactivity which is at best speculative, and subsequently only weakly supported by later experiments. In particular, the hallmarks of and the NO effects and bioactivity have not been addressed. The use of DAF immunofluorescence is not adequate as a quantitative readout of eNOS activity or NO bioactivity. The critical importance of this conclusion requires support by complementary methods, given that this is known to be a technically challenging read out. This might include radio labelled arginine to citrulline conversion, measurement of cyclic GMP, EPR spin trapping techniques to detect authentic NO. Also functional studies should be done of endothelial function on isolated conduit and resistance vessels to quantify some of the classic read out of endothelial NOS activity such as relaxation responses to appropriate agonists or flow. Similar studies should be done in ECs.

Oas1 deficiency reduced eNOS enzymatic activity, NO bioavailability, cGMP levels, and vascular tension, which were dependent on eNOS expression, and exert endothelial dysfunction-mediated vascular impairment.

- We apologize for the inconvenience and try to refrain from exaggerating expression. Based on the reviewer's comments, we performed additional experiments that provide support for the proposed *Oas1*-associated molecular mechanisms associated with the eNOS-nitric oxide (NO) bioavailability.
- To avoid overstatement and explore the mechanistic link between OASL1 and the NO biosynthetic process, we first identified the enzymatic activity of eNOS and it was reduced in whole aortas isolated from *Oas1^{-/-}Apoe^{-/-}* mice (**Fig. 5b**). Moreover, both *Oas1^{-/-}* MAECs and OASL-KD HUAECs exhibited decreased enzyme activity relative to the controls as was the case with L-NAME (eNOS inhibitor) treatment (**Supplementary Fig. 13c, d**), indicating an eNOS-dependent mechanism mediated by endothelial OASL1. In addition, OASL-KD HUAECs also showed a decrease in eNOS protein levels (**Supplementary Fig. 15a**).

Fig. 5 and Supplementary Fig. 13

Supplementary Fig. 15

→ Additionally, NO levels were directly quantified by ELISA using the Griess method to compare the results with that of the EC sub-clustering analysis. Through measurement of both nitrite and nitrate for determining total NO levels in the plasma (Fig. 5a) and supernatants of cultured ECs (Fig. 5i, m, Supplementary Fig. 13e, f), we could identify NO levels more directly, which were decreased by the absence of *Oas1*. Moreover, cyclic GMP levels, which indirectly reflect NO bioactivity, were also reduced in *Oas1*-deficient aortas (Supplementary Fig. 13b) and in vascular smooth muscle cells following co-culture with *Oas1*-deficient ECs compared with wild-type ECs (Supplementary Fig. 13g, h). Collectively, these findings are described in the Results section (lines 240-245, 284-287, 294-301) of the revised manuscript.

Fig. 5 [As shown in the EC sub-clustering analysis of scRNA-seq data, one of the hallmarks of endothelial dysfunction is attenuation of eNOS/NO

bioavailability accompanied by enhanced immune response.²⁵ Therefore, we determined whether decreased eNOS/NO bioavailability was reflected as a reduction in NO levels. NO levels in atherosclerotic plasma ($P = 0.0442$; Fig. 5a) and

aortic NO synthase enzymatic activity ($P = 0.0156$; Fig. 5b) in *Oasl1*^{-/-}*ApoE*^{-/-} mice were reduced compared with that of *ApoE*^{-/-} controls, which supports our scRNA-seq results.]

[NO ($P = 0.0002$) and cyclic GMP (cGMP; $P < 0.0001$) production was also reduced in the aortas of *Oasl1*^{-/-}*Apoe*^{-/-} mice compared with *Apoe*^{-/-} mice (Supplementary Fig. 13a, b), suggesting that endothelial *Oasl1* deficiency triggers athero-prone features in aortas through down-regulation of eNOS expression and NO production.]

Supplementary Fig. 13

[Moreover, total NO levels

were reduced in the supernatant of *Oasl1*-deficient MAECs (Fig. 5i) and OASL-KD HUAECs (Fig. 5m) compared with the controls indicating that, similar to murine *Oasl1*, a reduction of human *OASL* decreased *NOS3* mRNA levels in athero-prone ECs, thus lowering NO bioavailability. Concurrently, we determined that pretreatment of cells with L-NAME, an eNOS inhibitor, markedly attenuated eNOS activity (Supplementary Fig. 13c, d) and reduced secreted NO (Supplementary Fig. 13e, f) and intracellular cGMP levels (Supplementary Fig. 13g, h). This was similar to the results observed in the absence of *Oasl1*, which suggests an eNOS-dependent regulatory mechanism for endothelial OASL1.]

→ Based on the reviewer's comment, we conducted endothelial function studies which measured the relaxation response to quantify endothelial NOS activity. *Oasl1* deficiency in *Apoe*^{-/-} mice resulted in the impairment of acetylcholine (ACh)-mediated concentration-dependent relaxation (Fig. 5d), whereas there was no difference in sodium nitroprusside (SNP)-mediated relaxation (Supplementary Fig. 11a), indicating endothelium-dependent results. Consistent with this, *Oasl1* deficiency augments vessel resistance, which is reflected as an increase in vascular constriction in response to phenylephrine (PE) (Supplementary Fig. 11b). Furthermore, ODQ (soluble guanine cyclase inhibitor) treatment reduced vascular relaxation to a similar level between *Oasl1*^{-/-}*Apoe*^{-/-} and *Apoe*^{-/-} mice (Supplementary Fig. 11d), indicating that impairment of vascular function during

Oasl1 deficiency is dependent upon endothelial malfunction, rather than a smooth muscle defect. These findings are described in the Results section (lines 263-269) of the revised manuscript.

Fig. 5 and Supplementary Fig. 11

[Moreover, aortic vascular relaxation was significantly reduced and dependent upon acetylcholine, whereas constriction was increased following a decrease in vascular fractional area change in *Oasl1*^{-/-}*Apoe*^{-/-} mice compared with controls (Fig. 5d, Supplementary Fig. 11). Treatment with the soluble guanylyl cyclase inhibitor, ODO, reduced the percentage of relaxation in both groups by same amount (Supplementary Fig. 11c), indicating that *Oasl1* deficiency results in vascular functional impairment primarily derived from endothelial malfunction, rather than a defect in smooth muscle.]

4. The finding that the OASL1 knockout mouse has a high blood pressure is interesting, and consistent with abnormalities in endothelial cell function. However, this finding raises the potential of confounding in the observed effects on atherosclerosis. Specifically, high blood pressure would be expected to increase endothelial cell activation and increase atherosclerosis, irrespective of the underlying mechanism. Ideally, this should be addressed to see whether the high blood pressure phenotype is specifically mediated by endothelial OASL1, and how the effects on atherosclerosis could be controlled for, with appropriate blood pressure comparisons.

→ Based on the reviewer's comments, we compared time-dependent blood pressure (BP) values under both basal and hyperlipidemic conditions. Only *Oasl1* deficiency did not affect a change in BP (Supplementary Fig. 10d), whereas *Oasl1*-deficient *Apoe*^{-/-} (*Oasl1*^{-/-}*Apoe*^{-/-}) mice showed significantly increased BP at 28 weeks of a normal chow diet (NCD). Because there was no significant difference in the measurements from 8 to 20 weeks, increased BP was secondary or a concurrent phenotype together with increased plaque formation (Fig. 5a, Supplementary Fig. 8a).

Supplementary Fig. 10

Fig. 5 and Supplementary Fig. 8

→ Based on the reviewer's suggestion, we identified the endothelial-specific deletion effect of *Oas1* on pressure overload by comparing the BP of *Oas1^{fl/fl}Cdh5-cre/ERT2⁺Apoe^{-/-}* and *Oas1^{-/-}Cdh5-cre/ERT2⁺Apoe^{-/-}* mice. After a western diet for 13 weeks, *Oas1^{fl/fl}Cdh5-cre/ERT2⁺Apoe^{-/-}* mice showed increased BP compared with the controls (Supplementary Fig. 9), indicating the effect of endothelial OASL1 on the hypertensive phenotype and plaque formation. Collectively, these results are presented in Fig. 5, Supplementary Fig. 8, Supplementary Fig. 9, and Supplementary Fig. 10 and described in the Results section (lines 245-255) of the revised manuscript.

Supplementary Fig. 9

[A decrease in eNOS/NO bioavailability is known to contribute to vascular resistance and augmentation of BP.^{26, 27} *Oas1* deficiency augmented mean blood pressure (BP; $P = 0.0465$ at 28 weeks) following an increase in both systolic ($P = 0.0425$) and diastolic ($P = 0.0085$) BP in *Apoe^{-/-}* mice during atherosclerotic conditions compared with controls (Fig. 5c, Supplementary Fig. 8a), whereas NO supplementation significantly reduced BP in both groups (Supplementary Fig. 8b). There was no difference in heart rate or function, including ejection fraction, fractional shortening, or cardiac output between *Oas1^{-/-}Apoe^{-/-}* and *Apoe^{-/-}* mice (Supplementary Fig. 8c-e). This indicates that the

augmentation of BP in *Oasl1* deficiency was dependent on vascular NO level regardless of heart function. The result of increased BP was repeated in *Oasl1^{fl/fl}/Cdh5-cre/ERT2⁺ApoE^{-/-}* mice compared with the corresponding controls, suggesting an endothelial-specific *Oasl1* effect on hypertension following plaque formation (Supplementary Fig. 9).]

5. As the authors describe, multiple different pathways in endothelial and other cell types are impacted by OASL1. How do the authors have confidence in their conclusion that the effects of OASL1 are mediated specifically by endothelial nitric oxide synthase and NO, rather than the multiple other inflammatory pathways, all of which are credible and/or proven causal pathways in atherosclerosis? Is OASL1 implicated in human GWAS? Would a OASL1 knockout in ECs with eNOS knock down, or in a NOS3 knockout mouse no longer impact the inflammatory pathways or the development of high blood pressure or atherosclerosis?

It is well known that endothelial cell (EC) dysfunction can initiate immune cell infiltration into the intima of the aorta and inflammation resulting in the progression of atherogenesis (Davignon et al, Circ. 2004, Weber et al, Nat Med. 2011). And further our EC-specific knockout studies (Cdh5-Cre/ERT2, and Tie2-Cre) clearly showed that EC-specific *Oasl1* deficiency accelerated atherogenesis.

We agree that we should consider and address other mechanisms, but we did not observe differences in type I interferon (IFN) related pathways in *Oasl1*-deficient ECs. *Nos3* deficiency initiated an inflammatory response by up-regulating adhesion molecules and activating ERK1/2 and NF-κB in addition to *Oasl1* deficiency, which were derived from their anti-inflammatory and protective roles in the vasculature.

- First we checked type I interferon (IFN) related molecules, which were previously known to be associated with and regulated by OASL1. However, aortic type I IFN expression and plasma IFN γ levels were similar between *Oasl1^{+/+}ApoE^{-/-}* and *Oasl1^{-/-}ApoE^{-/-}* mice under atherosclerotic conditions (**Supplementary Fig. 17a, b**). That is why we searched for other targets of OASL1 in endothelial cells (ECs). Moreover, we could not find differences in type I IFN levels, including IFN α and IFN β (**Supplementary Fig. 17d-f**) or the activation of IFN-regulatory factor 7 (IRF7) and IRF3 in ECs following stimulation (**Supplementary Fig. 17g**).
- Furthermore, we examined human ECs by mining our transcriptional data of HUVECs and identified that human OASL-KD also did not affect the level of type I IFN and IFN-stimulated genes (**Supplementary Fig. 16a, b**). Consistent with the results of MAECs, secreted IFN γ levels (**Supplementary Fig. 16c**) and the activation of IFN-related signaling pathways (**Supplementary**

Fig. 16d) were not different between *OASL*-KD and control HUAECs, suggesting the presence of other inflammatory-associated pathways that drive *Oas1* deficiency-mediated endothelial dysfunction. These data have been added in Supplementary Fig.16 and in Supplementary Fig.17 and are described in the Results section (lines 342-346) of the revised manuscript.

Supplementary Fig. 17

Supplementary Fig. 16

[However, the type I IFN-related pathway, which is a pro-inflammatory mechanism affected by OASL and OASL1, did not significantly change with *Oas1* deficiency in HUAECs and MAECs (Supplementary Fig. 16, 17), demonstrating that pro-inflammatory and adhesion events are aggravated in *OASL*-KD

HUVECs primarily through attenuated eNOS expression and enhanced MAPK and NF- κ B signaling.]

→ There have already been reports in several papers that a reduction in Nos3-eNOS expression intensifies inflammatory activation and leukocyte adhesion during atherosclerosis (Lefer et al. Am J Physiol. 1999; Otsuka et al. Nat Rev Cardiol. 2007). Based on the reviewer's comment, we conducted additional mechanistic studies by reducing NOS3 expression. *Oas1*^{-/-} MAECs were transfected with *Nos3* siRNA to knockdown (KD) *Nos3* and HUAECs were concurrently transfected with siOASL and siNOS3. When we compared the inflammatory pathways affected by both *Oas1* deficiency and OASL-KD, NOS3-KD had an additive effect on the reduction of OASL with respect to inflammatory activation because of its anti-inflammatory and protective effects on NOS3 (Supplementary Fig. 15). These results are presented in the Results section (lines 325-330).

Supplementary Fig. 15

[Reduced eNOS in ECs exerts a variety of physiological responses including an increase in inflammatory signaling during pathological conditions.^{25, 28} Whether the OASL-mediated decrease in *NOS3* affects inflammatory activation was further confirmed by *NOS3*-KD in HUAECs and MAECs. *OASL/NOS3*-KD HUAECs and *Nos3*-KD/*Oas1*^{-/-} ECs increased Erk1/2 and NF-κB activation with up-regulation of adhesion molecules including *ICAM1* and *SELE* compared with *OASL*-KD or *Oas1*^{-/-} and the controls (Supplementary Fig. 15).

- Our data suggested that reduced *NOS3* expression following a reduction of *OASL* (Fig. 7b) preceded the activation of inflammatory pathways, such as Erk1/2 or NF-κB (Fig. 6d, e), and increased adhesion molecule expression (Fig. 6g), which may be a possible cause of later inflammatory responses (Fig. 2d, 4a). Moreover, the expression of *Nos3* mRNA and eNOS protein was already reduced in the *Oas1*-deficient endothelium from 8-week-old *ApoE*^{-/-} mice before disease progression (Fig. 5f, g). Thus, we could infer that inflammatory signaling was accelerated in *NOS3*-KD with *Oas1*^{-/-} or *OASL*-KD ECs compared with only *Oas1* deficiency. Collectively, increased leukocyte infiltration, the inflammatory response, and plaque formation corresponding to the final in vivo phenotypes in *Oas1*-deficient mice may be caused by augmentation of endothelial dysfunction following a reduction of endothelial *Nos3* levels during early atherosclerosis.

6. Similarly, what are the other known or potential targets for Mir584? What are the effects of Mir584 agomirs or antagomirs in *NOS3*-deleted cells?

- There are several potential targets of microRNA-(miR-) 584 including rho-associated protein kinase 1 (Rock1; Ueno K, et al. Br J Cancer. 2011), metalloproteinase 14 (Mmp14; Xiang et al. Biochim Biophys Acta. 2015), Sp7, Runx2, and Col1a1 (Wang et al. Front Genet. 2021). These were already published or predicted with their sequence and putative binding by in silico analysis.
- Based on the reviewer's helpful comment, we first identified the global changes in gene expression following *OASL*-KD, which were predicted as targets of miR584. After setting the cumulative weighted context++ score of TargetScan (<-0.4) as a cutoff, DESeq2 results were derived only for genes present in bulk RNA sequencing. The genes that passed the cut-off criterion were not clearly clustered in relation to the biological pathway. Moreover, considering the genes that passed the p < 0.05 criterion, genes that could be involved in inflammatory pathway were not identified.

(data not shown in manuscript)

- Therefore, we specifically checked the difference in expression of the predicted and reported target genes of miR-584 associated with inflammation or endothelial function.
- For this, we transfected the miR-584 mimic or inhibitor in *OASL*-KD HUAECs with or without *NOS3*-KD, and evaluated the expression of genes including *CXCL5*, *CCL5*, *VEGFC*, *PDGFRA*, and *LIPG* as predicted, as well as *ROCK1* and *MMP14* as reported. As a result, *NOS3* levels were decreased in *OASL*-KD or miR-584 mimic transfected HUAECs, but there was no significant difference in the expression of other candidate genes compared with the controls.
- Moreover, there was no difference in candidate gene expression between *OASL*-KD and the *OASL/NOS3*-double KD HUAECs, indicating little effect of competitive activity between *NOS3* and other genes. We checked that the expression of most of candidate genes exhibited a tendency for reduction by miR-584 mimic treatment, but they were increased following administration of the miR-584 inhibitor. The small difference in the expression of previously reported target genes of miR-584 appears may be the result of variation in experimental conditions or cell types.
- Collectively, we confirmed that the expression of the candidate genes of miR-584 was not affected by *OASL*- and *NOS3*-KD, and that above all, inflammation-related target (candidate) genes could not support the results of our study, because the inflammatory response increased in the *OASL*-KD HUAECs was mediated by a decrease in protective factors caused by an increase of miR-584.

(data not shown in manuscript)

Reviewer #3 (Remarks to the Author):

We appreciate your valuable comments regarding our study. Your comments have helped to improve our manuscript. Below, we provide answers to your comments point by point. The original referee's comments are shown in black, whereas our answers are in blue and appropriate changes in the revised manuscript are green.

In this study the authors uncover a regulatory role of endothelial expression of 2'-5' oligoadenylate synthetase-like 1 (OASL1) in maintaining eNOS mRNA stability under athero-prone conditions and consider its clinical implications. Lacking endothelial Oasl1 accelerated plaque progression, an effect preceded by endothelial dysfunction, following an elevated inflammatory response accompanied by decreased NO bioavailability, reflecting impaired eNOS expression. Mechanistically, knockdown of PI3K/Akt signaling-dependent OASL expression induced an increase in Erk1/2 and NF-κB activation and decreased the NOS3 mRNA levels by up-regulating the negative regulatory, miR-584, while miR-584inhibitor rescued OASL knockdown effects. Their findings suggest that OASL1/OASL is a novel protector of NOS3 mRNA and that targeting miR-584 could be a viable therapeutic strategy for eNOS maintenance in vascular diseases.

Although the paper is well written and data support conclusion I have several issues with the manuscript:

Fig 1: The authors focus in human tissues on atherome plaques in the aorta, will it possible to show data from more distal arteries like coronary in which atherosclerosis will have a greater impact. N=3 should be increase to 5 and quantifiacion of the IF should be provided. Finally, western blot should be performed.

- We first apologize that we could not acquire distal coronary arteries, because of the problem in sample approval by the Institutional Review Board. Nonetheless, we conducted additional tests including immunofluorescence (IF) staining and western blot (WB) analysis of human tissues with atheroma to support the previous results based on the comments of the reviewer. We used additional thoracic and abdominal aortic tissue samples and increased sample size from 3 to 5. WB analysis also revealed that OASL was expressed in tissues with atheroma, rather than without plaques, which was consistent with the IF results (Fig. 1b, c). These results are presented in Fig. 1 and Supplementary Fig.1 and described in the Results section (lines 134-135) of the revised

manuscript.

Fig. 1
[Immunofluorescence (IF) and immunoblot analyses confirmed that OASL was primarily expressed in ECs of plaque-containing atheroma tissues (Fig. 1b, c, Supplementary Fig. 1b).]

Fig 2 do you see any apoptosis of EC?

(data not shown in the manuscript)

- Our study noted that there were more dysfunctional and inflammatory endothelial cells (ECs) following *Oas1* deletion.

Based on the reviewer's comment, we evaluated the expression of apoptosis-related genes

(Casp1, Casp3, Casp7, Casp8, Casp9, Htra2, Aifm1, Bax, Bak1, and Bok) in our scRNA seq database. We discovered that some of the genes, such as Bak1 and Bok, in the *Oas1*-deficient group were up-regulated without showing significant differences in the other genes (data not shown in manuscript). We thought that this may be a secondary effect of decreased *Nos3* levels and increased inflammation as only *Oas1* deficiency or *OASL*-KD caused a reduction in *Nos3* expression even from basal levels (**Supplementary Fig. 19**).

Fig 3: is there any way to quantify directly NO production ?, what are the effects on other NOS expression.

→ Based on the reviewer's comment, additional experiments using the Griess method that directly quantitates nitric oxide (NO) levels by ELISA were conducted to support the previous results. Through the measurement of both nitrite and nitrate for determining total NO level in the plasma (**Fig. 5a**) and supernatants of cultured endothelial cells (ECs) (**Fig. 5l, m, Supplementary Fig. 13e, f**), we identified NO levels more directly, which were decreased by the absence of *Oas1*. These results are presented in Fig. 5 and Supplementary Fig.13 and described in the Results section (lines 294-301) of the revised manuscript.

Fig. 5 Supplementary Fig. 13

supernatant of *Oas1*-deficient MAECs (**Fig. 5l**) and *OASL*-KD HUAECs (**Fig. 5m**) compared to each of controls indicating that, as was the case for murine *Oas1*, reduction of human *OASL* decreases *NOS3* mRNA levels in athero-prone ECs, lowering NO bioavailability. Concurrently, we validated that pretreatment of cells with L-NAME, an eNOS inhibitor, markedly attenuated NOS activity (**Supplementary Fig. 13c, d**) and reduced secreted NO (**Supplementary Fig. 13e, f**) and intracellular cGMP level (**Supplementary Fig. 13g, h**) as was in the absence of *Oas1*, implying eNOS-dependent regulatory mechanism of endothelial *OASL1* .]

→ When we checked whether the reduction of NO production in ECs affects the expression of other NOS genes, *Nos1* (nNos) and *Nos2* (iNOS) levels in the *Oas1*-deficient MAECs and aortas of *Apoe*^{-/-} mice were comparable to the controls (**Supplementary Fig. 12**). This indicated that the *Oas1* deficiency-mediated decrease in NO production is primarily associated with the reduction of Nos3 (eNOS) levels and has little influence on *Nos1* and *Nos2* expression. These results are presented in Fig. 5 and Supplementary Fig.12 and described in the Results section (lines 270-276) of the revised manuscript.

Fig. 5 and Supplementary Fig. 12
 [As a first step toward developing a mechanistic understanding of the

preceding phenomena, we determined whether these changes were attributable to changes in *Nos3* expression. We found that *Nos3* mRNA levels in atherosclerotic aortic tissues of *Oas1*^{-/-}*Apoe*^{-/-} mice were reduced compared with those of *Apoe*^{-/-} controls ($P = 0.05$; Fig. 5e), without affecting the expression of other *Nos* genes (Supplementary Fig. 12). This suggests that a reduction in eNOS-mediated NO synthesis caused by the absence of *Oas1* contributes to an increase in susceptibility of arteries to atherosclerosis following endothelial dysfunction and systemic vascular

resistance.]

Fig 4: what are the effects on cardiac output? Can you reverse the increase in pressure by giving NO to the mice?

- Cardiac output is another major cause of increase in blood pressure (BP); thus, we measured BP concurrently with cardiac output to verify its effect. Furthermore, we checked cardiac function, such as stroke volume and heart rate, which was found to be proportional to the value of cardiac output besides the ejection fraction and fractional shortening. However, *Oas1* deficiency did not affect cardiac function including cardiac output, but increased BP following plaque formation in *Apoe*^{-/-} mice, indicating that a reduction of NO levels mainly affects vascular resistance, rather than cardiac output.

- Following the reviewer's comment, we used DETA NONATE, which was previously shown to supply NO *in vivo*. When 1 mg/kg DETA NONATE was administered to each mouse every 24 hours for 7 days, increased BP was significantly reversed in both *Oas1*^{-/-}*Apoe*^{-/-} and *Oas1*^{+/+}*Apoe*^{-/-} mice, respectively, after a normal chow diet for 28 weeks (Supplementary Fig. 8b). However, we did not observe changes in cardiac output in these models (Supplementary Fig. 8f). These results also supported our findings that *Oas1* deficiency exacerbates endothelial dysfunction and pressure overload by reducing endothelial NO levels following a reduction of eNOS expression, whereas NO supplementation significantly reduced BP and exhibited an eNOS-NO-dependent mechanism. Together, these results are presented in Fig. 5 and Supplementary Fig.8 and described in the Results section (lines 243-253) of the revised manuscript.

Fig. 5 and Supplementary Fig. 8

Supplementary Fig. 8

[A decrease in eNOS/NO bioavailability is known to contribute to vascular resistance and augmentation of BP.^{26, 27}

Oas1 deficiency augmented mean blood pressure (BP; $P = 0.0465$ at 28 weeks) following an increase

in both systolic ($P = 0.0425$) and diastolic ($P = 0.0085$) BP in *Apoe*^{-/-} mice during atherosclerotic conditions compared with controls (Fig. 5c, Supplementary Fig. 8a), whereas NO supplementation significantly reduced BP in both groups (Supplementary Fig. 8b). There was no difference in heart rate or function, including ejection fraction, fractional shortening, or cardiac output between *Oasl1*^{-/-}*Apoe*^{-/-} and *Apoe*^{-/-} mice (Supplementary Fig. 8c-f). This indicates that the augmentation of BP in *Oasl1* deficiency was dependent on vascular NO level regardless of heart function.]

Fig 5: given the effect on AKT signaling, GSK3 a regulator of NFAT a critical TF regulating inflammatory is likely affected. Can you provide data on GSK3 and NFAT?

→ Based on the reviewer's comments, some additional experiments were conducted regarding AKT with GSK3 and NFAT signaling using human aortic endothelial cells. When we examined GSK3 α and GSK3 β phosphorylation, both were inhibited following the inhibition of Akt activation, which indicated GSK3 α/β activation and potential to phosphorylate their targets. NFATc levels were reduced following a decrease in GSK3 α/β phosphorylation, however, the degree of phosphorylation and NFATc level was not significantly different between OASL-KD and control HUAECs following Akt inhibition.

Reviewers' Comments:

Reviewer #1:

Remarks to the Author:

The authors performed a significant amount of work to answer my comments very carefully.

1. They demonstrate that the effect on Oasl1 loss on endothelial dysfunction (higher mean velocity) precedes the atherosclerotic lesion development (Suppl Fig 10) and match the early upregulation of Oasl1 at atheroprone region. This set of new kinetics data is convincing and it goes a long way towards unravelling the complex phenotype. This, together with the EC-specific loss of Oasl1 points towards a specific role of Oasl1 in EC-dependent regulation of atherogenesis.
2. The use of a more appropriate model of arterial endothelial cells in vitro and close the mechanistic loop by measuring NO bioavailability, lending support to the idea that NOS3 regulation secondary to endothelial dysfunction is the linchpin of the phenotype downstream of Oasl1.
3. The authors use also a VECadherin-cre which is much more specific tool strain to target gene deficiency to endothelial cells, significantly enhancing the manuscript and the validity of their conclusions on EC biology.
4. The new data of the authors point towards a role of Oasl1 in mitigating EC activation and dysfunction which in turn leads through NOS3 to changes in NO availability, vascular inflammation, and increased blood pressure and atherogenesis. In other words, Oasl1's functions go beyond mere inflammation but it regulates also endothelial responses and biology with a protective net effect on the vasculature. I feel that this conclusion should be streamlined and better communicated than the authors do in the current discussion. The English language use should be carefully reviewed.

Reviewer #2:

Remarks to the Author:

-

Reviewer #3:

Remarks to the Author:

I wish to congratulate the authors for all the responses to all reviewers. I have no further comments

NCOMMS-21-45303A

2nd revision

Point-by point response

REVIEWERS' COMMENTS

Reviewer #1 (Remarks to the Author):

The authors performed a significant amount of work to answer my comments very carefully.

- We really appreciate your positive review of our revised manuscript.

1. They demonstrate that the effect on Oasl1 loss on endothelial dysfunction (higher mean velocity) precedes the atherosclerotic lesion development (Suppl Fig 10) and match the early upregulation of Oasl1 at atheroprone region. This set of new kinetics data is convincing and it goes a long way towards unravelling the complex phenotype. This, together with the EC-specific loss of Oasl1 points towards a specific role of Oasl1 in EC-dependent regulation of atherogenesis.

2. The use a more appropriate model of arterial endothelial cells in vitro and close the mechanistic loop by measuring NO bioavailability, lending support to the idea that NOS3 regulation secondary to endothelial dysfunction is the linchpin of the phenotype downstream of Oasl1.

3. The authors use also a VECadherin-cre which is much more specific tool strain to target gene deficiency to endothelial cells, significantly enhancing the manuscript and the validity of their conclusions on EC biology.

4. The new data of the authors point towards a role of Oasl1 in mitigating EC activation and dysfunction which in turns leads through NOS3 to changes in NO availability, vascular inflammation, and increased blood pressure and atherogenesis. In other words, Oasl1's functions go beyond mere inflammation but it regulates also endothelial responses and biology with a protective net effect on the vasculature. I feel that this conclusion should be streamlined and better communicated than the authors do in the current discussion. The English language use should be carefully reviewed.

- Thank you so much for your valuable comments. As you pointed out, our study is mainly about the regulatory role of OASL1 regarding endothelial responses and biology with a protective net effect on the vasculature, and we agree that it is the crucial novelty of the study. Thus, we revised the statements of abstract [page 1, line 40 and 44-45] and discussion [page 16, line 395-396, page 17, line 442-443, and page 19, line 472-473] in the 2nd revised version of the manuscript, to better communicate the meaning and novelty of our study.

- Following the reviewer's comment, we had English proofreading of our manuscript.

Reviewer #2 (Remarks to the Author):

-

- We really appreciate your positive review of our revised manuscript.

Reviewer #3 (Remarks to the Author):

I wish to congratulate the authors for all the responses to all reviewers. I have no further comments

- We really appreciate your positive review of our revised manuscript.